# A Kernel Perspective on
# Distillation-based Collaborative Learning

**Sejun Park**    **Kihun Hong**    **Ganguk Hwang**[*]
Department of Mathematical Sciences
Korea Advanced Institute of Science and Technology
{`sejunpark, nuri9911, guhwang`}@kaist.ac.kr

## Abstract

Over the past decade, there is a growing interest in collaborative learning that can enhance AI models of multiple parties. However, it is still challenging to enhance performance them without sharing private data and models from individual parties. One recent promising approach is to develop distillation-based algorithms that exploit unlabeled public data but the results are still unsatisfactory in both theory and practice. To tackle this problem, we rigorously analyze a representative distillation-based algorithm in the view of kernel regression. This work provides the first theoretical results to prove the (nearly) minimax optimality of the nonparametric collaborative learning algorithm that does not directly share local data or models in massively distributed statistically heterogeneous environments. Inspired by our theoretical results, we also propose a practical distillation-based collaborative learning algorithm based on neural network architecture. Our algorithm successfully bridges the gap between our theoretical assumptions and practical settings with neural networks through feature kernel matching. We simulate various regression tasks to verify our theory and demonstrate the practical feasibility of our proposed algorithm.

## 1   Introduction

Collaborative learning of AI models in decentralized settings is an important problem covered in various fields of machine learning such as distributed learning [10, 70], Federated Learning (FL) [23], peer-to-peer learning [3], and miscellaneous collaborative learning [47]. In particular, this theme has been most actively discussed in the context of FL [24, 31, 45, 61]. In this context, each local party is typically viewed as a subordinate entity within the collective learning system. For example, most FL algorithms mandate the exchange of local AI model information among participating local parties. Under this scheme, local AI models are usually subjected to restrictions in their architecture. However, from the perspective of collaboration, each local party may have to be regarded as an independent learning agent, meaning they are not obligated to fully share their model information. In short, the model (or parameter) exchange in FL algorithms can emerge as a critical issue in collaborative learning.

Fundamentally, addressing this issue necessitates an alternative medium for sharing learning information distinct from model exchange. Indeed, Distillation-based Collaborative Learning (DCL) [14, 30, 44] provides a good answer. In these algorithms, local training information is shared via the outcomes of AI models on additional unlabeled public data. The collected information is then utilized for knowledge distillation [21] to each local AI model. As mentioned in [14, 48], this procedure is agnostic to model heterogeneity and avoids the direct sharing of local AI model information. This is a key advantage that distinguishes DCL from traditional FL.

---

[*]Corresponding author

38th Conference on Neural Information Processing Systems (NeurIPS 2024).

Despite its pioneering nature and potential utility, DCL has not been sufficiently explored. A significant reason for this is the lack of theoretical understanding regarding knowledge distillation and its effectiveness in massively distributed statistically heterogeneous environments. Our work stems from the fundamental question of whether DCL algorithms can be theoretically effective in these settings. Inspired by [48, 58], we analyze FedMD [30, 41], the most standard DCL algorithm from a nonparametric perspective. Specifically, we adopt an operator-theoretic approach [5, 15, 37, 53, 65] to obtain an upper rate of convergence for the nonparametric version of FedMD (named **DCL-KR**) in the expected sense. Remarkably, our analysis reveals that DCL-KR achieves a nearly minimax optimal convergence rate, where the prefactor is independent of the number of participating local parties. It is worth noting that DCL-KR is the first nearly minimax optimal collaborative learning algorithm that does not directly share local data or models in massively distributed statistically heterogeneous environments. The novelty of our theoretical results and their comparison to prior works are provided in Section 2 and 3.

Nevertheless, our theoretical analysis does not fully demonstrate the efficacy of DCL algorithms based on neural network architectures. Instead, our theoretical results serve as inspiration for designing a novel DCL algorithm for regression that refines existing approaches. Consequently, we propose a Distillation-based Collaborative Learning algorithm over heterogeneous Neural Networks (named **DCL-NN**) for regression tasks. DCL-NN leverages kernel matching to align the feature kernels from the last hidden layer of each local AI model with an ensemble kernel. This procedure brings heterogeneous neural networks into the regime of DCL-KR.

Finally, we conduct experiments on DCL-KR and DCL-NN. To illustrate the superiority of our algorithms, we compare them with several baselines on various regression tasks. Experimental results show that DCL-KR achieves the same performance as the centralized model, even beyond the theoretical results. We also observe that DCL-NN significantly outperforms previous DCL frameworks in most settings.

In summary, our contributions are as follows:

1. In Section 3, we theoretically prove that a nonparametric version of the most standard distillation-based collaborative learning algorithm (named DCL-KR) is nearly minimax optimal in massively distributed statistically heterogeneous environments.

2. Inspired by the results provided in Section 3, we propose a distillation-based collaborative learning algorithm with heterogeneous neural networks (named DCL-NN) in Section 4.

3. In Section 5, we conduct experiments to empirically confirm our theoretical results and show the practical feasibility of our proposed algorithms.

## 2 Related Work

**Federated Learning** Most FL algorithms [45] communicate model parameters for collaboration. This approach has been extensively studied under various constraints, including data privacy [1], statistical heterogeneity [24, 31], communication efficiency [51], personalization [13, 59], and robustness [25]. While it has been successful both theoretically and experimentally, this type of FL is limited in terms of the privacy and flexibility of local AI models, as the algorithms directly access the structures and parameters of the local models. Our study focuses on distillation-based collaborative learning, where the privacy and flexibility of local AI models are fully guaranteed.

**Distillation-based Collaborative (or Federated) Learning** The type of algorithms we investigate operates by communicating the functional information of local AI models. These algorithms typically assume the availability of additional public data points. In this case, the outcomes of local models on the public dataset are used for collaboration. For instance, Li and Wang [30], Lin et al. [41], Park et al. [48] iteratively collect predictions of local models on the public dataset and then aggregate them into a naive ensemble (with or without a fixed linear transformation) to distribute. On the other hand, Cho et al. [7], Zhang et al. [69], Fan et al. [14] apply personalized ensemble strategies by additionally learning the mutual trust between models. Makhija et al. [44] propose FedHeNN, which distills training information in the form of matching feature kernels instead of the predictions of local AI models on the public data. Both FedHeNN and DCL-NN utilize centered kernel alignment [8] to match feature kernels of local models, but DCL-NN uses the ensemble distillation for predictions as well. Thus, DCL-NN enables parties to learn from the entire input space.

Table 1: Comparative analysis of decentralized environments for (nearly) minimax optimality of representative collaborative learning algorithms with kernel regression. nFedAvg indicates the nonparametric version of FedAvg in [58]. Note that IED [48] achieves a weaker version of minimax optimality.

| Methods | interaction method | local data privacy | massively distributed | non-i.i.d.& unbalanced |
|---------|--------------------|--------------------|-----------------------|------------------------|
| DKRR [38] | divide-and-conquer | | | |
| DC-NY [66] | divide-and-conquer | ✓ | | |
| DKRR-CM [40] | model exchange | | | |
| DKRR-RF-CM [43] | model exchange | | | |
| DKRR-NY-CM [67] | model exchange | ✓ | | |
| nFedAvg [58] | model exchange | | ✓ | ✓ |
| IED* [48] | knowledge distillation | ✓ | ✓ | |
| DCL-KR (ours) | knowledge distillation | ✓ | ✓ | ✓ |

**Decentralized Learning with Kernel Regression** A number of studies have investigated the minimax optimal rate of regularized kernel regression algorithms such as kernel ridge regression and gradient descent-based kernel regression with early stopping [5, 15, 37, 65]. In particular, over the past decade, the growing interest in decentralized learning has led to active research in the generalization analysis of decentralized kernel regression. While divide-and-conquer algorithms [34, 38, 66, 70] play a significant role in this research flow, most of them fail to account for statistical heterogeneity and massively distributed cases, along with privacy preservation, which has received a lot of attention recently. On the other hand, decentralized kernel regression algorithms with multiple communication rounds [40, 43, 48, 58, 67] achieve superior theoretical results compared to the divide-and-conquer algorithms. However, the discussions of these algorithms primarily focus on the efficiency of resource costs [40, 43, 67], while research on relaxing environmental constraints has been scarce. For example, most of these works assume a limited number of parties to prove the optimality in a minimax sense.

To the best of our knowledge, [48, 58] stand as the only investigations that consider general decentralized environments. Similar to our work, Park et al. [48] study the convergence rate of distillation-based collaborative learning with kernel regression. However, their results demonstrate a weaker version of minimax optimality and do not cover statistically heterogeneous environments. In this regard, Su et al. [58] offer a promising methodology. They analyze nonparametric versions of FedAvg [45] and FedProx [31], representative FL algorithms involving model exchange, in general decentralized environments such as statistically heterogeneous and massively distributed scenarios. In this work, we extend their methodology to analyze FedMD [30, 41] from a nonparametric perspective in massively distributed statistically heterogeneous environments. We summarize the comparison between our work and prior studies in Table 1. Note that algorithms that do not employ Nyström scheme (including nonparametric FedAvg [58]) fail to preserve local data privacy due to the inherent characteristics of kernel regression. On the other hand, DC-NY [66] and DKRR-NY-CM [67] can achieve the local data privacy preservation by utilizing the public data as Nyström centers.

## 3 DCL-KR: A Nonparametric View of FedMD

In this section, we establish the theory of a nonparametric version of FedMD [30, 41], the most standard distillation-based collaborative learning algorithm.

### 3.1 Preliminaries

Let $\rho_{\mathbf{x},y} = \rho_{\mathbf{x}} \cdot \rho_{y|\mathbf{x}}$ be a Borel probability measure on $\mathcal{X} \times \mathbb{R}$ where $\mathcal{X}$ is a compact subset of $\mathbb{R}^d$ and we assume the support of $\rho_{\mathbf{x}}$ is $\mathcal{X}$. The goal of the regression problem is to find a minimizer of the population risk, i.e.,

$$\min_{h:\mathcal{X}\to\mathbb{R}} \mathcal{E}(h), \qquad \mathcal{E}(h) := \frac{1}{2} \mathbb{E}_{(\mathbf{x},y)\sim\rho_{\mathbf{x},y}} |y - h(\mathbf{x})|^2.$$

Then, the function $f_0^* : \mathcal{X} \to \mathbb{R}$ defined by $\mathbf{x}_0 \mapsto \mathbb{E}_{y\sim\rho_{y|\mathbf{x}}(\cdot|\mathbf{x}_0)}[y]$, $\mathbf{x}_0 \in \mathcal{X}$ is a target function.

Let $k : \mathcal{X} \times \mathcal{X} \to \mathbb{R}$ be a Mercer kernel [9] where $\kappa := (\sup_{\mathbf{x} \in \mathcal{X}} k(\mathbf{x}, \mathbf{x}))^{1/2} < \infty$ and $\mathbb{H}_k$ be a reproducing kernel Hilbert space associated to $k$. We set $k_{\mathbf{x}} := k(\cdot, \mathbf{x})$ and the covariance operator $T_{k,\nu} : \mathbb{H}_k \to \mathbb{H}_k$ with respect to any Borel probability measure $\nu$ on $\mathcal{X}$ defined as

$$T_{k,\nu} h = \int_{\mathcal{X}} h(\mathbf{x}) k_{\mathbf{x}} \, d\nu(\mathbf{x}).$$

Then we can see that $T_{k,\nu} = \iota_\nu^\top \iota_\nu$ where $\iota_\nu : \mathbb{H}_k \to L_\nu^2$ is a natural embedding, $L_\nu^2 = L^2(\mathcal{X}, \nu)$ denotes the $L^2$ space, and a superscript $\top$ denotes the adjoint operator of a given operator. We also define the sampling operator $S_D : \mathbb{H}_k \to \mathbb{R}^n$ by $h \mapsto [h(\mathbf{x}^1), \cdots, h(\mathbf{x}^n)]^\top$ and $T_{k,X} := S_D^\top S_D$ when $D = \{(\mathbf{x}^1, y^1), \cdots, (\mathbf{x}^n, y^n)\}$ with $X = \{\mathbf{x}^1, \cdots, \mathbf{x}^n\}$ is given. Since $S_D$ depends only on data inputs $X$, we can define the sampling operator for unlabeled datasets in the same way. See Appendix A.1 for further details.

### 3.1.1 Kernel Gradient Descent with Early Stopping

Given a dataset $D = \{(\mathbf{x}^1, y^1), \cdots, (\mathbf{x}^n, y^n)\}$ generated from $\rho_{\mathbf{x},y}$, consider the empirical risk $\widetilde{\mathcal{E}}_D : \mathbb{H}_k \to \mathbb{R}$ given by

$$\widetilde{\mathcal{E}}_D(h) = \frac{1}{2} \|S_D h - \mathbf{y}\|_2^2$$

where $\mathbf{y} = [y^1, \cdots, y^n]^\top$. Here, $\|\cdot\|_2$ denotes a scaled Euclidean norm $\|\mathbf{v}\|_2 = (\frac{1}{n} \sum_{i=1}^n \mathbf{v}_i^2)^{1/2}$. From the functional derivative $\nabla \widetilde{\mathcal{E}}_D(h) = S_D^\top(S_D h - \mathbf{y})$, the gradient descent scheme becomes

$$\nu_1 = 0, \quad \nu_{t+1} = \nu_t - \eta_t S_D^\top(S_D \nu_t - \mathbf{y}) \quad (t = 1, 2, \cdots)$$

where $\{\eta_t\}_{t \in \mathbb{N}}$ is a set of learning rates. In this work, we set $\eta_t = \eta$, $t \in \mathbb{N}$ for a fixed $\eta \in (0, 1/\kappa^2)$. Then, a simple calculation gives $\nu_t \to S_D^\top(S_D S_D^\top)^{-1} \mathbf{y}$ as $t \to \infty$ provided that the operator $S_D S_D^\top$ is invertible. The limit is known as the minimum norm interpolation [49] of $D$. Since the interpolation regressor generalizes poorly unless there is no noise [32, 39], early stopping strategies are usually applied to avoid the overfitting issue. With adequate stopping rules, gradient descent-based kernel regression has an optimal rate in a minimax sense [36, 37, 65].

## 3.2 DCL-KR Algorithm

From now on, we consider the setting that there are $m$ parties and the $i$th party has a private local data $D_i = \{(\mathbf{x}_i^j, y_i^j) : j = 1, \cdots, n_i\}$ for $i = 1, \cdots, m$. Assume that all data $D = \bigcup_{i=1}^m D_i$ are i.i.d. with the distribution $\rho_{\mathbf{x},y}$ but each local dataset does not need to have the same distribution. Let $Z = \{\mathbf{z}^1, \cdots, \mathbf{z}^{n_0}\} \subset \mathcal{X}$ be the additional public inputs. The goal of all parties is to have their models that perform well on the distribution $\rho_{\mathbf{x}}$. In other words, each party expects to be able to make good predictions not only for its local data distribution but also for unseen data distribution through collaborative learning.

Similar to [58], we construct a nonparametric version of FedMD (called **DCL-KR**), which is presented in Algorithm 1. In Algorithm 1, $\mathcal{G}_i$ is a one-step local gradient descent update on $\widetilde{\mathcal{E}}_{D_i}$, i.e., $\mathcal{G}_i h = h - \eta S_{D_i}^\top(S_{D_i} h - \mathbf{y}_i)$ where $\mathbf{y}_i = [y_i^1, \cdots, y_i^{n_i}]^\top$. Similarly, $\tilde{\mathcal{G}}_t$ is a one-step gradient descent update on $\widetilde{\mathcal{E}}_{(Z, \mathbf{y}_{p,t})}$, i.e., $\tilde{\mathcal{G}}_t h = h - \eta S_Z^\top(S_Z h - \mathbf{y}_{p,t})$.

## 3.3 Theoretical Results

In this subsection, we show the nearly minimax optimality of DCL-KR. To derive theoretical results, we assume the following conditions regarding regularity of noise, the kernel $k$, and the target function $f_0^*$ as below.

**Assumption 3.1.** We assume $\mathbb{E}_{y \sim \rho_y} y^2 < \infty$ and

$$\int \left( \exp\left( \frac{|y - f_0^*(\mathbf{x})|}{M} \right) - \frac{|y - f_0^*(\mathbf{x})|}{M} - 1 \right) d\rho_{y|\mathbf{x}}(y|\mathbf{x}) \leq \frac{\gamma^2}{2M^2}, \quad \forall \mathbf{x} \in \mathcal{X}$$

where $M$ and $\gamma$ are positive constants.

---

**Algorithm 1** DCL-KR Algorithm

---

1: **Hyperparameters:** $T$: total communication round, $E$: the number of local iterations at each communication round, $\eta$: learning rate
2: Initialize local models $f_{i,0} = 0$ for $i = 1, \cdots, m$.
3: **for** $t = 0, \cdots, T-1$ **do**
4:     **for** party $i = 1, \cdots, m$ **do**
5:         Update the local model $E$ times by gradient descent on the empirical risk $\widetilde{\mathcal{E}}_{D_i}$

$$f'_{i,t} \leftarrow \mathcal{G}_i^E f_{i,t}.$$

6:         Upload the local predictions on $Z$ to the server

$$\mathbf{y}_{p,t}^i = S_Z f'_{i,t}.$$

7:     **end for**
8:     The server aggregates the local predictions to compute the consensus prediction

$$\mathbf{y}_{p,t} = \sum_{i=1}^{m} \frac{n_i}{n} \mathbf{y}_{p,t}^i$$

    and then distributes $\mathbf{y}_{p,t}$ to all local parties.
9:     For party $i$ ($i = 1, \cdots, m$), update the local model by infinitely many iterations of gradient descent on the empirical risk $\widetilde{\mathcal{E}}_{(Z,\mathbf{y}_{p,t})}$

$$f_{i,t+1} \leftarrow \tilde{\mathcal{G}}_t^\infty g_{i,t} \tag{1}$$

    with an initialization $g_{i,t}$ chosen from a subspace spanned by $k_{\mathbf{z}^1}, \cdots, k_{\mathbf{z}^{n_0}}$.
10: **end for**

---

**Assumption 3.2.** Let $\lambda_1 \geq \lambda_2 \geq \cdots > 0$ be eigenvalues of $T_{k,\rho_\mathbf{x}}$. There are fixed positive constants $C_s$ and $c_s$ such that

$$c_s i^{-1/s} \leq \lambda_i \leq C_s i^{-1/s}, \ \forall i \in \mathbb{N}$$

for some $s \in (0, 1)$.

**Assumption 3.3.** The target function $f_0^*$ satisfies

$$f_0^* \in \left\{ h \in \mathbb{H}_k : h = T_{k,\rho_\mathbf{x}}^{r-1/2} g \ \text{where} \ \|g\|_{\mathbb{H}_k} \leq R \right\}$$

for some $r \in [\frac{1}{2}, 1]$ where $T_{k,\rho_\mathbf{x}}^{r-1/2}$ is the $(r - 1/2)$ power of operator $T_{k,\rho_\mathbf{x}}$ and $R > 0$ is a fixed constant. In particular, $f_0^* \in \mathbb{H}_k$.

The above assumptions determine the minimax lower rate [5] and are standard assumptions in many prior works [5, 15, 33, 35]. In detail,

- Assumption 3.1 implies that the noise is not excessively large. This assumption is a general noise condition that encompasses a wide range of cases. For instance, noise with Bernstein condition such as sub-Gaussian noise satisfies Assumption 3.1.

- Assumption 3.2 is about the eigenvalue decay of $T_{k,\rho_\mathbf{x}}$. From this assumption, one can derive bounds on the effective dimension that is related to covering and entropy number conditions [15].

- Assumption 3.3 is related to the regularity of the target function, specifically how well the RKHS induced by the kernel $k$ represents the target function.

Under these assumptions, we can theoretically show the performance guarantee of DCL-KR. The proof is provided in Appendix A.2. Note that $\mathcal{E}(h) - \mathcal{E}(f_0^*) = \frac{1}{2} \|\iota_{\rho_\mathbf{x}}(h - f_0^*)\|_{L_{\rho_\mathbf{x}}^2}^2$ is the excess risk of a regressor $h$ and so the quantity $\|\iota_{\rho_\mathbf{x}}(h - f_0^*)\|_{L_{\rho_\mathbf{x}}^2}$ indicates the generalization ability of $h$.

**Theorem 3.4.** *Under Assumption 3.1, 3.2, and 3.3, with $n_0 \geq n^{\frac{1}{2r+s}}(\log n)^3$ public inputs independently generated from $\tilde{\rho}_{\mathbf{x}}$ such that the Radon-Nikodym derivative $\frac{d\rho_{\mathbf{x}}}{d\tilde{\rho}_{\mathbf{x}}}$ satisfies*

$$0 \leq \frac{d\rho_{\mathbf{x}}}{d\tilde{\rho}_{\mathbf{x}}} \leq B \text{ on } \mathcal{X} \text{ for some } B \in [1, \infty), \tag{2}$$

*DCL-KR gives the performance guarantee*

$$\mathbb{E}\|\iota_{\rho_{\mathbf{x}}}(f_{i,T} - f_0^*)\|_{L^2_{\rho_{\mathbf{x}}}} \leq C \cdot B^r n^{-\frac{r}{2r+s}} \log n$$

*for all $i = 1, \cdots, m$ where $\eta \in (0, 1/\kappa^2)$ is a fixed learning rate, $T$ is an adequate stopping rule, and the prefactor $C$ does not depend on $B$, $m$, and $n$.*

Since the convergence rate $n^{-\frac{r}{2r+s}}$ is the minimax lower rate under Assumption 3.1, 3.2, and 3.3, Theorem 3.4 implies that DCL-KR has an almost same convergence rate as the minimax optimal central training when there are sufficiently many public inputs. To the best of our knowledge, this is the first work to prove the (nearly) minimax optimality of a collaborative learning algorithm that does not directly share local data or models in massively distributed statistically heterogeneous environments. For example, divide-and-conquer algorithms work for limited $m$. Specifically, DC-NY [66] assumes $m \leq O(n^{\frac{2r-1}{2r+s}})$ and DKRR-NY-CM [67] assumes $m \leq O(n^{\frac{2r+s-1}{2r+s}})$. However, Theorem 3.4 does not require any condition on $m$. Moreover, Theorem 3.4 deals with a more general setting than the theory in [48, 58]. For example, Su et al. [58] only cover $r = \frac{1}{2}$ of Assumption 3.3. On the other hand, Park et al. [48] do not consider Assumption 3.2 which gives a finer result. Compared with [48], we also reduce the required size of public inputs and drop the statistical homogeneity condition.

The convergence rate in Theorem 3.4 has an additional factor $\log n$ compared with a minimax lower rate [5, 15], but this logarithm term grows slower than any polynomial. Note that an additional logarithm term commonly appears in the context of gradient descent-based kernel regression with Nyström scheme [35, 36].

Theorem 3.4 allows that the public input distribution $\tilde{\rho}_{\mathbf{x}}$ can be different from the local input distribution $\rho_{\mathbf{x}}$. It is natural that the condition (2) is required since $\tilde{\rho}_{\mathbf{x}}$ should cover $\rho_{\mathbf{x}}$ for fully distilling training information. We can see that the discrepancy between $\rho_{\mathbf{x}}$ and $\tilde{\rho}_{\mathbf{x}}$ affects the upper bound in Theorem 3.4 as the multiplication of $B^r$. We can remove $B^r$ in the upper bound by increasing public inputs. See Appendix A.3 for details.

### 3.3.1 Proof Sketch of Theorem 3.4 and Comments

In the proof of Theorem 3.4, we decompose the term $\iota_{\rho_{\mathbf{x}}}(f_{i,T} - f_0^*)$ into four parts, say (I), (II), (III), and (IV) (see Eq. (7)). The proof is to bound the norms of these terms. Note that DCL-KR can also be understood as a Nyström version of nonparametric FedAvg [58] from the recurrence relation (6).

(I) and (II) appear similarly in [58], except that (I) and (II) incorporate projections. To handle these terms, we reinterpret the proof presented in [58] in operator form instead of matrix form and extend it to our setting. We obtain a norm bound of (II) containing a quantity linked to the local Rademacher complexity. (Appendix A.2.2 and A.2.3)

Comparing with [58], (III) and (IV) are additional terms induced by the procedure that distills functional information from the local regressors. We apply techniques used in [35, 48, 53] to bound (III) and (IV). (Appendix A.2.4)

Note that previous works applying local Rademacher complexity-based stopping rule [50, 58] deal with the case of $r = \frac{1}{2}$ only. In this work, we set a new stopping rule $T$ which is an extension of previous works [50, 58] and prove an extended version (Lemma A.6) of a well-known property [60]. As a result, our theory covers $r \in [\frac{1}{2}, 1]$ which affects the minimax lower rate. (Appendix A.2.5)

## 4 DCL-NN Algorithm

In this section, we retain the problem setting from Section 3 but employ heterogeneous neural networks as the local models. Based on the theoretical results in Section 3, we propose a novel

distillation-based collaborative learning algorithm **DCL-NN** across heterogeneous neural networks in a decentralized setting.

A key factor contributing to the successful theoretical guarantee of DCL-KR lies not only in the linearity of kernel regression but also in the equality of kernels across local models. In fact, the public data predictions can vary in different directions, even if the same training data points are used when kernels differ (See Appendix B). Therefore, we match the kernels of local AI models. Specifically, we use linear feature kernels [18, 64] induced by the features from the last hidden layers of local AI models for kernel matching. For example, for a neural network $f : \mathcal{X} \to \mathbb{R}$ where $f(\cdot) = \mathbf{w}^\top g(\cdot) + b$, $g : \mathcal{X} \to \mathbb{R}^c$, $\mathbf{w} \in \mathbb{R}^c$, and $b \in \mathbb{R}$ we use

$$k_f(\mathbf{x}^1, \mathbf{x}^2) = g(\mathbf{x}^1)^\top g(\mathbf{x}^2), \quad \mathbf{x}^1, \mathbf{x}^2 \in \mathcal{X} \tag{3}$$

as the feature kernel of $f$. Through this idea, we can bring the setting closer to the regime of DCL-KR. Note that our theoretical results suggest that the target kernel should be a good kernel. Indeed, we observe that the naive ensemble

$$k = \sum_{i=1}^{m} \frac{n_i}{n} k_{f_i}. \tag{4}$$

has a significantly better performance than individual feature kernels $k_{f_1}, \cdots, k_{f_m}$ (See Section 5 and Appendix B). Here, $f_i$ is the local model of the $i$th party with its local feature kernel $k_{f_i}$ obtained by (3) ($i = 1, \cdots, m$). Therefore, we align local feature kernels $k_{f_1}, \cdots, k_{f_m}$ in a kernel distillation manner with the ensemble kernel $k$ obtained by (4).

For this purpose, we introduce Centered Kernel Alignment (CKA) [8] as a kernel similarity measure. CKA is a typical measure associated with the similarity of two representations of neural networks [27] and is often used for kernel matching in neural networks [44]. To compute empirical CKA between two kernels $k_1$ and $k_2$ on inputs $\{\mathbf{c}^1, \cdots, \mathbf{c}^p\}$, we first calculate the Gram matrices $K_1 = [k_1(\mathbf{c}^{j_1}, \mathbf{c}^{j_2})]_{1 \leq j_1, j_2 \leq p}$ and $K_2 = [k_2(\mathbf{c}^{j_1}, \mathbf{c}^{j_2})]_{1 \leq j_1, j_2 \leq p}$. We then compute the empirical CKA via

$$\widehat{\mathrm{CKA}}(k_1, k_2) = \frac{\widehat{\mathrm{HSIC}}(K_1, K_2)}{\sqrt{\widehat{\mathrm{HSIC}}(K_1, K_1)\widehat{\mathrm{HSIC}}(K_2, K_2)}}.$$

Here, $\widehat{\mathrm{HSIC}}$ is an estimator of the Hilbert-Schmidt Independence Criterion (HSIC) defined as

$$\widehat{\mathrm{HSIC}}(K_1, K_2) = \frac{1}{(p-1)^2} \mathrm{tr}(K_1 H K_2 H)$$

where $H := I_p - \frac{1}{p} \mathbf{1}\mathbf{1}^\top$ is the centering matrix. In the kernel distillation procedure, the $i$th local party maximizes $\widehat{\mathrm{CKA}}(k_{f_i}, k)$ on public inputs $Z$ ($i = 1, \cdots, m$). Notably, this procedure requires only a single communication round for exchanging pairwise feature kernel values on public inputs, ensuring that our algorithm operates exclusively within the function space.

After the kernel distillation procedure, all local AI models have similar feature kernels up to constants. So we can follow an analogous process as in DCL-KR. Note that we perform learning rate scaling described in Appendix B to compensate the kernel scale difference. It makes the impact of local iterations consistent. We also provide the complete algorithm (Algorithm 2) and further details for Section 4 in Appendix B.

## 5 Experiments

In this section, we evaluate the performance of DCL-KR and DCL-NN. We compare them with baselines on various regression tasks.

**Datasets**   We use the following six regression datasets to evaluate the performance. Target variables are one-dimensional in all datasets. (1) **Toy-1D** [33] and (2) **Toy-3D** [6] are synthetic datasets with one-dimensional and three-dimensional inputs, respectively. (3) **Energy** is a tabular dataset from the UCI database [12] to predict appliances energy use with 28 features. (4) **RotatedMNIST** is an image dataset where it aims to predict the rotation angles for given rotated images of the MNIST [11] images. (5) **UTKFace** [71] and (6) **IMDB-WIKI** [42, 52] are image datasets for age estimation.

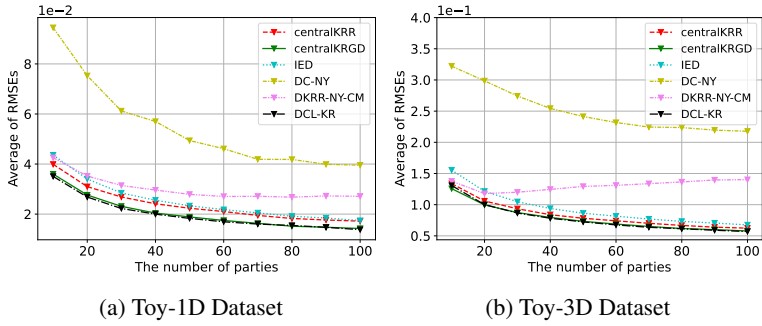

(a) Toy-1D Dataset

(b) Toy-3D Dataset

Figure 1: Performance of central Kernel Ridge Regression (centralKRR), central Kernel Regression with Gradient Descent (centralKRGD), DC-NY, DKRR-NY-CM, IED, and DCL-KR on Toy-1D and Toy-3D

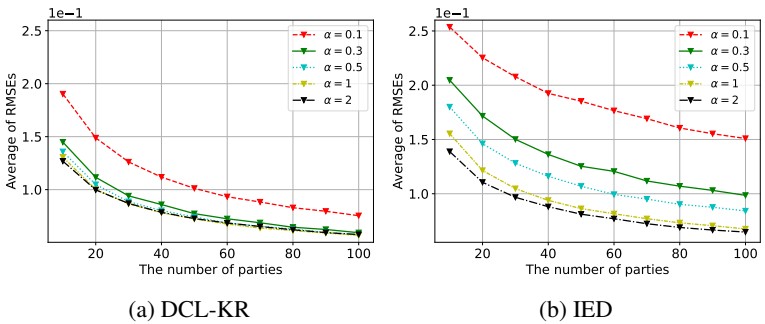

(a) DCL-KR

(b) IED

Figure 2: Performance of IED and DCL-KR with $n_0 \approx \alpha \cdot n^{\frac{1}{2r+s}} (\log_{10} n)^3$ on Toy-3D

We compare kernel machine-based collaborative learning algorithms on two datasets Toy-1D and Toy-3D. On the other hand, we compare neural network-based collaborative learning algorithms on five datasets Toy-3D, Energy, RotatedMNIST, UTKFace, and IMDB-WIKI.

**Baselines** We compare DCL-KR with two central kernel regression models to verify our theoretical results. These two central models have the minimax optimal convergence rate. We also utilize existing decentralized kernel regression algorithms that does not directly share local data and models (DC-NY [66], DKRR-NY-CM [67], IED [48]) as baselines for DCL-KR. On the other hand, we adopt FedMD with unlabeled public inputs [30, 41], FedHeNN [44], and KT-pFL [69] as baselines for DCL-NN.

**Setup** The number of parties ranges from 10 to 100 for kernel machine-based algorithms and is 50 for neural network-based algorithms. We construct statistically heterogeneous decentralized environments with Algorithm 3. For neural network-based algorithms, we use 4 different neural network architectures for local models in all settings. For instance, we use ResNet-18, ResNet-34, ResNet-50 [20], and MobileNetv2 [54] for large-scale image datasets. We utilize the average of Root Mean Squared Errors (RMSEs) of the local AI models on a test dataset as a performance metric. The test data points have the same distribution as the whole local data distribution. We apply FedMD with a few communication rounds for pretraining of DCL-NN. See Appendix C for detailed experimental configurations.

### 5.1 Results on Kernel Machine-based Algorithms

The performance of DCL-KR and its baselines is presented in Figure 1. We set the number of parties $m = 10, 20, \cdots, 100$, the number of private data points $n = 50m$, and the number of public inputs $n_0 = n^{\frac{1}{2r+s}} (\log_{10} n)^3$. We first set $\rho_{\mathbf{x}} = \tilde{\rho}_{\mathbf{x}}$, i.e., the public data distribution is the same as the entire local input distribution. As shown in Figure 1, DCL-KR outperforms the baselines in all

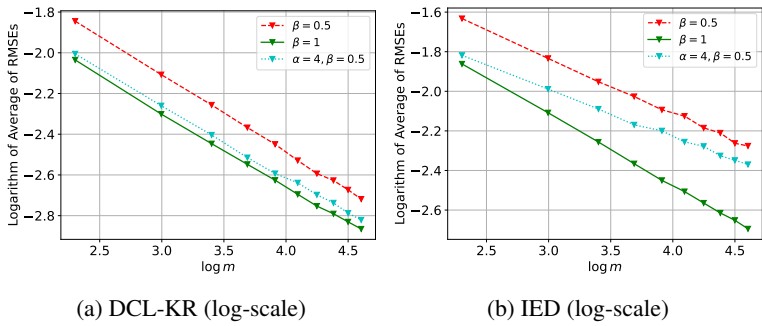

(a) DCL-KR (log-scale)                    (b) IED (log-scale)

Figure 3: Performance of IED and DCL-KR with $\tilde{\rho}_{\mathbf{x}} \neq \rho_{\mathbf{x}}$ on Toy-3D

experimental settings and achieves comparable performance to the central models. This result implies that DCL-KR has not only the nearly optimal convergence rate but also the same performance as central kernel regression models. In contrast, DC-NY and DKRR-NY-CM exhibit significantly lower performance compared with DCL-KR in massively distributed environments where their theory does not cover. IED does not show a significant performance drop in massively distributed environments even though its theory is built on the statistical homogeneity condition of local data distributions.

To further compare the performance of DCL-KR and IED, which show similar results to central models, we analyze the effect of $n_0$ and $\tilde{\rho}_{\mathbf{x}}$ on their performance (Figure 2 and 3). Figure 2 illustrates that, as expected from the theoretical results, IED requires more public inputs than DCL-KR to achieve good performance. Moreover, when there is a public distribution shift, DCL-KR maintains its convergence rate, whereas the convergence rate of IED deteriorates. (See Appendix C.3.3 for experimental details.) Overall, our experiments validate the theoretical results of DCL-KR and demonstsrate its superiority over previous results. For additional experimental results and analyses, please refer to Appendix C.3.

## 5.2 Results on Neural Network-based Algorithms

Table 2 shows the performance of DCL-NN and baselines on five regression tasks. We also present the performance of standalone models and centralized models to assess the performance of the collaborative algorithms. For some cases exhibiting training instability, we report the best test error (marked with asterisks) observed across all communication rounds, while relying on a fixed number of communication rounds for the other cases.

As can be seen in Table 2, DCL-NN outperforms the baselines on all regression tasks. Note that FedHeNN employs kernel matching similar to DCL-NN, but it lacks supervision of label prediction through collaboration, resulting in insufficient performance improvement compared to standalone models. Given the superior performance of DCL-NN, it is evident that incorporating supervised learning for label prediction alongside kernel matching is desirable. On the other hand, while FedMD performs significantly better than standalone models, the performance of DCL-NN is consistently better. Considering that we utilize FedMD for pretraining of DCL-NN, we can see that it performs better than FedMD-only collaborative learning by first training local models with FedMD and then using DCL-NN. In conclusion, the experimental results support the practical effectiveness and superiority of DCL-NN over baselines.

**Kernel Distillation Procedure**    To verify the necessity of kernel distillation, we examine the changes in the performance of local feature kernels and the CKA between them during the kernel distillation procedure. We conduct this experiment on UTKFace. We utilize the RMSE of a kernel linear regression model trained on all local data as a kernel performance measure. The results are presented in Figure 4. As shown in Figure 4, both kernel performance and CKA undergo a temporary degradation due to the change of the objective function at the initial stages. However, as training progresses, both metrics recover and kernel performance surpasses its initial level. Since kernel distillation aims to ensure that all local feature kernels are similar with high performance, the experimental results verify the effectiveness of kernel distillation.

Table 2: Performance comparison of FedMD, FedHeNN, KT-pFL, and DCL-NN on five datasets. The values are presented as the average of RMSEs along with standard deviations. For calibration, the performance of standalone models and centralized models is also provided.

| | Toy-3D | Energy | RotatedMNIST | UTKFace | IMDB-WIKI |
|---|---|---|---|---|---|
| Central | 0.041 | 0.085 | 0.139 | 0.143 | 0.095 |
| Standalone | $0.288 \pm 0.008$ | $0.095 \pm 0.000$ | $0.680 \pm 0.003$ | $0.216 \pm 0.004$ | $0.137 \pm 0.000$ |
| FedMD | $0.200 \pm 0.008$ | $0.093 \pm 0.000$ | $0.249 \pm 0.001$ | $0.151 \pm 0.004$ | $0.113 \pm 0.000$ |
| FedHeNN | $0.264^* \pm 0.009$ | $0.094^* \pm 0.000$ | $0.405 \pm 0.016$ | $0.177 \pm 0.000$ | $0.140^* \pm 0.000$ |
| KT-pFL | $0.243 \pm 0.002$ | $0.093^* \pm 0.000$ | $0.317 \pm 0.003$ | $0.167 \pm 0.001$ | $0.130^* \pm 0.002$ |
| **DCL-NN** | $\mathbf{0.079 \pm 0.005}$ | $\mathbf{0.087 \pm 0.001}$ | $\mathbf{0.227 \pm 0.003}$ | $\mathbf{0.148 \pm 0.001}$ | $\mathbf{0.110 \pm 0.000}$ |

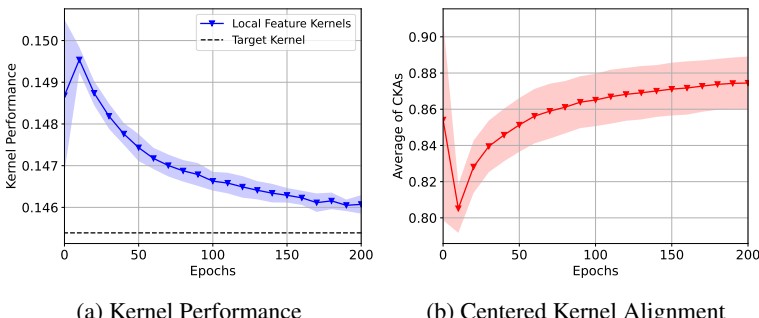

(a) Kernel Performance      (b) Centered Kernel Alignment

Figure 4: Kernel performance and CKA (with standard deviations) during the kernel distillation procedure. The performance of the target kernel obtained by (4) is also provided.

For additional experimental results, please refer to Appendix C.4.

# 6 Conclusions

In this work, we analyze distillation-based collaborative learning from a nonparametric perspective and propose DCL-NN, a practical algorithm as an extension. We demonstrate that DCL-KR, a nonparametric version of FedMD, has a nearly minimax optimal convergence rate in massively distributed statistically heterogeneous environments. Inspired by DCL-KR, we propose DCL-NN, a novel distillation-based collaborative learning algorithm for heterogeneous neural networks. Our experiments confirm the theoretical results of DCL-KR and demonstrate the practical effectiveness of DCL-NN. For a discussion of the limitations of our work, please refer to Appendix D.

**Broader Impact**   Our work explores the methodologies of collaborative learning under data and model privacy preservation. In this regard, our research holds the potential to positively impact the facilitation of collaboration among AI models without raising concerns about information disclosure. On the other hand, our work does not pose any particularly noteworthy negative consequences, given its aim to contribute to the advancement of the general field of machine learning.

# Acknowledgments and Disclosure of Funding

This work was supported by the National Research Foundation of Korea(NRF) grant funded by the Korea government(MSIT) (Grant No. RS-2019-NR040050).

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

Table 3: List of some notations

| Notation | Meaning |
|---|---|
| $a \wedge b$ | minimum of $a$ and $b$ |
| $a \vee b$ | maximum of $a$ and $b$ |
| $\mathbb{R}^d$ | $d$-dimensional Euclidean space |
| $\mathcal{X}$ | the input space contained in $\mathbb{R}^d$ |
| $C(\mathcal{X})$ | the collection of all continuous functions from $\mathcal{X}$ into $\mathbb{R}$ |
| $\rho_{\mathbf{x},y}$ | the data generating distribution on $\mathcal{X} \times \mathbb{R}$ |
| $\rho_{\mathbf{x}}, \rho_y$ | the marginal distribution of $\rho_{\mathbf{x},y}$ on $\mathcal{X}$ and $\mathbb{R}$, respectively |
| $\rho_{y\|\mathbf{x}}(\cdot\|\mathbf{x}_0)$ | the conditional distribution on $\mathbb{R}$ w.r.t. $\mathbf{x}_0 \in \mathcal{X}$ and $\rho_{\mathbf{x},y}$ |
| $\tilde{\rho}_{\mathbf{x}}$ | the public input distribution on $\mathcal{X}$ |
| $k$ | a given Mercer kernel |
| $k_{\mathbf{x}}$ | $k(\cdot, \mathbf{x})$ |
| $\kappa$ | $(\sup_{\mathbf{x} \in \mathcal{X}} k(\mathbf{x}, \mathbf{x}))^{1/2}$ |
| $L_\nu^2$ | the $L^2$ space on $\mathcal{X}$ w.r.t. measure $\nu$ |
| $\mathbb{H}_k$ | a reproducing kernel Hilbert space associated to kernel $k$ |
| $T_{k,\nu}$ | the covariance operator on $\mathbb{H}_k$ w.r.t. measure $\nu$, $T_{k,\nu} : h \mapsto \int_{\mathcal{X}} h(\mathbf{x}) k_{\mathbf{x}} \, d\nu(\mathbf{x})$ |
| $\iota_\nu$ | a natural embedding from $\mathbb{H}_k$ into $L_\nu^2$ |
| $S_D$ | a sampling operator from $\mathbb{H}_k$ into $\mathbb{R}^n$, $S_D : h \mapsto [h(\mathbf{x}^1), \cdots, h(\mathbf{x}^n)]^\top$ where $D = \{(\mathbf{x}^i, y^i)\}_{i=1}^n$ |
| $S_Z$ | a sampling operator from $\mathbb{H}_k$ into $\mathbb{R}^n$, $S_Z : h \mapsto [h(\mathbf{z}^1), \cdots, h(\mathbf{z}^n)]^\top$ where $Z = \{\mathbf{z}^i\}_{i=1}^n$ |
| $T_{k,X}$ | $S_X^\top S_X$ |
| $\mathcal{E}(h)$ | the population risk of $h$, $\mathcal{E}(h) = \frac{1}{2}\mathbb{E}_{(\mathbf{x},y)\sim\rho_{\mathbf{x},y}}\|y - h(\mathbf{x})\|^2$ |
| $\widetilde{\mathcal{E}}_D(h)$ | the empirical risk of $h$ over $D = \{(\mathbf{x}^i, y^i)\}_{i=1}^n$, $\widetilde{\mathcal{E}}_D(h) = \frac{1}{2}\|S_D h - \mathbf{y}\|_2^2$ |
| $f_0^*$ | a target function from $\mathcal{X}$ into $\mathbb{R}$ defined as $f_0^*(x_0) = \mathbb{E}_{y\sim\rho_{y\|\mathbf{x}}(\cdot\|\mathbf{x}_0)}[y]$, $\mathbf{x}_0 \in \mathcal{X}$ |
| $\eta$ | a learning rate, $\eta \in (0, 1/\kappa^2)$ |
| $m$ | the number of parties |
| $D_i$ | private local data of the $i$th party, $\{(\mathbf{x}_i^j, y_i^j) : j = 1, \cdots, n_i\}$ $(i = 1, \cdots, m)$ |
| $X_i$ | inputs of $D_i$, $\{\mathbf{x}_i^j : j = 1, \cdots, n_i\}$ $(i = 1, \cdots, m)$ |
| $\mathbf{y}_i$ | labels of $D_i$, $[y_i^1, \cdots, y_i^{n_i}]^\top$ $(i = 1, \cdots, m)$ |
| $D$ | $D = \bigcup_{i=1}^m D_i$ |
| $Z$ | unlabeled public data, $\{\mathbf{z}^1, \cdots, \mathbf{z}^{n_0}\}$ |
| $M, \gamma$ | the parameters related to the regularity of noise (Assumption 3.1) |
| $\{(\lambda_i, \phi_i)\}_{i=1}^\infty$ | eigenvalues and eigenvectors of $T_{k,\rho_{\mathbf{x}}}$ such that $\lambda_1 \geq \lambda_2 \geq \cdots > 0$ from Mercer's representation (5). |
| $s, C_s, c_s$ | the parameters related to the eigenvalue decay of $T_{k,\rho_{\mathbf{x}}}$ (Assumption 3.2) |
| $C_s'$ | $C_s^s/(1-s)$ |
| $\mathcal{N}_\nu(\lambda)$ | $\text{tr}(T_{k,\nu}(T_{k,\nu} + \lambda I)^{-1})$ where $\nu$ is a probability measure |
| $\mathcal{N}(\lambda)$ | $\mathcal{N}_{\rho_{\mathbf{x}}}(\lambda)$ |
| $r, R$ | the parameters related to the regularity of $f_0^*$ (Assumption 3.3) |
| $B$ | the uniform bound of the Radon-Nikodym derivative $\frac{d\rho_{\mathbf{x}}}{d\tilde{\rho}_{\mathbf{x}}}$ in (2) |
| $E$ | the number of local iterations at each communication round in DCL-KR (Algorithm 1) |
| $T$ | total communication round in DCL-KR (Algorithm 1) |

## A Details on Section 3

Before we start the proof of Theorem 3.4, we present basic notions.

### A.1 Basic Notions

In Subsection 3.1, the reproducing kernel Hilbert space $\mathbb{H}_k$ is a subset of $C(\mathcal{X})$, i.e., all elements in $\mathbb{H}_k$ are continuous [57]. Since

$$\iota_{\rho_{\mathbf{x}}}^\top h(\cdot) = \langle \iota_{\rho_{\mathbf{x}}}^\top h, k. \rangle_{\mathbb{H}_k} = \langle h, \iota_{\rho_{\mathbf{x}}} k. \rangle_{L_{\rho_{\mathbf{x}}}^2} = \int_{\mathcal{X}} h(\mathbf{x}) k(\cdot, \mathbf{x}) \, d\rho_{\mathbf{x}}(\mathbf{x}),$$

we have $T_{k,\rho_{\mathbf{x}}} = \iota_{\rho_{\mathbf{x}}}^{\top} \iota_{\rho_{\mathbf{x}}}$. The compactness of $\iota_{\rho_{\mathbf{x}}}^{\top}$ [57] gives the fact that $T_{k,\rho_{\mathbf{x}}}$ is compact, self-adjoint, and positive. Furthermore, Mercer's theorem [62] gives a Mercer representation

$$k(\mathbf{x}^1, \mathbf{x}^2) = \sum_{i=1}^{\infty} \lambda_i \phi_i(\mathbf{x}^1)\phi_i(\mathbf{x}^2). \tag{5}$$

The fact that $\mathbb{H}_k \subset C(\mathcal{X})$ and $T_{k,\rho_{\mathbf{x}}} = \iota_{\rho_{\mathbf{x}}}^{\top} \iota_{\rho_{\mathbf{x}}}$ implies the injectivity of $T_{k,\rho_{\mathbf{x}}}$ and so $\lambda_i \neq 0$ for all $i \in \mathbb{N}$. We define

$$\mathcal{N}_\nu(\lambda) := \mathrm{tr}(T_{k,\nu}(T_{k,\nu} + \lambda I)^{-1})$$

for any probability measure $\nu$. For convenience, $\mathcal{N}(\lambda) := \mathcal{N}_{\rho_{\mathbf{x}}}(\lambda)$. From [5, 15], we have $\mathcal{N}(\lambda) \leq C_s' \lambda^{-s}$ where $C_s' := C_s^s/(1-s)$ and $\mathcal{N}_\nu(\lambda) \leq \kappa^2 \lambda^{-1}$. Given a dataset $D = \{(\mathbf{x}^i, y^i)\}_{i=1}^{n}$, a similar argument as above gives

$$S_D^{\top} : \mathbf{c} = [\mathbf{c}_1, \cdots, \mathbf{c}_n] \mapsto \frac{1}{n} \sum_{i=1}^{n} \mathbf{c}_i k_{\mathbf{x}^i}$$

and $T_{k,X} : h \mapsto \frac{1}{n} \sum_{i=1}^{n} h(\mathbf{x}^i) k_{\mathbf{x}^i}$.

Note that Assumption 3.1 implies that

$$\mathcal{E}(f_0^*) = \frac{1}{2}\mathbb{E}_{(\mathbf{x},y)\sim\rho_{\mathbf{x},y}}|y - f_0^*(\mathbf{x})|^2 \leq \frac{\gamma^2}{2} < \infty.$$

We have

$$\mathbb{E}_{(\mathbf{x},y)\sim\rho_{\mathbf{x},y}}|y - h(\mathbf{x})|^2 = \mathbb{E}_{\mathbf{x}\sim\rho_{\mathbf{x}}}|h(\mathbf{x}) - f_0^*(\mathbf{x})|^2 + \mathbb{E}_{(\mathbf{x},y)\sim\rho_{\mathbf{x},y}}|y - f_0^*(\mathbf{x})|^2$$

and so the excess risk becomes

$$\mathcal{E}(h) - \mathcal{E}(f_0^*) = \frac{1}{2}\mathbb{E}_{\mathbf{x}\sim\rho_{\mathbf{x}}}|h(\mathbf{x}) - f_0^*(\mathbf{x})|^2 = \frac{1}{2}\|\iota_{\rho_{\mathbf{x}}}(h - f_0^*)\|_{L_{\rho_{\mathbf{x}}}^2}^2.$$

Therefore, $\|\iota_{\rho_{\mathbf{x}}}(h - f_0^*)\|_{L_{\rho_{\mathbf{x}}}^2}^2$ indicates the generalization ability of $h$.

Table 3 presents meaning of some notations.

## A.2 Proof of Theorem 3.4

Without loss of generality, we assume $n \wedge n_0 \geq \kappa^2 e$.

### A.2.1 Recurrence Relation of DCL-KR

Consider a subspace $W$ of $\mathbb{H}_k$ spanned by $\{k_{\mathbf{z}^1}, \cdots, k_{\mathbf{z}^{n_0}}\}$. We first show that for a fixed $h^* \in \mathbb{H}_k$ and a gradient update $\mathcal{G}u = u - \eta S_Z^{\top}(S_Z u - S_Z h^*)$ we have $\mathcal{G}^t u_1 \to P_Z h^*$ as $t \to \infty$ for any $u_1 \in W$ where $P_Z$ is an orthogonal projection onto the subspace $W$. Set $u_{t+1} = \mathcal{G}u_t$ for $t \geq 1$. Then

$$u_{t+1} = (I - \eta S_Z^{\top} S_Z)u_t + \eta S_Z^{\top} S_Z h^* = (I - \eta S_Z^{\top} S_Z)^t u_1 + \sum_{k=0}^{t-1}(I - \eta S_Z^{\top} S_Z)^k \eta S_Z^{\top} S_Z h^*.$$

Since $S_Z h^* = S_Z P_Z h^*$, we have

$$\sum_{k=0}^{t-1}(I - \eta S_Z^{\top} S_Z)^k \eta S_Z^{\top} S_Z h^* = \sum_{k=0}^{t-1}(I - \eta S_Z^{\top} S_Z)^k \eta S_Z^{\top} S_Z P_Z h^* = P_Z h^* - (I - \eta S_Z^{\top} S_Z)^t P_Z h^*.$$

Note that there exists $\{\tilde{\mathbf{z}}^1, \cdots, \tilde{\mathbf{z}}^{\tilde{n}_0}\} \subset Z$ such that $\{k_{\tilde{\mathbf{z}}^1}, \cdots, k_{\tilde{\mathbf{z}}^{\tilde{n}_0}}\}$ is a basis of $W$. Define a matrix

$$B = \begin{bmatrix} b_{11} & \cdots & b_{1\tilde{n}_0} \\ \vdots & \ddots & \vdots \\ b_{n_01} & \cdots & b_{n_0\tilde{n}_0} \end{bmatrix} \in \mathbb{R}^{n_0 \times \tilde{n}_0}$$

such that $k_{\mathbf{z}^i} = \sum_{j=1}^{\tilde{n}_0} b_{ij} k_{\tilde{\mathbf{z}}^j}$. Then $K_{Z\tilde{Z}} = BK_{\tilde{Z}\tilde{Z}}$ where

$$K_{Z\tilde{Z}} = \begin{bmatrix} k(\mathbf{z}^1, \tilde{\mathbf{z}}^1) & \cdots & k(\mathbf{z}^1, \tilde{\mathbf{z}}^{\tilde{n}_0}) \\ \vdots & \ddots & \vdots \\ k(\mathbf{z}^{n_0}, \tilde{\mathbf{z}}^1) & \cdots & k(\mathbf{z}^{n_0}, \tilde{\mathbf{z}}^{\tilde{n}_0}) \end{bmatrix} \in \mathbb{R}^{n_0 \times \tilde{n}_0}$$

and

$$K_{\tilde{Z}\tilde{Z}} = \begin{bmatrix} k(\tilde{\mathbf{z}}^1, \tilde{\mathbf{z}}^1) & \cdots & k(\tilde{\mathbf{z}}^1, \tilde{\mathbf{z}}^{\tilde{n}_0}) \\ \vdots & \ddots & \vdots \\ k(\tilde{\mathbf{z}}^{\tilde{n}_0}, \tilde{\mathbf{z}}^1) & \cdots & k(\tilde{\mathbf{z}}^{\tilde{n}_0}, \tilde{\mathbf{z}}^{\tilde{n}_0}) \end{bmatrix} \in \mathbb{R}^{\tilde{n}_0 \times \tilde{n}_0}.$$

Set $P_Z h^* = \sum_{j=1}^{\tilde{n}_0} a_j k_{\tilde{\mathbf{z}}_j}$. Then we can see that

$$(I - \eta S_Z^\top S_Z)\left(\sum_{j=1}^{\tilde{n}_0} a_j k_{\tilde{\mathbf{z}}_j}\right) = \sum_{r=1}^{\tilde{n}_0}\left(a_r - \frac{\eta}{n}\sum_{i=1}^{n_0}\sum_{j=1}^{\tilde{n}_0} a_j k(\tilde{\mathbf{z}}_j, \mathbf{z}_i) b_{ir}\right) k_{\tilde{\mathbf{z}}_r}$$

$$= \sum_{r=1}^{\tilde{n}_0}\left(a_r - \frac{\eta}{n}[B^\top B K_{\tilde{Z}\tilde{Z}} \mathbf{a}]_r\right) k_{\tilde{\mathbf{z}}_r} = \sum_{r=1}^{\tilde{n}_0}\left[\left(I - \frac{\eta}{n} B^\top B K_{\tilde{Z}\tilde{Z}}\right)\mathbf{a}\right]_r k_{\tilde{\mathbf{z}}_r}$$

where $[\cdot]_r$ is the $r$th component of the given vector and $\mathbf{a} = [a_1, \cdots, a_{\tilde{n}_0}]^\top$. Note that $K_{\tilde{Z}\tilde{Z}}$ is invertible since $\mathbf{v}^\top K_{\tilde{Z}\tilde{Z}} \mathbf{v} = 0$ implies $\mathbf{v} = 0$. We can also see that $K_{ZZ} = B K_{\tilde{Z}\tilde{Z}} B^\top$ where

$$K_{ZZ} = \begin{bmatrix} k(\mathbf{z}^1, \mathbf{z}^1) & \cdots & k(\mathbf{z}^1, \mathbf{z}^{n_0}) \\ \vdots & \ddots & \vdots \\ k(\mathbf{z}^{n_0}, \mathbf{z}^1) & \cdots & k(\mathbf{z}^{n_0}, \mathbf{z}^{n_0}) \end{bmatrix} \in \mathbb{R}^{n_0 \times n_0}.$$

So

$$\left\| K_{\tilde{Z}\tilde{Z}}^{1/2} B^\top B K_{\tilde{Z}\tilde{Z}}^{1/2}\right\| = \left\| B K_{\tilde{Z}\tilde{Z}} B^\top\right\| \le \left\| B K_{\tilde{Z}\tilde{Z}} B^\top\right\|_F \le n\kappa^2.$$

Thus, $0 < \frac{\eta}{n} K_{\tilde{Z}\tilde{Z}}^{1/2} B^\top B K_{\tilde{Z}\tilde{Z}}^{1/2} < I$ and

$$(I - \eta S_Z^\top S_Z)^t P_Z h^* = \sum_{r=1}^{\tilde{n}_0}\left[ K_{\tilde{Z}\tilde{Z}}^{-1/2}\left(I - \frac{\eta}{n} K_{\tilde{Z}\tilde{Z}}^{1/2} B^\top B K_{\tilde{Z}\tilde{Z}}^{1/2}\right)^t K_{\tilde{Z}\tilde{Z}}^{1/2} \mathbf{a}\right]_r k_{\tilde{\mathbf{z}}_r} \to 0$$

as $t \to \infty$. Similarly, we get $(I - \eta S_Z^\top S_Z)^t u_1 \to 0$ as $t \to \infty$. Therefore, we attain $\mathcal{G}^t u_1 \to P_Z h^*$ as $t \to \infty$ for any $u_1 \in W$.

From this fact, DCL-KR has the recurrence relation

$$f_t = P_Z \sum_{i=1}^{m} \frac{n_i}{n}\left(\overline{T}_{k,X_i}^E f_{t-1} + \eta \sum_{s=0}^{E-1} \overline{T}_{k,X_i}^s S_{D_i}^\top \mathbf{y}_i\right) \tag{6}$$

where $f_t = f_{i,t}$ for any $i = 1, \cdots, m$ and $\overline{T}_{k,X_i} := I - \eta T_{k,X_i}$ for $i = 1, \cdots, m$. Then we obtain a closed form

$$f_t = \left(P_Z \sum_{i=1}^{m} \frac{n_i}{n} \overline{T}_{k,X_i}^E\right)^t f_0 + \sum_{j=0}^{t-1}\left(P_Z \sum_{i=1}^{m} \frac{n_i}{n} \overline{T}_{k,X_i}^E\right)^j P_Z \sum_{i=1}^{m} \frac{n_i}{n} \eta \sum_{s=0}^{E-1} \overline{T}_{k,X_i}^s S_{D_i}^\top \mathbf{y}_i.$$

We first compute

$$f_0^* - \left(\left(P_Z \sum_{i=1}^{m} \frac{n_i}{n} \overline{T}_{k,X_i}^E\right)^t f_0^* + \sum_{j=0}^{t-1}\left(P_Z \sum_{i=1}^{m} \frac{n_i}{n} \overline{T}_{k,X_i}^E\right)^j P_Z \sum_{i=1}^{m} \frac{n_i}{n} \eta \sum_{s=0}^{E-1} \overline{T}_{k,X_i}^s T_{k,X_i} f_0^*\right).$$

From

$$\eta \sum_{s=0}^{E-1} \overline{T}_{k,X_i}^s T_{k,X_i} = \eta \sum_{s=0}^{E-1}(I - \eta T_{k,X_i})^s T_{k,X_i} = I - (I - \eta T_{k,X_i})^E,$$

we have

$$f_0^* - \left(\left(P_Z \sum_{i=1}^{m} \frac{n_i}{n} \overline{T}_{k,X_i}^E\right)^t f_0^* + \sum_{j=0}^{t-1}\left(P_Z \sum_{i=1}^{m} \frac{n_i}{n} \overline{T}_{k,X_i}^E\right)^j P_Z \sum_{i=1}^{m} \frac{n_i}{n} \eta \sum_{s=0}^{E-1} \overline{T}_{k,X_i}^s T_{k,X_i} f_0^*\right)$$

$$= f_0^* - \left(\left(P_Z \sum_{i=1}^{m} \frac{n_i}{n} \overline{T}_{k,X_i}^E\right)^t f_0^* + \sum_{j=0}^{t-1}\left(P_Z \sum_{i=1}^{m} \frac{n_i}{n} \overline{T}_{k,X_i}^E\right)^j P_Z\left(I - \sum_{i=1}^{m} \frac{n_i}{n} \overline{T}_{k,X_i}^E\right) f_0^*\right)$$

$$= \left(I + \left(\sum_{i=1}^{m} \frac{n_i}{n} P_Z \overline{T}_{k,X_i}^E\right) + \cdots + \left(\sum_{i=1}^{m} \frac{n_i}{n} P_Z \overline{T}_{k,X_i}^E\right)^{t-1}\right)(I - P_Z) f_0^*$$

$$= (I - P_Z) f_0^* + \left(I + \cdots + \left(\sum_{i=1}^{m} \frac{n_i}{n} P_Z \overline{T}_{k,X_i}^E\right)^{t-2}\right) P_Z\left(\sum_{i=1}^{m} \frac{n_i}{n} \overline{T}_{k,X_i}^E - I\right)(I - P_Z) f_0^*.$$

where the last equality follows from $P_Z(I - P_Z) = 0$. Thus, we obtain the equality

$$
\iota_{\rho_{\mathbf{x}}}(f_t - f_0^*)
$$

$$
= \iota_{\rho_{\mathbf{x}}} \left( P_Z \sum_{i=1}^{m} \frac{n_i}{n} \overline{T}_{k,X_i}^E \right)^t (f_0 - f_0^*)
$$

$$
+ \iota_{\rho_{\mathbf{x}}} \sum_{j=0}^{t-1} \left( P_Z \sum_{i=1}^{m} \frac{n_i}{n} \overline{T}_{k,X_i}^E \right)^j P_Z \sum_{i=1}^{m} \frac{n_i}{n} \eta \sum_{s=0}^{E-1} \overline{T}_{k,X_i}^s S_{D_i}^\top (\mathbf{y}_i - S_{D_i} f_0^*) - \iota_{\rho_{\mathbf{x}}}(I - P_Z) f_0^*
$$

$$
+ \iota_{\rho_{\mathbf{x}}} \left( I + \cdots + \left( \sum_{i=1}^{m} \frac{n_i}{n} P_Z \overline{T}_{k,X_i}^E \right)^{t-2} \right) P_Z \left( I - \sum_{i=1}^{m} \frac{n_i}{n} \overline{T}_{k,X_i}^E \right) (I - P_Z) f_0^*. \tag{7}
$$

### A.2.2 Norm Bound of First Term in (7)

We first bound the norm of the first term in (7) as

$$
\left\| \iota_{\rho_{\mathbf{x}}} \left( P_Z \sum_{i=1}^{m} \frac{n_i}{n} \overline{T}_{k,X_i}^E \right)^t (f_0 - f_0^*) \right\|_{L_{\rho_{\mathbf{x}}}^2}
$$

$$
\leq \left\| T_{k,\rho_{\mathbf{x}}}^{1/2}(T_{k,X} + \lambda I)^{-1/2} \right\| \left\| (T_{k,X} + \lambda I)^{1/2} \left( P_Z \sum_{i=1}^{m} \frac{n_i}{n} \overline{T}_{k,X_i}^E \right)^t P_Z f_0^* \right\|_{\mathbb{H}_k}
$$

$$
+ \left\| T_{k,\rho_{\mathbf{x}}}^{1/2}(T_{k,X} + \lambda I)^{-1/2} \right\| \left\| (T_{k,X} + \lambda I)^{1/2} \left( P_Z \sum_{i=1}^{m} \frac{n_i}{n} \overline{T}_{k,X_i}^E \right)^t \right\| \|(I - P_Z) f_0^*\|_{\mathbb{H}_k} \tag{8}
$$

where $\lambda > 0$. The first term in (8) is bounded as

$$
\left\| T_{k,\rho_{\mathbf{x}}}^{1/2}(T_{k,X} + \lambda I)^{-1/2} \right\| \left\| (T_{k,X} + \lambda I)^{1/2} \left( P_Z \sum_{i=1}^{m} \frac{n_i}{n} \overline{T}_{k,X_i}^E \right)^t P_Z f_0^* \right\|_{\mathbb{H}_k}
$$

$$
\leq \left\| T_{k,\rho_{\mathbf{x}}}^{1/2}(T_{k,X} + \lambda I)^{-1/2} \right\| \left\| (T_{k,X} + \lambda I)^{1/2} \left( P_Z \sum_{i=1}^{m} \frac{n_i}{n} \overline{T}_{k,X_i}^E \right)^t P_Z (T_{k,X} + \lambda I)^{r-1/2} \right\|
$$

$$
\cdot \left\| (T_{k,X} + \lambda I)^{-(r-1/2)} T_{k,\rho_{\mathbf{x}}}^{r-1/2} g_0^* \right\|_{\mathbb{H}_k}.
$$

Note that

$$
\left\| (T_{k,X} + \lambda I)^{1/2} \left( P_Z \sum_{i=1}^{m} \frac{n_i}{n} \overline{T}_{k,X_i}^E \right)^t P_Z (T_{k,X} + \lambda I)^{r-1/2} \right\|
$$

$$
\leq \left\| (T_{k,X} + \lambda I)^{1/2} \left( P_Z \sum_{i=1}^{m} \frac{n_i}{n} \overline{T}_{k,X_i}^E P_Z \right)^{t/2r} \right\|^{2r}
$$

by Lemma A.8. Set $A_i = \overline{T}_{k,X_i}^E \Leftrightarrow T_{k,X_i} = \frac{1}{\eta}(I - A_i^{1/E})$. We observe that

$$
\left\| (T_{k,X} + \lambda I)^{1/2} \left( P_Z \sum_{i=1}^{m} \frac{n_i}{n} \overline{T}_{k,X_i}^E P_Z \right)^{t/2r} \right\|^{2r}
$$

$$
= \left\| \left( P_Z \sum_{i=1}^{m} \frac{n_i}{n} \overline{T}_{k,X_i}^E P_Z \right)^{t/2r} P_Z (T_{k,X} + \lambda I) P_Z \left( P_Z \sum_{i=1}^{m} \frac{n_i}{n} \overline{T}_{k,X_i}^E P_Z \right)^{t/2r} \right\|^r
$$

$$
\leq \left( \frac{1}{\eta} \left\| \left( \sum_{i=1}^{m} \frac{n_i}{n} P_Z A_i P_Z \right)^{t/2r} \left( I - \sum_{i=1}^{n} \frac{n_i}{n} P_Z A_i P_Z \right) \left( \sum_{i=1}^{m} \frac{n_i}{n} P_Z A_i P_Z \right)^{t/2r} \right\| + \lambda \right)^r
$$

where the equality follows from $0 \leq A_i \leq I \Rightarrow A_i^{1/E} \geq A_i$ and $I \geq P_Z$. Since $\sup_{x \in [0,1]} x^{t/r}(1 - x) = \frac{r}{t+r} \cdot (\frac{t}{t+r})^{t/r}$, we attain the inequality

$$
\left\| (T_{k,X} + \lambda I)^{1/2} \left( P_Z \sum_{i=1}^{m} \frac{n_i}{n} \overline{T}_{k,X_i}^E P_Z \right)^{t/2r} \right\|^{2r} \leq \left( \frac{r}{t+r} \cdot \frac{1}{\eta} \left( \frac{t}{t+r} \right)^{t/r} + \lambda \right)^r. \tag{9}
$$

Next, Lemma A.8 gives

$$\left\| (T_{k,X} + \lambda I)^{-(r-1/2)} T_{k,\rho_{\mathbf{x}}}^{r-1/2} \right\| \leq \left\| (T_{k,X} + \lambda I)^{-(r-1/2)} (T_{k,\rho_{\mathbf{x}}} + \lambda I)^{r-1/2} \right\|$$

$$\leq \left\| (T_{k,X} + \lambda I)^{-1} (T_{k,\rho_{\mathbf{x}}} + \lambda I) \right\|^{r-1/2}$$

and

$$\left\| T_{k,\rho_{\mathbf{x}}}^{1/2} (T_{k,X} + \lambda I)^{-1/2} \right\| \leq \left\| (T_{k,\rho_{\mathbf{x}}} + \lambda I)^{1/2} (T_{k,X} + \lambda I)^{-1/2} \right\| \leq \left\| (T_{k,\rho_{\mathbf{x}}} + \lambda I)(T_{k,X} + \lambda I)^{-1} \right\|^{1/2}.$$

By Lemma A.10,

$$\| (T_{k,\rho_{\mathbf{x}}} + \lambda I)(T_{k,X} + \lambda I)^{-1} \| \leq 2 + 2 \left( \left( \frac{2\kappa^2}{n\lambda} + \sqrt{\frac{4\kappa^2 \mathcal{N}(\lambda)}{n\lambda}} \right) \log(2/\delta) \right)^2 \tag{10}$$

holds with confidence at least $1 - \delta$ where $\delta \in (0, 1)$. Combining (9) and (10) and applying $\frac{r}{t+r} \leq \frac{1}{t}$ and $(\frac{t}{t+r})^{t/r} \leq \frac{1}{2}$ yield

$$\left\| T_{k,\rho_{\mathbf{x}}}^{1/2} (T_{k,X} + \lambda I)^{-1/2} \right\| \left\| (T_{k,X} + \lambda I)^{1/2} \left( P_Z \sum_{i=1}^m \frac{n_i}{n} \overline{T}_{k,X_i}^E \right)^t P_Z f_0^* \right\|_{\mathbb{H}_k}$$

$$\leq \left( \frac{r}{t+r} \cdot \frac{1}{\eta} \left( \frac{t}{t+r} \right)^{t/r} + \lambda \right)^r \| g_0^* \|_{\mathbb{H}_k} \left( 2 + 2 \left( \left( \frac{2\kappa^2}{n\lambda} + \sqrt{\frac{4\kappa^2 \mathcal{N}(\lambda)}{n\lambda}} \right) \log(2/\delta) \right)^2 \right)^r$$

$$\leq R \left( \frac{1}{2\eta t} + \lambda \right)^r \left( 2 + 2 \left( \frac{2\kappa^2}{n\lambda} + \sqrt{\frac{4\kappa^2 \mathcal{N}(\lambda)}{n\lambda}} \right)^2 \right)^r (\log(4/\delta))^{2r}$$

with confidence at least $1 - \delta$ where $\delta \in (0, 1)$. Therefore, putting $\lambda = n^{-\frac{1}{2r+s}}$ yields

$$\mathbb{E} \left[ \left\| T_{k,\rho_{\mathbf{x}}}^{1/2} (T_{k,X} + \lambda I)^{-1/2} \right\| \left\| (T_{k,X} + \lambda I)^{1/2} \left( P_Z \sum_{i=1}^m \frac{n_i}{n} \overline{T}_{k,X_i}^E \right)^t P_Z f_0^* \right\|_{\mathbb{H}_k} \right]$$

$$\leq \left( \frac{1}{2\eta t} + n^{-\frac{1}{2r+s}} \right)^r R \cdot 4\Gamma(2r+1) \left( 2 + 2(2\kappa^2 + 2\kappa \sqrt{C_s'})^2 \right)^r$$

$$\lesssim \left( \frac{1}{t} + n^{-\frac{1}{2r+s}} \right)^r.$$

Here, we apply the fact that $\mathbb{E} A = \int_0^\infty \mathbb{P}(A \geq t)\, dt$ for $A \geq 0$.

We next turn to bound the second term in (8). Note that

$$\left\| (T_{k,X} + \lambda I)^{1/2} \left( P_Z \sum_{i=1}^m \frac{n_i}{n} \overline{T}_{k,X_i}^E \right)^t \right\|$$

$$= \left\| \left( \sum_{i=1}^m \frac{n_i}{n} \overline{T}_{k,X_i}^E P_Z \right)^t (T_{k,X} + \lambda I) \left( P_Z \sum_{i=1}^m \frac{n_i}{n} \overline{T}_{k,X_i}^E \right)^t \right\|^{1/2}$$

$$\leq \left\| \sum_{i=1}^m \frac{n_i}{n} \overline{T}_{k,X_i}^E \right\| \cdot \left\| \left( P_Z \sum_{i=1}^m \frac{n_i}{n} \overline{T}_{k,X_i}^E P_Z \right)^{t-1} P_Z (T_{k,X} + \lambda I) P_Z \left( P_Z \sum_{i=1}^m \frac{n_i}{n} \overline{T}_{k,X_i}^E P_Z \right)^{t-1} \right\|^{1/2}$$

$$\leq \left\| \left( P_Z \sum_{i=1}^m \frac{n_i}{n} \overline{T}_{k,X_i}^E P_Z \right)^{t-1} P_Z (T_{k,X} + \lambda I) P_Z \left( P_Z \sum_{i=1}^m \frac{n_i}{n} \overline{T}_{k,X_i}^E P_Z \right)^{t-1} \right\|^{1/2}.$$

Set $A_i = \overline{T}_{k,X_i}^E$. Using a similar argument as before gives

$$\left\| \left( P_Z \sum_{i=1}^m \frac{n_i}{n} \overline{T}_{k,X_i}^E P_Z \right)^{t-1} P_Z (T_{k,X} + \lambda I) P_Z \left( P_Z \sum_{i=1}^m \frac{n_i}{n} \overline{T}_{k,X_i}^E P_Z \right)^{t-1} \right\|^{1/2}$$

$$\leq \left( \frac{1}{\eta(2t-1)} + \lambda \right)^{1/2} \leq \left( \frac{1}{\eta t} + \lambda \right)^{1/2}.$$

Since $Z$ and $X$ are independent, we have

$$\mathbb{E}\left[\left\|T_{k,\rho_{\mathbf{x}}}^{1/2}(T_{k,X}+\lambda I)^{-1/2}\right\|\left\|(T_{k,X}+\lambda I)^{1/2}\left(P_Z\sum_{i=1}^{m}\frac{n_i}{n}\overline{T}_{k,X_i}^{E}\right)^{t}\right\|\|(I-P_Z)f_0^*\|_{\mathbb{H}_k}\right]$$

$$\leq\left(\frac{1}{\eta t}+\lambda\right)^{1/2}\mathbb{E}\left\|T_{k,\rho_{\mathbf{x}}}^{1/2}(T_{k,X}+\lambda I)^{-1/2}\right\|\cdot\mathbb{E}\|(I-P_Z)f_0^*\|_{\mathbb{H}_k}.$$

We already see that

$$\left\|T_{k,\rho_{\mathbf{x}}}^{1/2}(T_{k,X}+\lambda I)^{-1/2}\right\|\leq\left(2+2\left(\left(\frac{2\kappa^2}{n\lambda}+\sqrt{\frac{4\kappa^2\mathcal{N}(\lambda)}{n\lambda}}\right)\log(2/\delta)\right)^2\right)^{1/2}$$

$$\leq\left(2+2\left(\frac{2\kappa^2}{n\lambda}+\sqrt{\frac{4\kappa^2\mathcal{N}(\lambda)}{n\lambda}}\right)^2\right)^{1/2}\log(4/\delta)$$

holds with confidence at least $1-\delta$ where $\delta\in(0,1)$ and so

$$\mathbb{E}\left\|T_{k,\rho_{\mathbf{x}}}^{1/2}(T_{k,X}+\lambda I)^{-1/2}\right\|\leq 4\left(2+2(2\kappa^2+2\kappa\sqrt{C_s'})^2\right)^{1/2}$$

by putting $\lambda=n^{-\frac{1}{2r+s}}$ as before.

The remaining part is to bound $\mathbb{E}\|(I-P_Z)f_0^*\|_{\mathbb{H}_k}$. Applying Lemma A.9 yields $\|(I-P_Z)f_0^*\|_{\mathbb{H}_k}\leq\lambda_0^{1/2}\|(T_{k,Z}+\lambda_0 I)^{-1/2}T_{k,\rho_{\mathbf{x}}}^{r-1/2}\|\|g_0^*\|_{\mathbb{H}_k}$ where $\lambda_0>0$. Then Lemma A.8 gives

$$\lambda_0^{1/2}\|(T_{k,Z}+\lambda_0 I)^{-1/2}T_{k,\rho_{\mathbf{x}}}^{r-1/2}\|\|g_0^*\|_{\mathbb{H}_k}$$

$$\leq R\lambda_0^{1/2}\|(T_{k,Z}+\lambda_0 I)^{-(1-r)}\|\|(T_{k,Z}+\lambda_0 I)^{-(r-1/2)}T_{k,\rho_{\mathbf{x}}}^{r-1/2}\|$$

$$\leq R\lambda_0^{r-1/2}\|(T_{k,Z}+\lambda_0 I)^{-1/2}T_{k,\rho_{\mathbf{x}}}^{1/2}\|^{2r-1}.$$

From $\frac{d\rho_{\mathbf{x}}}{d\tilde{\rho}_{\mathbf{x}}}\leq B$, we obtain

$$\|T_{k,\rho_{\mathbf{x}}}^{1/2}(T_{k,Z}+\lambda_0 I)^{-1/2}\|=\|\iota_{\rho_{\mathbf{x}}}(T_{k,Z}+\lambda_0 I)^{-1/2}\|$$

$$\leq B^{1/2}\|\iota_{\tilde{\rho}_{\mathbf{x}}}(T_{k,Z}+\lambda_0 I)^{-1/2}\|=B^{1/2}\|T_{k,\tilde{\rho}_{\mathbf{x}}}^{1/2}(T_{k,Z}+\lambda_0 I)^{-1/2}\|.$$

Set $\lambda_0=128(\kappa^2+1)^2(\log n_0)^3/n_0$ where we assume $n$ is sufficiently large such that $\lambda_0\leq 1$ and $\mathcal{N}_{\tilde{\rho}_{\mathbf{x}}}(\lambda_0)\geq 1$ for $n_0\geq n^{\frac{1}{2r+s}}(\log n)^3$. By Lemma A.8 and Lemma A.12,

$$\|T_{k,\tilde{\rho}_{\mathbf{x}}}^{1/2}(T_{k,Z}+\lambda_0 I)^{-1/2}\|\leq\|(T_{k,\tilde{\rho}_{\mathbf{x}}}+\lambda_0 I)^{1/2}(T_{k,Z}+\lambda_0 I)^{-1/2}\|\leq\sqrt{2}$$

holds with confidence at least $1-\delta$ where $\delta\in[4\exp(-1/4(\kappa^2+1)\mathcal{B}_0),1)$ and

$$\mathcal{B}_0=\frac{1+\log\mathcal{N}_{\tilde{\rho}_{\mathbf{x}}}(\lambda_0)}{\lambda_0 n_0}+\sqrt{\frac{1+\log\mathcal{N}_{\tilde{\rho}_{\mathbf{x}}}(\lambda_0)}{\lambda_0 n_0}}.$$

Since

$$\|(I-P_Z)f_0^*\|_{\mathbb{H}_k}\leq\|f_0^*\|_{\mathbb{H}_k}=\|T_{k,\rho_{\mathbf{x}}}^{r-1/2}g_0^*\|_{\mathbb{H}_k}\leq\|T_{k,\rho_{\mathbf{x}}}\|^{r-1/2}\|g_0^*\|_{\mathbb{H}_k}\leq R\kappa^{2r-1},$$

we have

$$\mathbb{E}\|(I-P_Z)f_0^*\|_{\mathbb{H}_k}\leq R\lambda_0^{r-1/2}B^{r-1/2}2^{r-1/2}+R\kappa^{2r-1}\cdot 4\exp\left(-\frac{1}{4(\kappa^2+1)\mathcal{B}_0}\right).$$

From $n_0\geq\kappa^2 e$, we get

$$\mathcal{B}_0\leq\frac{\log\kappa^2 e+\log n_0}{128(\kappa^2+1)^2(\log n_0)^3}+\sqrt{\frac{\log\kappa^2 e+\log n_0}{128(\kappa^2+1)^2(\log n_0)^3}}$$

$$\leq\frac{2\log n_0}{128(\kappa^2+1)^2(\log n_0)^3}+\sqrt{\frac{2\log n_0}{128(\kappa^2+1)^2(\log n_0)^3}}\leq\frac{1}{4(\kappa^2+1)\log n_0}$$

and so $R\kappa^{2r-1} \cdot 4\exp\left(-\frac{1}{4(\kappa^2+1)\mathcal{B}_0}\right) \le 4R\kappa^{2r-1} \cdot \frac{1}{n_0}$. Therefore,

$$\mathbb{E}\left[\left\|T_{k,\rho_\mathbf{x}}^{1/2}(T_{k,X}+\lambda I)^{-1/2}\right\|\left\|(T_{k,X}+\lambda I)^{1/2}\left(P_Z\sum_{i=1}^m \frac{n_i}{n}\overline{T}_{k,X_i}^E\right)^t\right\|\|(I-P_Z)f_0^*\|_{\mathbb{H}_k}\right]$$

$$\le \left(\frac{1}{\eta t}+n^{-\frac{1}{2r+s}}\right)^{1/2} 4\left(2+2(2\kappa^2+2\kappa\sqrt{C_s'})^2\right)^{1/2}\left(R\lambda_0^{r-1/2}B^{r-1/2}2^{r-1/2}+4R\kappa^{2r-1}\cdot\frac{1}{n_0}\right)$$

$$\lesssim B^{r-1/2}\left(\frac{1}{t}+n^{-\frac{1}{2r+s}}\right)^{1/2} n^{-\frac{r-1/2}{2r+s}}$$

where the last inequality comes from $n_0 \ge n^{\frac{1}{2r+s}}(\log n)^3$.

### A.2.3   Norm Bound of Second Term in (7)

Set

$$P = \begin{bmatrix} \sum_{s=0}^{E-1}(I-\eta S_{D_1}S_{D_1}^\top)^s & 0 & \cdots & 0 \\ 0 & \sum_{s=0}^{E-1}(I-\eta S_{D_2}S_{D_2}^\top)^s & \cdots & 0 \\ \vdots & \vdots & \ddots & \vdots \\ 0 & 0 & \cdots & \sum_{s=0}^{E-1}(I-\eta S_{D_m}S_{D_m}^\top)^s \end{bmatrix}.$$

Note that

$$I-\eta S_D^\top P S_D = I - \sum_{i=1}^m \frac{n_i}{n}\eta S_{D_i}^\top \sum_{s=0}^{E-1}(I-\eta S_{D_i}S_{D_i}^\top)^s S_{D_i}$$

$$= I - \sum_{i=1}^m \frac{n_i}{n}(I-(I-\eta S_{D_i}^\top S_{D_i})^E) = \sum_{i=1}^m \frac{n_i}{n}\overline{T}_{k,X_i}^E. \qquad (11)$$

Then the second term in (7) becomes

$$\iota_{\rho_\mathbf{x}}\sum_{j=0}^{t-1}\left(P_Z\sum_{i=1}^m \frac{n_i}{n}\overline{T}_{k,X_i}^E\right)^j P_Z\sum_{i=1}^m \frac{n_i}{n}\eta\sum_{s=0}^{E-1}\overline{T}_{k,X_i}^s S_{D_i}^\top(\mathbf{y}_i-S_{D_i}f_0^*)$$

$$= \iota_{\rho_\mathbf{x}}\sum_{j=0}^{t-1}\left(P_Z-\eta P_Z S_D^\top P S_D\right)^j \eta P_Z S_D^\top P(\mathbf{y}-S_D f_0^*).$$

We can see that

$$\left\|\iota_{\rho_\mathbf{x}}\sum_{j=0}^{t-1}\left(P_Z-\eta P_Z S_D^\top P S_D\right)^j \eta P_Z S_D^\top P(\mathbf{y}-S_D f_0^*)\right\|_{L_{\rho_\mathbf{x}}^2}$$

$$\le \|T_{k,\rho_\mathbf{x}}^{1/2}(T_{k,X}+\lambda I)^{-1/2}\|\left\|(T_{k,X}+\lambda I)^{1/2}\sum_{j=0}^{t-1}\left(P_Z-\eta P_Z S_D^\top P S_D\right)^j \eta P_Z S_D^\top P(\mathbf{y}-S_D f_0^*)\right\|_{\mathbb{H}_k}$$

$$\le \|T_{k,\rho_\mathbf{x}}^{1/2}(T_{k,X}+\lambda I)^{-1/2}\|\left(\left\|T_{k,X}^{1/2}\sum_{j=0}^{t-1}\left(P_Z-\eta P_Z S_D^\top P S_D\right)^j \eta P_Z S_D^\top P(\mathbf{y}-S_D f_0^*)\right\|_{\mathbb{H}_k}\right.$$

$$\left.+\lambda^{1/2}\left\|\sum_{j=0}^{t-1}\left(P_Z-\eta P_Z S_D^\top P S_D\right)^j \eta P_Z S_D^\top P(\mathbf{y}-S_D f_0^*)\right\|_{\mathbb{H}_k}\right).$$

We first bound the expectation of the first term in the above. By the Cauchy-Schwartz inequality, we have

$$\mathbb{E}\left[\|T_{k,\rho_\mathbf{x}}^{1/2}(T_{k,X}+\lambda I)^{-1/2}\|\left\|T_{k,X}^{1/2}\sum_{j=0}^{t-1}\left(P_Z-\eta P_Z S_D^\top P S_D\right)^j \eta P_Z S_D^\top P(\mathbf{y}-S_D f_0^*)\right\|_{\mathbb{H}_k}\right]$$

$$\le \left(\mathbb{E}\|T_{k,\rho_\mathbf{x}}^{1/2}(T_{k,X}+\lambda I)^{-1/2}\|^2\right)^{1/2}\left(\mathbb{E}\left\|T_{k,X}^{1/2}\sum_{j=0}^{t-1}\left(P_Z-\eta P_Z S_D^\top P S_D\right)^j \eta P_Z S_D^\top P(\mathbf{y}-S_D f_0^*)\right\|_{\mathbb{H}_k}^2\right)^{1/2}.$$

Observe that

$$\left\| T_{k,X}^{1/2} \sum_{j=0}^{t-1} \left( P_Z - \eta P_Z S_D^\top P S_D \right)^j \eta P_Z S_D^\top P(\mathbf{y} - S_D f_0^*) \right\|_{\mathbb{H}_k}$$

$$= \left\| S_D \sum_{j=0}^{t-1} \left( P_Z - \eta P_Z S_D^\top P S_D \right)^j \eta P_Z S_D^\top P(\mathbf{y} - S_D f_0^*) \right\|_2$$

and

$$S_D \sum_{j=0}^{t-1} \left( P_Z - \eta P_Z S_D^\top P S_D \right)^j \eta P_Z S_D^\top P(\mathbf{y} - S_D f_0^*)$$

$$= P^{-1/2}(I - (I - \eta P^{1/2} S_D P_Z S_D^\top P^{1/2})^t) P^{1/2}(\mathbf{y} - S_D f_0^*)$$

$$= (I - (I - \eta S_D P_Z S_D^\top P)^t)(\mathbf{y} - S_D f_0^*).$$

Using $\mathbb{E}(\mathbf{y} - S_D f_0^*)(\mathbf{y} - S_D f_0^*)^\top \le \gamma^2 I$, we have

$$\mathbb{E}\left[ \left\| (I - (I - \eta S_D P_Z S_D^\top P)^t)(\mathbf{y} - S_D f_0^*) \right\|_2^2 \Big| X, Z \right]$$

$$= \frac{1}{n} \mathrm{tr}\left( (I - (I - \eta S_D P_Z S_D^\top P)^t) \mathbb{E}\left[ (\mathbf{y} - S_D f_0^*)(\mathbf{y} - S_D f_0^*)^\top \right] (I - (I - \eta S_D P_Z S_D^\top P)^t)^\top \right)$$

$$\le \frac{\gamma^2}{n} \left\| (I - (I - \eta S_D P_Z S_D^\top P)^t) \right\|_{HS}^2 \le \frac{\gamma^2 E}{n} \left\| (I - (I - \eta P^{1/2} S_D P_Z S_D^\top P^{1/2})^t) \right\|_{HS}^2$$

where the last inequality follows from the fact that $\|AB\|_{HS} \le \|A\|\|B\|_{HS}$, $\|AB\|_{HS} \le \|A\|_{HS}\|B\|$, and

$$\|P^{1/2}\|^2 \|P^{-1/2}\|^2 \le E \left( \sum_{s=0}^{E-1} (1 - \eta \kappa^2)^s \right)^{-1} \le E.$$

Since

$$0 \le \eta P^{1/2} S_D P_Z S_D^\top P^{1/2} \le \eta P^{1/2} S_D S_D^\top P^{1/2} \le I \tag{12}$$

which follows from (11), we can see that

$$0 \le \lambda_i(\eta P^{1/2} S_D P_Z S_D^\top P^{1/2}) \le \lambda_i(\eta P^{1/2} S_D S_D^\top P^{1/2}) \le 1$$

$$\Rightarrow \quad 0 \le \lambda_i(I - (I - \eta P^{1/2} S_D P_Z S_D^\top P^{1/2})^t) \le \lambda_i(I - (I - \eta P^{1/2} S_D S_D^\top P^{1/2})^t) \le 1$$

where $\lambda_i(\cdot)$ is the $i$th largest eigenvalue of a given operator. Therefore,

$$\frac{\gamma^2 E}{n} \left\| (I - (I - \eta P^{1/2} S_D P_Z S_D^\top P^{1/2})^t) \right\|_{HS}^2 \le \frac{\gamma^2 E}{n} \left\| (I - (I - \eta P^{1/2} S_D S_D^\top P^{1/2})^t) \right\|_{HS}^2.$$

Using (12) and $1 \wedge u^2 \le 1 \wedge u$ for $u \ge 0$ lead to

$$\lambda_i(I - (I - \eta P^{1/2} S_D S_D^\top P^{1/2})^t)^2 = (1 - (1 - \eta \lambda_i(P^{1/2} S_D S_D^\top P^{1/2}))^t)^2$$

$$\le 1 \wedge (\eta^2 t^2 \lambda_i(P^{1/2} S_D S_D^\top P^{1/2})^2)$$

$$\le 1 \wedge (\eta t \lambda_i(P^{1/2} S_D S_D^\top P^{1/2}))$$

$$\le 1 \wedge (\eta t E \hat{\lambda}_i)$$

where $\hat{\lambda}_1 \ge \cdots \ge \hat{\lambda}_n$ are eigenvalues of $S_D S_D^\top$, the first inequality comes from the Bernoulli inequality, and the last inequality follows from the fact that $\|P\| \le E$. We define

$$\mathcal{R}(\epsilon) = \sqrt{\frac{1}{n} \sum_{i=1}^n (\hat{\lambda}_i \wedge \epsilon^2)}.$$

Then

$$\frac{\gamma^2 E}{n} \left\| (I - (I - \eta P^{1/2} S_D S_D^\top P^{1/2})^t) \right\|_{HS}^2 \le \gamma^2 \eta t E^2 \cdot \mathcal{R}\left( \frac{1}{\sqrt{\eta t E}} \right)^2.$$

Similarly as in Appendix A.2.2, putting $\lambda = n^{-\frac{1}{2r+s}}$ gives

$$\mathbb{E}\|T_{k,\rho_\mathbf{x}}^{1/2}(T_{k,X} + \lambda I)^{-1/2}\|^2 \le 2 + 4\Gamma(3)(2\kappa^2 + 2\kappa\sqrt{C_s'})^2.$$

Therefore, the Cauchy-Schwartz inequality gives a bound as

$$
\mathbb{E}\left[\|T_{k,\rho_{\mathbf{x}}}^{1/2}(T_{k,X}+\lambda I)^{-1/2}\|\left\|T_{k,X}^{1/2}\sum_{j=0}^{t-1}\left(P_Z-\eta P_Z S_D^\top P S_D\right)^j \eta P_Z S_D^\top P(\mathbf{y}-S_D f_0^*)\right\|_{\mathbb{H}_k}\right]
$$

$$
\leq \sqrt{(2+4\Gamma(3)(2\kappa^2+2\kappa\sqrt{C_s'})^2)\gamma^2\eta t E^2}\cdot\left(\mathbb{E}\mathcal{R}\left(\frac{1}{\sqrt{\eta t E}}\right)^2\right)^{1/2}.
$$

We now bound the expectation of

$$
\lambda^{1/2}\|T_{k,\rho_{\mathbf{x}}}^{1/2}(T_{k,X}+\lambda I)^{-1/2}\|\left\|\sum_{j=0}^{t-1}\left(P_Z-\eta P_Z S_D^\top P S_D\right)^j \eta P_Z S_D^\top P(\mathbf{y}-S_D f_0^*)\right\|_{\mathbb{H}_k}.
$$

By the Cauchy-Schwartz inequality and the same argument as before, we have

$$
\mathbb{E}\left[\lambda^{1/2}\|T_{k,\rho_{\mathbf{x}}}^{1/2}(T_{k,X}+\lambda I)^{-1/2}\|\left\|\sum_{j=0}^{t-1}\left(P_Z-\eta P_Z S_D^\top P S_D\right)^j \eta P_Z S_D^\top P(\mathbf{y}-S_D f_0^*)\right\|_{\mathbb{H}_k}\right]
$$

$$
\leq (2+4\Gamma(3)(2\kappa^2+2\kappa\sqrt{C_s'})^2)^{1/2}\left(\lambda\cdot\mathbb{E}\left\|\sum_{j=0}^{t-1}\left(P_Z-\eta P_Z S_D^\top P S_D P_Z\right)^j \eta P_Z S_D^\top P(\mathbf{y}-S_D f_0^*)\right\|_{\mathbb{H}_k}^2\right)^{1/2}.
$$

Also, the same argument as before yields

$$
\mathbb{E}\left\|\sum_{j=0}^{t-1}\left(P_Z-\eta P_Z S_D^\top P S_D P_Z\right)^j \eta P_Z S_D^\top P(\mathbf{y}-S_D f_0^*)\right\|_{\mathbb{H}_k}^2 = \frac{1}{n}\mathbb{E}\left[(\mathbf{y}-S_D f_0^*)^\top A(\mathbf{y}-S_D f_0^*)\right]
$$

$$
\leq \frac{\gamma^2}{n}\mathbb{E}[\operatorname{tr}(A)]
$$

where

$$
A = \eta P S_D P_Z\left(\sum_{j=0}^{t-1}\left(P_Z-\eta P_Z S_D^\top P S_D P_Z\right)^j\right)^2 \eta P_Z S_D^\top P
$$

$$
= \eta P S_D P_Z\left(\sum_{j=0}^{t-1}\left(I-\eta P_Z S_D^\top P S_D P_Z\right)^j\right)^2 \eta P_Z S_D^\top P.
$$

To bound $\mathbb{E}[\operatorname{tr}(A)]$, note that

$$
\operatorname{tr}(A) \leq E\cdot\operatorname{tr}\left(\eta P^{1/2}S_D P_Z\left(\sum_{j=0}^{t-1}\left(I-\eta P_Z S_D^\top P S_D P_Z\right)^j\right)^2 \eta P_Z S_D^\top P^{1/2}\right)
$$

$$
= \eta E\cdot\operatorname{tr}\left(\eta P^{1/2}S_D P_Z S_D^\top P^{1/2}\left(\sum_{j=0}^{t-1}(I-\eta P^{1/2}S_D P_Z S_D^\top P^{1/2})^j\right)^2\right).
$$

Let $B = \eta P^{1/2}S_D P_Z S_D^\top P^{1/2}$. Then $0\leq B\leq I$ and

$$
\eta E\cdot\operatorname{tr}\left(\eta P^{1/2}S_D P_Z S_D^\top P^{1/2}\left(\sum_{j=0}^{t-1}(I-\eta P^{1/2}S_D P_Z S_D^\top P^{1/2})^j\right)^2\right)
$$

$$
= \eta E\sum_{i=1}^{n}\lambda_i(B)\left(\sum_{j=0}^{t-1}(1-\lambda_i(B))^j\right)^2 = \eta E\sum_{i=1}^{n}\frac{1}{\lambda_i(B)}(1-(1-\lambda_i(B))^t)^2
$$

$$
\leq \eta E\sum_{i=1}^{n}\frac{1}{\lambda_i(B)}\wedge(t^2\lambda_i(B)) \leq \eta E\sum_{i=1}^{n}t\wedge(t^2\lambda_i(B))
$$

where the first inequality follows from $1-x^t\leq 1\wedge t(1-x)$ and the second inequality follows from $1/x\wedge t^2x\leq t\wedge t^2x$ for all $t\geq 0$ and $x\in[0,1]$. From the fact that

$$
\lambda_i(B)\leq\eta\|P^{1/2}\|^2\lambda_i(S_D P_Z S_D^\top)\leq\eta E\lambda_i(S_D S_D^\top)=\eta E\hat{\lambda}_i,
$$

we have

$$\eta E \sum_{i=1}^{n} t \wedge (t^2 \lambda_i(B)) \le \eta E \sum_{i=1}^{n} t \wedge (\eta t^2 E \hat{\lambda}_i) = n\eta^2 t^2 E^2 \cdot \mathcal{R} \left( \frac{1}{\sqrt{\eta t E}} \right)^2 .$$

Therefore, we obtain

$$\mathbb{E} \left[ \lambda^{1/2} \| T_{k,\rho_{\mathbf{x}}}^{1/2} (T_{k,X} + \lambda I)^{-1/2} \| \left\| \sum_{j=0}^{t-1} \left( P_Z - \eta P_Z S_D^\top P S_D \right)^j \eta P_Z S_D^\top P(\mathbf{y} - S_D f_0^*) \right\|_{\mathbb{H}_k} \right]$$

$$\le \sqrt{(2 + 4\Gamma(3)(2\kappa^2 + 2\kappa\sqrt{C_s'})^2)(\lambda\gamma^2\eta^2 t^2 E^2)} \cdot \left( \mathbb{E}\mathcal{R} \left( \frac{1}{\sqrt{\eta t E}} \right)^2 \right)^{1/2} .$$

In conclusion, we have an upper bound of the norm of the second term in (7) as

$$\mathbb{E} \left\| \iota_{\rho_{\mathbf{x}}} \sum_{j=0}^{t-1} \left( P_Z - \eta P_Z S_D^\top P S_D \right)^j \eta P_Z S_D^\top P(\mathbf{y} - S_D f_0^*) \right\|_{L_{\rho_{\mathbf{x}}}^2}$$

$$\le \sqrt{(2 + 4\Gamma(3)(2\kappa^2 + 2\kappa\sqrt{C_s'})^2)\gamma^2\eta t E^2} \cdot \left( \mathbb{E}\mathcal{R} \left( \frac{1}{\sqrt{\eta t E}} \right)^2 \right)^{1/2}$$

$$+ \sqrt{(2 + 4\Gamma(3)(2\kappa^2 + 2\kappa\sqrt{C_s'})^2)(\lambda\gamma^2\eta^2 t^2 E^2)} \cdot \left( \mathbb{E}\mathcal{R} \left( \frac{1}{\sqrt{\eta t E}} \right)^2 \right)^{1/2}$$

$$\lesssim \left( t^{1/2} + n^{-\frac{1/2}{2r+s}} t \right) \cdot \left( \mathbb{E}\mathcal{R} \left( \frac{1}{\sqrt{\eta t E}} \right)^2 \right)^{1/2}$$

by taking $\lambda = n^{-\frac{1}{2r+s}}$. We will bound $\mathbb{E}\mathcal{R} \left( \frac{1}{\sqrt{\eta t E}} \right)^2$ in Appendix A.2.5.

### A.2.4 Norm Bound of Third and Last Term in (7)

Note that

$$\left\| \iota_{\rho_{\mathbf{x}}} \left( I + \cdots + \left( \sum_{i=1}^{m} \frac{n_i}{n} P_Z \overline{T}_{k,X_i}^E \right)^{t-2} \right) P_Z \left( I - \sum_{i=1}^{m} \frac{n_i}{n} \overline{T}_{k,X_i}^E \right) (I - P_Z) f_0^* \right\|_{L_{\rho_{\mathbf{x}}}^2}$$

$$\le \| T_{k,\rho_{\mathbf{x}}}^{1/2} (T_{k,X} + \lambda I)^{-1/2} \|$$

$$\cdot \left\| (T_{k,X} + \lambda I)^{1/2} \left( I + \cdots + \left( \sum_{i=1}^{m} \frac{n_i}{n} P_Z \overline{T}_{k,X_i}^E \right)^{t-2} \right) P_Z \left( I - \sum_{i=1}^{m} \frac{n_i}{n} \overline{T}_{k,X_i}^E \right)^{1/2} \right\|$$

$$\cdot \left\| \left( I - \sum_{i=1}^{m} \frac{n_i}{n} \overline{T}_{k,X_i}^E \right)^{1/2} (I - P_Z) f_0^* \right\|_{\mathbb{H}_k}$$

where $0 < \lambda \le 1$. From (11) and $0 \le P \le EI$, we have $0 \le I - \sum_{i=1}^{m} \frac{n_i}{n} \overline{T}_{k,X_i}^E = \eta S_D^\top P S_D \le \eta E S_D^\top S_D = \eta E T_{k,X}$. Using this fact, we find that

$$\left\| (T_{k,X} + \lambda I)^{1/2} \left( I + \cdots + \left( \sum_{i=1}^{m} \frac{n_i}{n} P_Z \overline{T}_{k,X_i}^E \right)^{t-2} \right) P_Z \left( I - \sum_{i=1}^{m} \frac{n_i}{n} \overline{T}_{k,X_i}^E \right)^{1/2} \right\|$$

$$\le (\eta E)^{1/2} \left( \left\| T_{k,X}^{1/2} \left( I + \cdots + \left( \sum_{i=1}^{m} \frac{n_i}{n} P_Z \overline{T}_{k,X_i}^E \right)^{t-2} \right) P_Z T_{k,X}^{1/2} \right\| \right.$$

$$\left. + \lambda^{1/2} \left\| \left( I + \cdots + \left( \sum_{i=1}^{m} \frac{n_i}{n} P_Z \overline{T}_{k,X_i}^E \right)^{t-2} \right) P_Z T_{k,X}^{1/2} \right\| \right) .$$

To bound this, we first observe that

$$\left\| T_{k,X}^{1/2} \left( I + \cdots + \left( \sum_{i=1}^{m} \frac{n_i}{n} P_Z \overline{T}_{k,X_i}^E \right)^{t-2} \right) P_Z T_{k,X}^{1/2} \right\|$$

$$\leq \sum_{j=0}^{t-2} \left\| T_{k,X}^{1/2} P_Z \left( \sum_{i=1}^{m} \frac{n_i}{n} P_Z \overline{T}_{k,X_i}^E P_Z \right)^j P_Z T_{k,X}^{1/2} \right\|$$

$$\leq \sum_{j=0}^{t-2} \left\| \left( \sum_{i=1}^{m} \frac{n_i}{n} P_Z \overline{T}_{k,X_i}^E P_Z \right)^{j/2} P_Z T_{k,X} P_Z \left( \sum_{i=1}^{m} \frac{n_i}{n} P_Z \overline{T}_{k,X_i}^E P_Z \right)^{j/2} \right\| \leq \frac{1}{\eta} \left( 1 + \sum_{j=1}^{t-2} \frac{1}{j} \right)$$

where the last inequality follows by a similar calculation as before. On the other hand, we have

$$\left\| \left( I + \cdots + \left( \sum_{i=1}^{m} \frac{n_i}{n} P_Z \overline{T}_{k,X_i}^E \right)^{t-2} \right) P_Z T_{k,X}^{1/2} \right\|$$

$$\leq \sum_{j=0}^{t-2} \left\| \left( \sum_{i=1}^{m} \frac{n_i}{n} P_Z \overline{T}_{k,X_i}^E P_Z \right)^j P_Z T_{k,X}^{1/2} \right\|$$

$$= \sum_{j=0}^{t-2} \left\| \left( \sum_{i=1}^{m} \frac{n_i}{n} P_Z \overline{T}_{k,X_i}^E P_Z \right)^j P_Z T_{k,X} P_Z \left( \sum_{i=1}^{m} \frac{n_i}{n} P_Z \overline{T}_{k,X_i}^E P_Z \right)^j \right\|^{1/2} \leq \frac{1}{\sqrt{\eta}} \left( 1 + \sum_{j=1}^{t-2} \frac{1}{\sqrt{2j}} \right)$$

by the same argument. Using a simple calculation, we get

$$\frac{1}{\eta} \left( 1 + \sum_{j=1}^{t-2} \frac{1}{j} \right) \leq \frac{1}{\eta E} (2 + \log t) E$$

and

$$\frac{1}{\sqrt{\eta}} \left( 1 + \sum_{j=1}^{t-2} \frac{1}{\sqrt{2j}} \right) \leq \frac{1}{\sqrt{\eta}} + \frac{1}{\sqrt{2\eta}} (2\sqrt{t-2} - 1) \leq \frac{1}{(\eta E)^{1/2}} \cdot \sqrt{6tE}.$$

Note that the norm of the third term in (7) is bounded as

$$\| \iota_{\rho_{\mathbf{x}}} (I - P_Z) f_0^* \|_{L^2_{\rho_{\mathbf{x}}}} \leq \left\| T_{k,\rho_{\mathbf{x}}}^{1/2} (T_{k,X} + \lambda I)^{-1/2} \right\| \left\| (T_{k,X} + \lambda I)^{1/2} (I - P_Z) f_0^* \right\|_{\mathbb{H}_k}.$$

Therefore, the norm of the sum of the third and last terms in (7) is bounded by

$$(1 + 2E + E \log t + \sqrt{6\eta t\lambda} E) \left\| T_{k,\rho_{\mathbf{x}}}^{1/2} (T_{k,X} + \lambda I)^{-1/2} \right\| \left\| (T_{k,X} + \lambda I)^{1/2} (I - P_Z) f_0^* \right\|_{\mathbb{H}_k}.$$

To bound $\left\| (T_{k,X} + \lambda I)^{1/2} (I - P_Z) f_0^* \right\|_{\mathbb{H}_k}$, observe that

$$\| (T_{k,X} + \lambda I)^{1/2} (I - P_Z) f_0^* \|_{\mathbb{H}_k} \leq \| (T_{k,X} + \lambda I)^{1/2} (T_{k,\rho_{\mathbf{x}}} + \lambda I)^{-1/2} \| \| (T_{k,\rho_{\mathbf{x}}} + \lambda I)^{1/2} (I - P_Z) f_0^* \|_{\mathbb{H}_k}$$

and

$$\| (T_{k,\rho_{\mathbf{x}}} + \lambda I)^{1/2} (I - P_Z) f_0^* \|_{\mathbb{H}_k}^2 = \| \iota_{\rho_{\mathbf{x}}} (I - P_Z) f_0^* \|_{L^2_{\rho_{\mathbf{x}}}}^2 + \lambda \| (I - P_Z) f_0^* \|_{\mathbb{H}_k}^2$$

$$\leq B \| \iota_{\tilde{\rho}_{\mathbf{x}}} (I - P_Z) f_0^* \|_{L^2_{\tilde{\rho}_{\mathbf{x}}}}^2 + \lambda \| (I - P_Z) f_0^* \|_{\mathbb{H}_k}^2$$

$$= B \| (T_{k,\tilde{\rho}_{\mathbf{x}}} + \lambda I)^{1/2} (I - P_Z) f_0^* \|_{\mathbb{H}_k}^2.$$

Under Assumption 3.3, we have

$$B^{1/2} \| (T_{k,\tilde{\rho}_{\mathbf{x}}} + \lambda I)^{1/2} (I - P_Z) f_0^* \|_{\mathbb{H}_k}$$

$$\leq B^{1/2} \| (T_{k,\tilde{\rho}_{\mathbf{x}}} + \lambda I)^{1/2} (I - P_Z) (T_{k,\tilde{\rho}_{\mathbf{x}}} + \lambda I)^{r-1/2} \| \| (T_{k,\tilde{\rho}_{\mathbf{x}}} + \lambda I)^{-(r-1/2)} T_{k,\rho_{\mathbf{x}}}^{r-1/2} \| \| g_0^* \|_{\mathbb{H}_k}.$$

Since

$$\| (T_{k,\tilde{\rho}_{\mathbf{x}}} + \lambda I)^{-(r-1/2)} T_{k,\rho_{\mathbf{x}}}^{r-1/2} \| \leq \| (T_{k,\tilde{\rho}_{\mathbf{x}}} + \lambda I)^{-1/2} T_{k,\rho_{\mathbf{x}}}^{1/2} \|^{2r-1} = \| T_{k,\rho_{\mathbf{x}}}^{1/2} (T_{k,\tilde{\rho}_{\mathbf{x}}} + \lambda I)^{-1/2} \|^{2r-1}$$

which follows from Lemma A.8 and

$$\| T_{k,\rho_{\mathbf{x}}}^{1/2} (T_{k,\tilde{\rho}_{\mathbf{x}}} + \lambda I)^{-1/2} \| = \| \iota_{\rho_{\mathbf{x}}} (T_{k,\tilde{\rho}_{\mathbf{x}}} + \lambda I)^{-1/2} \| \leq B^{1/2} \| \iota_{\tilde{\rho}_{\mathbf{x}}} (T_{k,\tilde{\rho}_{\mathbf{x}}} + \lambda I)^{-1/2} \|$$

$$\leq B^{1/2} \| T_{k,\tilde{\rho}_{\mathbf{x}}}^{1/2} (T_{k,\tilde{\rho}_{\mathbf{x}}} + \lambda I)^{-1/2} \| \leq B^{1/2},$$

we have $\|(T_{k,\tilde{\rho}_{\mathbf{x}}} + \lambda I)^{-(r-1/2)} T_{k,\rho_{\mathbf{x}}}^{r-1/2}\| \le B^{r-1/2}$. On the other hand,

$$\|(T_{k,\tilde{\rho}_{\mathbf{x}}} + \lambda I)^{1/2}(I - P_Z)(T_{k,\tilde{\rho}_{\mathbf{x}}} + \lambda I)^{r-1/2}\|$$
$$\le \|(T_{k,\tilde{\rho}_{\mathbf{x}}} + \lambda I)^{1/2}(I - P_Z)\|\|(I - P_Z)^{2r-1}(T_{k,\tilde{\rho}_{\mathbf{x}}} + \lambda I)^{r-1/2}\|$$
$$\le \|(T_{k,\tilde{\rho}_{\mathbf{x}}} + \lambda I)^{1/2}(I - P_Z)\|^{2r}$$

by Lemma A.8. Therefore,

$$\left\| T_{k,\rho_{\mathbf{x}}}^{1/2}(T_{k,X} + \lambda I)^{-1/2} \right\| \left\| (T_{k,X} + \lambda I)^{1/2}(I - P_Z)f_0^* \right\|_{\mathbb{H}_k}$$
$$\le R B^r \left\| (T_{k,\rho_{\mathbf{x}}} + \lambda I)^{1/2}(T_{k,X} + \lambda I)^{-1/2} \right\|$$
$$\cdot \|(T_{k,X} + \lambda I)^{1/2}(T_{k,\rho_{\mathbf{x}}} + \lambda I)^{-1/2}\|\|(T_{k,\tilde{\rho}_{\mathbf{x}}} + \lambda I)^{1/2}(I - P_Z)\|^{2r}.$$

Since $X$ and $Z$ are independent, we have

$$\mathbb{E}\left[ \left\| T_{k,\rho_{\mathbf{x}}}^{1/2}(T_{k,X} + \lambda I)^{-1/2} \right\| \left\| (T_{k,X} + \lambda I)^{1/2}(I - P_Z)f_0^* \right\|_{\mathbb{H}_k} \right]$$
$$\le R B^r \mathbb{E}\left[ \left\| (T_{k,\rho_{\mathbf{x}}} + \lambda I)^{1/2}(T_{k,X} + \lambda I)^{-1/2} \right\| \|(T_{k,X} + \lambda I)^{1/2}(T_{k,\rho_{\mathbf{x}}} + \lambda I)^{-1/2}\| \right]$$
$$\cdot \mathbb{E}\|(T_{k,\tilde{\rho}_{\mathbf{x}}} + \lambda I)^{1/2}(I - P_Z)\|^{2r}.$$

By Lemma A.8 and Lemma A.10,

$$\|(T_{k,\rho_{\mathbf{x}}} + \lambda I)^{1/2}(T_{k,X} + \lambda I)^{-1/2}\| \le \left( 2 + 2\left( \left( \frac{2\kappa^2}{n\lambda} + \sqrt{\frac{4\kappa^2 \mathcal{N}(\lambda)}{n\lambda}} \right) \log(2/\delta) \right)^2 \right)^{1/2}$$

holds with confidence at least $1 - \delta$ where $\delta \in (0,1)$. Also, by Lemma A.8 and Lemma A.11

$$\|(T_{k,X} + \lambda I)^{1/2}(T_{k,\rho_{\mathbf{x}}} + \lambda I)^{-1/2}\| \le \left( 1 + \left( \frac{2\kappa^2}{n\lambda} + \sqrt{\frac{4\kappa^2 \mathcal{N}(\lambda)}{n\lambda}} \right) \log(2/\delta) \right)^{1/2}$$

holds with confidence at least $1 - \delta$ where $\delta \in (0,1)$. Thus,

$$\|(T_{k,\rho_{\mathbf{x}}} + \lambda I)^{1/2}(T_{k,X} + \lambda I)^{-1/2}\|\|(T_{k,X} + \lambda I)^{1/2}(T_{k,\rho_{\mathbf{x}}} + \lambda I)^{-1/2}\|$$
$$\le \left( 2 + 2\left( \frac{2\kappa^2}{n\lambda} + \sqrt{\frac{4\kappa^2 \mathcal{N}(\lambda)}{n\lambda}} \right)^2 \right)^{1/2} \left( 1 + \left( \frac{2\kappa^2}{n\lambda} + \sqrt{\frac{4\kappa^2 \mathcal{N}(\lambda)}{n\lambda}} \right) \right)^{1/2} (\log(4/\delta))^{3/2}$$

with confidence at least $1 - \delta$ where $\delta \in (0,1)$. Set $\lambda = 128(\kappa^2 + 1)^2 n^{-\frac{1}{2r+s}}$ where $n$ is sufficiently large such that $\lambda \le 1$ and $\mathcal{N}_{\tilde{\rho}_{\mathbf{x}}}(\lambda) \ge 1$. Then

$$\mathbb{E}\left[ \left\| (T_{k,\rho_{\mathbf{x}}} + \lambda I)^{1/2}(T_{k,X} + \lambda I)^{-1/2} \right\| \|(T_{k,X} + \lambda I)^{1/2}(T_{k,\rho_{\mathbf{x}}} + \lambda I)^{-1/2}\| \right]$$
$$\le 4\Gamma(2.5)(2 + 2(2\kappa^2 + 2\kappa\sqrt{C_s'})^2)^{1/2}(1 + 2\kappa^2 + 2\kappa\sqrt{C_s'})^{1/2} \lesssim 1.$$

We now bound $\mathbb{E}\|(T_{k,\tilde{\rho}_{\mathbf{x}}} + \lambda I)^{1/2}(I - P_Z)\|^{2r}$. By Lemma A.9, we have

$$\|(T_{k,\tilde{\rho}_{\mathbf{x}}} + \lambda I)^{1/2}(I - P_Z)\|^{2r} \le \lambda^r \|(T_{k,\tilde{\rho}_{\mathbf{x}}} + \lambda I)^{1/2}(T_{k,Z} + \lambda I)^{-1/2}\|^{2r}.$$

By Lemma A.12, $\|(T_{k,\tilde{\rho}_{\mathbf{x}}} + \lambda I)^{1/2}(T_{k,Z} + \lambda I)^{-1/2}\| \le \sqrt{2}$ with confidence at least $1 - 4\exp(-1/4(\kappa^2 + 1)\mathcal{B}_0)$ where

$$\mathcal{B}_0 = \frac{1 + \log \mathcal{N}_{\tilde{\rho}_{\mathbf{x}}}(\lambda)}{\lambda n_0} + \sqrt{\frac{1 + \log \mathcal{N}_{\tilde{\rho}_{\mathbf{x}}}(\lambda)}{\lambda n_0}}.$$

Also, $\|(T_{k,\tilde{\rho}_{\mathbf{x}}} + \lambda I)^{1/2}(I - P_Z)\| \le (\kappa^2 + 1)^{1/2}$ almost surely. Thus,

$$\mathbb{E}\|(T_{k,\tilde{\rho}_{\mathbf{x}}} + \lambda I)^{1/2}(I - P_Z)\|^{2r} \le 2^r \lambda^r + (\kappa^2 + 1)^r \cdot 4\exp\left( -\frac{1}{4(\kappa^2 + 1)\mathcal{B}_0} \right).$$

Note that

$$\mathcal{B}_0 \le \frac{\log \kappa^2 e + \log(1/\lambda)}{\lambda n_0} + \sqrt{\frac{\log \kappa^2 e + \log(1/\lambda)}{\lambda n_0}} \le \frac{1}{4(\kappa^2 + 1)\log n}$$

and so $(\kappa^2 + 1)^r \cdot 4 \exp\left(-\frac{1}{4(\kappa^2+1)\mathcal{B}_0}\right) \leq 4(\kappa^2 + 1)^r \cdot \frac{1}{n}$. Therefore,

$$\mathbb{E}\|(T_{k,\tilde{\rho}_\mathbf{x}} + \lambda I)^{1/2}(I - P_Z)\|^{2r} \leq 2^r 128^r (\kappa^2 + 1)^{2r} \cdot n^{-\frac{r}{2r+s}} + 4(\kappa^2 + 1)^r \cdot n^{-1} \lesssim n^{-\frac{r}{2r+s}}.$$

We can conclude that

$$\mathbb{E}\left\|-\iota_{\rho_\mathbf{x}}(I - P_Z)f_0^* + \iota_{\rho_\mathbf{x}}\left(I + \cdots + \left(\sum_{i=1}^m \frac{n_i}{n} P_Z \overline{T}_{k,X_i}^E\right)^{t-2}\right) P_Z \left(I - \sum_{i=1}^m \frac{n_i}{n} \overline{T}_{k,X_i}^E\right)(I - P_Z)f_0^*\right\|_{L_{\rho_\mathbf{x}}^2}$$

$$\leq (1 + 2E + E\log t + \sqrt{6\eta t\lambda} E)RB^r \cdot 4\Gamma(2.5)(2 + 2(2\kappa^2 + 2\kappa\sqrt{C_s'})^2)^{1/2}(1 + 2\kappa^2 + 2\kappa\sqrt{C_s'})^{1/2}$$

$$\cdot \left(2^r 128^r (\kappa^2 + 1)^{2r} \cdot n^{-\frac{r}{2r+s}} + 4(\kappa^2 + 1)^r \cdot n^{-1}\right)$$

$$\lesssim B^r (1 + \log t + t^{1/2} n^{-\frac{1/2}{2r+s}}) n^{-\frac{r}{2r+s}}.$$

### A.2.5 Stopping Rule and Rademacher Complexity Bound

For convenience, we abuse the notation $D = \{(\mathbf{x}^1, y^1), \cdots, (\mathbf{x}^n, y^n)\}$ and $X = \{\mathbf{x}^1, \cdots, \mathbf{x}^n\}$. Define the local empirical Rademacher complexity

$$Q_n(\epsilon) = \mathbb{E}\left[\sup_{\|g\|_{\mathbb{H}_k}\leq 1, \|g\|_{L_{\rho_{\mathbf{x},n}}^2}\leq\epsilon}\left|\frac{1}{n}\sum_{i=1}^n w_i g(\mathbf{x}^i)\right| \,\Big|\, X\right]$$

and the local population Rademacher complexity

$$\overline{Q}_n(\epsilon) = \mathbb{E}\left[\sup_{\|g\|_{\mathbb{H}_k}\leq 1, \|g\|_{L_{\rho_\mathbf{x}}^2}\leq\epsilon}\left|\frac{1}{n}\sum_{i=1}^n w_i g(\mathbf{x}^i)\right|\right]$$

where $w_1, \cdots, w_n$ are independent Rademacher random variables and $\rho_{\mathbf{x},n} = \frac{1}{n}\sum_{i=1}^n \delta_{\mathbf{x}^i}$. We also define

$$\overline{\mathcal{R}}(\epsilon) = \sqrt{\frac{1}{n}\sum_{i=1}^\infty \lambda_i \wedge \epsilon^2}$$

where $\lambda_1 \geq \lambda_2 \geq \cdots \geq 0$ are eigenvalues of $T_{k,\rho_\mathbf{x}}$. We recall the following well-known property.

**Lemma A.1** ([46], [60]). *We have*

$$\overline{Q}_n(\epsilon) \leq \sqrt{2} \cdot \overline{\mathcal{R}}(\epsilon)$$

*for $\epsilon > 0$.*

We can prove the following lemma using a similar argument as in [46].

**Lemma A.2.** *There is an absolute constant $c > 0$ which satisfies that for every $\epsilon > 0$,*

$$c \cdot \overline{\mathcal{R}}(\epsilon) \leq Q_n(\epsilon).$$

*Proof of Lemma A.2.* We divide the proof into three parts.

**Part 1.** Since $T_{k,X} = S_D^\top S_D$, $\hat{\lambda}_1 \geq \hat{\lambda}_2 \geq \cdots \geq \hat{\lambda}_n \geq 0$ are eigenvalues of $T_{k,X}$. For convenience, set $\hat{\lambda}_i = 0$ for $i > n$ and define $\hat{n} \leq n$ such that $\hat{\lambda}_{\hat{n}} > 0$ and $\hat{\lambda}_{\hat{n}+1} = 0$. Choose an orthonormal basis $\{\hat{\psi}_i\}_{i=1}^\infty$ of $\mathbb{H}_k$ such that $\hat{\psi}_i$ is an eigenvector of $T_{k,X}$ corresponding to $\hat{\lambda}_i$. Then $\langle \hat{\psi}_i, \hat{\psi}_j\rangle_{L_{\rho_{\mathbf{x},n}}^2} = \langle S_D\hat{\psi}_i, S_D\hat{\psi}_j\rangle_2 = \langle T_{k,X}\hat{\psi}_i, \hat{\psi}_j\rangle_{\mathbb{H}_k} = \delta_{\{i=j\}}\hat{\lambda}_i$. We will show that

$$k_\mathbf{x} = \sum_{i=1}^{\hat{n}} \hat{\psi}_i(\mathbf{x})\hat{\psi}_i$$

where $\mathbf{x} \in \{\mathbf{x}^1, \cdots, \mathbf{x}^n\}$. Let $W_1$ be the subspace of $\mathbb{H}_k$ spanned by $\{\hat{\psi}_i : i = 1, \cdots, \hat{n}\}$ and $W_2$ be the subspace of $\mathbb{H}_k$ spanned by $\{k_{\mathbf{x}^i} : i = 1, \cdots, n\}$. Observe that $W_1^\perp = \ker T_{k,X}$ and $W_2^\perp \subset \ker T_{k,X}$ by the reproducing property. Thus, $W_1 \subset W_2$. Conversely, choose a basis $\{k_{\tilde{\mathbf{x}}^i} : i = 1, \cdots, \tilde{n}'\} \subset \{k_{\mathbf{x}^i} : i = 1, \cdots, n\}$ of $W_2$. Then, using a similar argument as in Appendix A.2.1 implies that there exists a matrix

$$B = \begin{bmatrix} b_{11} & \cdots & b_{1\tilde{n}'} \\ \vdots & \ddots & \vdots \\ b_{n1} & \cdots & b_{n\tilde{n}'} \end{bmatrix} \in \mathbb{R}^{n\times\tilde{n}'}$$

such that $k_{\mathbf{x}^i} = \sum_{j=1}^{\tilde{n}'} b_{ij} k_{\tilde{\mathbf{x}}^j}$. Then $K_{X\tilde{X}} = B K_{\tilde{X}\tilde{X}}$ where

$$K_{X\tilde{X}} = \begin{bmatrix} k(\mathbf{x}^1, \tilde{\mathbf{x}}^1) & \cdots & k(\mathbf{x}^1, \tilde{\mathbf{x}}^{\tilde{n}'}) \\ \vdots & \ddots & \vdots \\ k(\mathbf{x}^n, \tilde{\mathbf{x}}^1) & \cdots & k(\mathbf{x}^n, \tilde{\mathbf{x}}^{\tilde{n}'}) \end{bmatrix} \in \mathbb{R}^{n \times \tilde{n}'}$$

and

$$K_{\tilde{X}\tilde{X}} = \begin{bmatrix} k(\tilde{\mathbf{x}}^1, \tilde{\mathbf{x}}^1) & \cdots & k(\tilde{\mathbf{x}}^1, \tilde{\mathbf{x}}^{\tilde{n}'}) \\ \vdots & \ddots & \vdots \\ k(\tilde{\mathbf{x}}^{\tilde{n}'}, \tilde{\mathbf{x}}^1) & \cdots & k(\tilde{\mathbf{x}}^{\tilde{n}'}, \tilde{\mathbf{x}}^{\tilde{n}'}) \end{bmatrix} \in \mathbb{R}^{\tilde{n}' \times \tilde{n}'}.$$

Since $K_{\tilde{X},\tilde{X}}$ and $B^\top B$ are invertible,

$$T_{k,X} \left( \sum_{i=1}^{\tilde{n}'} [n K_{\tilde{X},\tilde{X}}^{-1} (B^\top B)^{-1} \mathbf{b}]_i k_{\tilde{\mathbf{x}}^i} \right) = \sum_{i=1}^{\tilde{n}'} \mathbf{b}_i k_{\tilde{\mathbf{x}}^i}$$

for any $\mathbf{b} = [\mathbf{b}_1, \cdots, \mathbf{b}_{\tilde{n}'}]^\top \in \mathbb{R}^{\tilde{n}'}$ where $[\cdot]_r$ is the $r$th component of the given vector. Therefore, $W_2 \subset \operatorname{ran} T_{k,X} = (\ker T_{k,X})^\perp = W_1$ and so $W_1 = W_2$. From this fact, we can see that $k_{\mathbf{x}^i} = \sum_{r=1}^{\hat{n}} a_r \hat{\psi}_r$ for some $a_1, \cdots, a_{\hat{n}} \in \mathbb{R}$. Then $a_r = \langle k_{\mathbf{x}^i}, \hat{\psi}_r \rangle_{\mathbb{H}_k} = \hat{\psi}_r(\mathbf{x}^i)$ for all $r = 1, \cdots, \hat{n}$ and so we are done. Note that $k_{\mathbf{x}} = \sum_{i=1}^{\infty} \hat{\psi}_i(\mathbf{x}) \hat{\psi}_i$ where $\mathbf{x} \in \{\mathbf{x}^1, \cdots, \mathbf{x}^n\}$ since $\hat{\psi}_i(\mathbf{x}) = 0$ for $\mathbf{x} \in \{\mathbf{x}^1, \cdots, \mathbf{x}^n\}$ and $i > \hat{n}$.

**Part 2.** Define

$$\mathcal{F} := \left\{ h : \|h\|_{\mathbb{H}_k} \leq 1 \text{ and } \|h\|_{L^2_{\hat{\rho}_{\mathbf{x},n}}} \leq \epsilon \right\} = \left\{ \sum_{i=1}^{\infty} h_i \hat{\psi}_i : \sum_{i=1}^{\infty} h_i^2 \leq 1 \text{ and } \sum_{i=1}^{\hat{n}} \hat{\lambda}_i h_i^2 \leq \epsilon^2 \right\}$$

and

$$\mathcal{E} := \left\{ \sum_{i=1}^{\infty} h_i \hat{\psi}_i : \sum_{i=1}^{\infty} \frac{\hat{\lambda}_i}{\hat{\lambda}_i \wedge \epsilon^2} h_i^2 \leq 1 \right\}$$

where $\frac{0}{0} = 1$. Then $\mathcal{E} \subset \mathcal{F}$ since

$$\left( \sum_{i=1}^{\infty} h_i^2 \right) \vee \left( \sum_{i=1}^{\infty} \frac{\hat{\lambda}_i}{\epsilon^2} h_i^2 \right) \leq \sum_{i=1}^{\infty} \left( 1 \vee \frac{\hat{\lambda}_i}{\epsilon^2} \right) h_i^2 = \sum_{i=1}^{\infty} \frac{\hat{\lambda}_i}{\hat{\lambda}_i \wedge \epsilon^2} h_i^2 \leq 1$$

for $h = \sum_{i=1}^{\infty} h_i \hat{\psi}_i \in \mathcal{E}$. Thus,

$$\mathbb{E}\left[ \sup_{h \in \mathcal{E}} \left| \sum_{i=1}^n \epsilon_i h(\mathbf{x}^i) \right|^2 \,\Big|\, X \right] \leq \mathbb{E}\left[ \sup_{h \in \mathcal{F}} \left| \sum_{i=1}^n \epsilon_i h(\mathbf{x}^i) \right|^2 \,\Big|\, X \right]$$

where $\epsilon_1, \cdots, \epsilon_n$ are i.i.d. Rademacher variables. By the reproducing property,

$$\sum_{i=1}^n \epsilon_i h(\mathbf{x}^i) = \langle h, \sum_{i=1}^n \epsilon_i k_{\mathbf{x}^i} \rangle_{\mathbb{H}_k} = \langle h, \sum_{i=1}^n \epsilon_i \sum_{j=1}^{\hat{n}} \hat{\psi}_j(\mathbf{x}^i) \hat{\psi}_j \rangle_{\mathbb{H}_k}$$

$$= \sum_{j=1}^{\hat{n}} h_j \sum_{i=1}^n \epsilon_i \hat{\psi}_j(\mathbf{x}^i) = \langle \sum_{j=1}^{\infty} \sqrt{\frac{\hat{\lambda}_j}{\hat{\lambda}_j \wedge \epsilon^2}} h_j \hat{\psi}_j, \sum_{j=1}^{\hat{n}} \sqrt{\frac{\hat{\lambda}_j \wedge \epsilon^2}{\hat{\lambda}_j}} \sum_{i=1}^n \epsilon_i \hat{\psi}_j(\mathbf{x}^i) \hat{\psi}_j \rangle_{\mathbb{H}_k}$$

where $h = \sum_{i=1}^{\infty} h_i \hat{\psi}_i$. Thus,

$$\sup_{h \in \mathcal{E}} \left| \sum_{i=1}^n \epsilon_i h(\mathbf{x}^i) \right|^2 = \left\| \sum_{j=1}^{\hat{n}} \sqrt{\frac{\hat{\lambda}_j \wedge \epsilon^2}{\hat{\lambda}_j}} \sum_{i=1}^n \epsilon_i \hat{\psi}_j(\mathbf{x}^i) \hat{\psi}_j \right\|_{\mathbb{H}_k}^2.$$

Since

$$\left\| \sum_{j=1}^{\hat{n}} \sqrt{\frac{\hat{\lambda}_j \wedge \epsilon^2}{\hat{\lambda}_j}} \sum_{i=1}^n \epsilon_i \hat{\psi}_j(\mathbf{x}^i) \hat{\psi}_j \right\|_{\mathbb{H}_k}^2 = \sum_{j=1}^{\hat{n}} \frac{\hat{\lambda}_j \wedge \epsilon^2}{\hat{\lambda}_j} \left( \sum_{i=1}^n \epsilon_i \hat{\psi}_j(\mathbf{x}^i) \right)^2,$$

we have

$$\mathbb{E}\left[\sup_{h\in\mathcal{E}}\left|\sum_{i=1}^{n}\epsilon_i h(\mathbf{x}^i)\right|^2 \,\bigg|\, X\right] = \mathbb{E}\left[\sum_{j=1}^{\hat{n}}\frac{\hat{\lambda}_j\wedge\epsilon^2}{\hat{\lambda}_j}\left(\sum_{i=1}^{n}\epsilon_i\hat{\psi}_j(\mathbf{x}^i)\right)^2 \,\bigg|\, X\right]$$

$$= \sum_{j=1}^{\hat{n}}\frac{\hat{\lambda}_j\wedge\epsilon^2}{\hat{\lambda}_j}\left(\sum_{i=1}^{n}\hat{\psi}_j(\mathbf{x}^i)^2\right) = n\sum_{j=1}^{n}\hat{\lambda}_j\wedge\epsilon^2.$$

Therefore,

$$\sqrt{n\sum_{j=1}^{n}\hat{\lambda}_j\wedge\epsilon^2} \le \mathbb{E}\left[\sup_{h\in\mathcal{F}}\left|\sum_{i=1}^{n}\epsilon_i h(\mathbf{x}^i)\right|^2 \,\bigg|\, X\right]^{1/2}.$$

**Part 3.** By Khintchine's inequality,

$$\mathbb{E}\left[\sup_{h\in\mathcal{F}}\left|\sum_{i=1}^{n}\epsilon_i h(\mathbf{x}^i)\right| \,\bigg|\, X\right] \ge \sup_{h\in\mathcal{F}}\mathbb{E}\left[\left|\sum_{i=1}^{n}\epsilon_i h(\mathbf{x}^i)\right| \,\bigg|\, X\right] \ge \frac{1}{\sqrt{2}}\sup_{h\in\mathcal{F}}\left(\sum_{i=1}^{n}h(\mathbf{x}^i)^2\right)^{1/2} = \sqrt{\frac{n}{2}}\epsilon.$$

Set $Z = g(\epsilon_1,\cdots,\epsilon_n)$ where

$$g(t_1,\cdots,t_n) = \sup_{h\in\mathcal{F}}\left|\sum_{i=1}^{n}t_i h(\mathbf{x}^i)\right|.$$

By Remark A.14, $g$ is convex and $\sup_{h\in\mathcal{F}}(\sum_{i=1}^{n}h(\mathbf{x}^i)^2)^{1/2} = \sqrt{n}\epsilon$-Lipschitz on $[-1,1]^n$. By Lemma A.13, we have

$$\mathbb{P}\left(Z - \mathbb{E}\left[Z|X\right] \ge t\mathbb{E}\left[Z|X\right]|X\right) \le \exp\left(-\frac{t^2}{16\epsilon^2 n}\mathbb{E}\left[Z|X\right]^2\right) \le \exp\left(-\frac{t^2}{32}\right).$$

From

$$\mathbb{E}\left[Z^2|X\right] = \mathbb{E}\left[Z^2\mathbf{1}_{\{Z<\mathbb{E}[Z|X]\}}|X\right] + \sum_{m=0}^{\infty}\mathbb{E}\left[Z^2\mathbf{1}_{\{(m+1)\mathbb{E}[Z|X]\le Z<(m+2)\mathbb{E}[Z|X]\}}|X\right]$$

$$\le \mathbb{E}[Z|X]^2 + \sum_{m=0}^{\infty}(m+2)^2\mathbb{E}[Z|X]^2\mathbb{P}(Z\ge(m+1)\mathbb{E}[Z|X]|X)$$

$$\le \mathbb{E}[Z|X]^2\left(1 + \sum_{m=0}^{\infty}(m+2)^2\exp\left(-\frac{m^2}{32}\right)\right),$$

we have

$$\mathbb{E}[Z|X] \ge c\cdot\mathbb{E}\left[Z^2|X\right]^{1/2} \ge c\cdot\sqrt{n\sum_{j=1}^{n}\hat{\lambda}_j\wedge\epsilon^2}$$

where $c = \left(1 + \sum_{m=0}^{\infty}(m+2)^2\exp\left(-\frac{m^2}{32}\right)\right)^{-1/2}$ is an absolute constant. Therefore,

$$c\cdot\sqrt{\frac{1}{n}\sum_{j=1}^{n}\hat{\lambda}_j\wedge\epsilon^2} \le \mathbb{E}\left[\sup_{h\in\mathcal{F}}\left|\frac{1}{n}\sum_{i=1}^{n}\epsilon_i h(\mathbf{x}^i)\right| \,\bigg|\, X\right].$$

$\square$

We set the population radius as

$$\epsilon_n = \inf\left\{\epsilon \ge 0 : \overline{Q}_n(\epsilon) \le \frac{\epsilon^{1+2r}}{16\kappa}\right\}.$$

We also define

$$\tilde{\epsilon}_n = \inf\left\{\epsilon \ge 0 : \overline{\mathcal{R}}(\epsilon) \le \frac{\epsilon^{1+2r}}{16\sqrt{2}\kappa}\right\}.$$

By Lemma A.1, we have $\epsilon_n \le \tilde{\epsilon}_n$. We can easily see that $\mathcal{R}, \overline{\mathcal{R}}, Q_n$, and $\overline{Q}_n$ are increasing functions. The following lemma can be shown by a similar argument as in [2].

**Lemma A.3.** *If $g : [0, \infty) \to [0, \infty)$ is a function such that $g$ is non-decreasing and $r \mapsto g(r)/r$ is non-increasing, then $g$ is continuous on $(0, \infty)$.*

Since

$$\frac{\mathcal{R}(\epsilon)}{\epsilon} = \sqrt{\frac{1}{n} \sum_{i=1}^{n} 1 \wedge \frac{\hat{\lambda}_i}{\epsilon^2}} \quad \text{and} \quad \frac{\overline{\mathcal{R}}(\epsilon)}{\epsilon} = \sqrt{\frac{1}{n} \sum_{i=1}^{\infty} 1 \wedge \frac{\lambda_i}{\epsilon^2}},$$

$\epsilon \mapsto \mathcal{R}(\epsilon)/\epsilon$ and $\epsilon \mapsto \overline{\mathcal{R}}(\epsilon)/\epsilon$ are non-increasing and so $\mathcal{R}$ and $\overline{\mathcal{R}}$ are continuous.

**Lemma A.4.** $\epsilon \mapsto Q_n(\epsilon)/\epsilon$ and $\epsilon \mapsto \overline{Q}_n(\epsilon)/\epsilon$ are non-increasing. In particular, $\overline{Q}_n$ is continuous and $\epsilon_n < \infty$.

*Proof.* From the fact that

$$\frac{Q_n(\epsilon)}{\epsilon} = \frac{1}{\epsilon} \mathbb{E}\left[ \sup_{\|g\|_{\mathbb{H}_k} \leq 1, \|g\|_{L^2_{\rho_{\mathbf{x},n}}} \leq \epsilon} \left| \frac{1}{n} \sum_{i=1}^{n} w_i g(\mathbf{x}_i) \right| \, \Bigg| \, X \right]$$

$$= \mathbb{E}\left[ \sup_{\|g\|_{\mathbb{H}_k} \leq 1/\epsilon, \|g\|_{L^2_{\rho_{\mathbf{x},n}}} \leq 1} \left| \frac{1}{n} \sum_{i=1}^{n} w_i g(\mathbf{x}_i) \right| \, \Bigg| \, X \right],$$

we can easily see that $\epsilon \mapsto Q_n(\epsilon)/\epsilon$ is non-increasing. Similarly, we can show that $\epsilon \mapsto \overline{Q}_n(\epsilon)/\epsilon$ is non-increasing. Note that

$$\lim_{\epsilon \to 0^+} \frac{Q_n(\epsilon)}{\epsilon} > 0 \quad \text{and} \quad \lim_{\epsilon \to 0^+} \frac{\overline{Q}_n(\epsilon)}{\epsilon} > 0.$$

Also, we can observe that

$$\lim_{\epsilon \to \infty} \frac{Q_n(\epsilon)}{\epsilon} = 0 \quad \text{and} \quad \lim_{\epsilon \to \infty} \frac{\overline{Q}_n(\epsilon)}{\epsilon} = 0.$$

Since $\epsilon \mapsto \epsilon^{2r}/16\kappa$ is increasing, goes $0$ as $\epsilon \to 0^+$, and goes $\infty$ as $\epsilon \to \infty$, we can conclude that $\epsilon_n < \infty$. $\square$

Similarly, we have $\tilde{\epsilon}_n < \infty$. In fact, we can find the lower and the upper bound of $\tilde{\epsilon}_n$ under Assumption 3.2.

**Lemma A.5.** *We have*

$$\left( 2^{9/(4r+2s)} \kappa^{1/(2r+s)} c_s^{s/(4r+2s)} \wedge c_s^{1/2} \left( \frac{s}{s+2} \right)^{1/2s} \right) n^{-\frac{1}{4r+2s}} \leq \tilde{\epsilon}_n$$

$$\leq 2^{9/(4r+2s)} \kappa^{1/(2r+s)} \left( \frac{2-s}{1-s} \right)^{1/(4r+2s)} C_s^{s/(4r+2s)} n^{-\frac{1}{4r+2s}}.$$

*Proof.* Since $c_s i^{-1/s} \leq \lambda_i \leq C_s i^{-1/s}$, we have

$$\sqrt{\frac{1}{n} \sum_{j=1}^{\infty} (c_s j^{-1/s}) \wedge \epsilon^2} \leq \overline{\mathcal{R}}(\epsilon) \leq \sqrt{\frac{1}{n} \sum_{j=1}^{\infty} (C_s j^{-1/s}) \wedge \epsilon^2}.$$

We first consider the lower bound of $\tilde{\epsilon}_n$. We first observe that

$$\sqrt{\frac{1}{n} \sum_{j=1}^{\infty} (c_s j^{-1/s}) \wedge \epsilon^2} = \sqrt{\frac{1}{n} \left( \left\lfloor \left( \frac{c_s}{\epsilon^2} \right)^s \right\rfloor \epsilon^2 + \sum_{j=\left\lfloor \left( \frac{c_s}{\epsilon^2} \right)^s \right\rfloor + 1}^{\infty} c_s j^{-1/s} \right)}.$$

Set

$$\epsilon = \left( 2^{9/(4r+2s)} \kappa^{1/(2r+s)} c_s^{s/(4r+2s)} \wedge c_s^{1/2} \left( \frac{s}{s+2} \right)^{1/2s} \right) n^{-\frac{1}{4r+2s}}.$$

Note that

$$c_s \left\lfloor \left( \frac{c_s}{\epsilon^2} \right)^s \right\rfloor^{-1/s} \geq \epsilon^2 \quad \text{and} \quad \frac{s}{1-s} \left( \left\lfloor \left( \frac{c_s}{\epsilon^2} \right)^s \right\rfloor + 1 \right)^{(s-1)/s} \geq \left\lfloor \left( \frac{c_s}{\epsilon^2} \right)^s \right\rfloor^{-1/s}$$

hold. The first formula is trivial. To show the second formula, we observe that the function $u(t) = \left(\frac{s}{1-s}\right)^s \cdot \frac{t}{(1+t)^{1-s}}$ is an increasing function. Thus, for $t \geq 2/s$

$$u(t) \geq u\left(\frac{2}{s}\right) = \frac{2}{(1-s)^s(s+2)^{1-s}} \geq \frac{2}{2-2s^2} \geq 1.$$

Here, we apply an elementary inequality: $a^s b^{1-s} \leq sa + (1-s)b \quad \forall a, b > 0$. Since

$$\epsilon \leq c_s^{1/2}\left(\frac{s}{s+2}\right)^{1/2s} \quad \Rightarrow \quad \left\lfloor \left(\frac{c_s}{\epsilon^2}\right)^s \right\rfloor \geq \left(\frac{c_s}{\epsilon^2}\right)^s - 1 \geq \frac{2}{s},$$

putting $t = \left\lfloor \left(\frac{c_s}{\epsilon^2}\right)^s \right\rfloor$ gives

$$\left(\frac{s}{1-s}\right)^s \cdot \frac{\left\lfloor \left(\frac{c_s}{\epsilon^2}\right)^s \right\rfloor}{(1 + \left\lfloor \left(\frac{c_s}{\epsilon^2}\right)^s \right\rfloor)^{1-s}} \geq 1$$

and so the second formula holds. Therefore,

$$\sum_{j=\left\lfloor \left(\frac{c_s}{\epsilon^2}\right)^s \right\rfloor + 1}^{\infty} c_s j^{-1/s} \geq c_s \int_{\left\lfloor \left(\frac{c_s}{\epsilon^2}\right)^s \right\rfloor + 1}^{\infty} \frac{1}{t^{1/s}}\, dt = \frac{sc_s}{1-s}\left(\left\lfloor \left(\frac{c_s}{\epsilon^2}\right)^s \right\rfloor + 1\right)^{(s-1)/s} \geq c_s \left\lfloor \left(\frac{c_s}{\epsilon^2}\right)^s \right\rfloor^{-1/s} \geq \epsilon^2$$

holds and so we have

$$\sqrt{\frac{1}{n}\left(\left\lfloor \left(\frac{c_s}{\epsilon^2}\right)^s \right\rfloor \epsilon^2 + \sum_{j=\left\lfloor \left(\frac{c_s}{\epsilon^2}\right)^s \right\rfloor + 1}^{\infty} c_s j^{-1/s}\right)} \geq \sqrt{\frac{1}{n}\left(\left\lfloor \left(\frac{c_s}{\epsilon^2}\right)^s \right\rfloor + 1\right)\epsilon^2} \geq \frac{c_s^{s/2}}{n^{1/2}}\epsilon^{1-s} \geq \frac{\epsilon^{1+2r}}{16\sqrt{2}\kappa}$$

where the last inequality follows from $\epsilon \leq 2^{9/(4r+2s)}\kappa^{1/(2r+s)}c_s^{s/(4r+2s)}n^{-\frac{1}{4r+2s}}$. We can conclude that

$$\tilde{\epsilon}_n \geq \left(2^{9/(4r+2s)}\kappa^{1/(2r+s)}c_s^{s/(4r+2s)} \wedge c_s^{1/2}\left(\frac{s}{s+2}\right)^{1/2s}\right)n^{-\frac{1}{4r+2s}}$$

by Lemma A.4. We now derive the upper bound of $\tilde{\epsilon}_n$. Note that

$$\sqrt{\frac{1}{n}\sum_{j=1}^{\infty}(C_s j^{-1/s}) \wedge \epsilon^2} = \sqrt{\frac{1}{n}\left(\left\lfloor \left(\frac{C_s}{\epsilon^2}\right)^s \right\rfloor \epsilon^2 + \sum_{j=\left\lfloor \left(\frac{C_s}{\epsilon^2}\right)^s \right\rfloor + 1}^{\infty} C_s j^{-1/s}\right)}.$$

Since

$$\sum_{j=\left\lfloor \left(\frac{C_s}{\epsilon^2}\right)^s \right\rfloor + 1}^{\infty} C_s j^{-1/s} - \int_{\left\lfloor \left(\frac{C_s}{\epsilon^2}\right)^s \right\rfloor + 1}^{\infty} \frac{C_s}{t^{1/s}}\, dt \leq C_s \left(\left\lfloor \left(\frac{C_s}{\epsilon^2}\right)^s \right\rfloor + 1\right)^{-1/s}$$

and

$$\int_{\left\lfloor \left(\frac{C_s}{\epsilon^2}\right)^s \right\rfloor + 1}^{\infty} \frac{C_s}{t^{1/s}}\, dt = \frac{sC_s}{1-s}\left(\left\lfloor \left(\frac{C_s}{\epsilon^2}\right)^s \right\rfloor + 1\right)^{1-1/s} \geq \frac{sC_s}{1-s}\left(\left\lfloor \left(\frac{C_s}{\epsilon^2}\right)^s \right\rfloor + 1\right)^{-1/s},$$

we have

$$\sum_{j=\left\lfloor \left(\frac{C_s}{\epsilon^2}\right)^s \right\rfloor + 1}^{\infty} C_s j^{-1/s} \leq \frac{1}{s}\int_{\left(\frac{C_s}{\epsilon^2}\right)^s}^{\infty} \frac{C_s}{t^{1/s}}\, dt = \frac{C_s^s}{1-s}\epsilon^{2-2s}.$$

Hence, we have

$$\sqrt{\frac{1}{n}\left(\left\lfloor \left(\frac{C_s}{\epsilon^2}\right)^s \right\rfloor \epsilon^2 + \sum_{j=\left\lfloor \left(\frac{C_s}{\epsilon^2}\right)^s \right\rfloor + 1}^{\infty} C_s j^{-1/s}\right)} \leq \sqrt{\frac{1}{n}\left(C_s^s + \frac{C_s^s}{1-s}\right)\epsilon^{2-2s}}.$$

Set

$$\epsilon = 2^{9/(4r+2s)}\kappa^{1/(2r+s)}\left(\frac{2-s}{1-s}\right)^{1/(4r+2s)}C_s^{s/(4r+2s)}n^{-\frac{1}{4r+2s}}$$

which is equivalent to $\sqrt{\frac{1}{n}\left(C_s^s + \frac{C_s^s}{1-s}\right)\epsilon^{2-2s}} = \frac{\epsilon^{1+2r}}{16\sqrt{2}\kappa}$. Therefore, we attain the upper bound of $\tilde{\epsilon}_n$:

$$\tilde{\epsilon}_n \leq 2^{9/(4r+2s)}\kappa^{1/(2r+s)}\left(\frac{2-s}{1-s}\right)^{1/(4r+2s)}C_s^{s/(4r+2s)}n^{-\frac{1}{4r+2s}}.$$

$\square$

Without loss of generality, we assume $n$ is sufficiently large such that

$$n \geq 2^9 \kappa^2 \left(\frac{2-s}{1-s}\right) C_s^s \Leftrightarrow 2^{9/(4r+2s)} \kappa^{1/(2r+s)} \left(\frac{2-s}{1-s}\right)^{1/(4r+2s)} C_s^{s/(4r+2s)} n^{-\frac{1}{4r+2s}} \leq 1.$$

Then $\epsilon_n \leq \tilde{\epsilon}_n \leq 1$. We now prove the following lemma. It is an extended version of Theorem 14.1 in [60].

**Lemma A.6.** *We have*

$$\mathbb{P}\left(\sup_{\|h\|_{\mathbb{H}_k} \leq 1} \frac{\left|\|h\|_{L^2_{\rho_{\mathbf{x}},n}}^2 - \|h\|_{L^2_{\rho_{\mathbf{x}}}}^2\right|}{\|h\|_{L^2_{\rho_{\mathbf{x}}}}^2 + t^2} \leq \frac{1}{2}\right) \geq 1 - \exp\left(-c_1 n t^{4r}\right)$$

*for any $t \in [\epsilon_n, 1]$ where $c_1$ is a constant independent of $t$ and $n$.*

*Proof of Lemma A.6.* We use a similar argument as in the proof of Theorem 14.1 in [60]. Define

$$Z_n(t) := \sup_{\|h\|_{\mathbb{H}_k} \leq 1, \|h\|_{L^2_{\rho_{\mathbf{x}}}} \leq t} \left|\|h\|_{L^2_{\rho_{\mathbf{x}},n}}^2 - \|h\|_{L^2_{\rho_{\mathbf{x}}}}^2\right|$$

where $t \in (0,1]$. Let

$$\mathcal{E} := \left\{\sup_{\|h\|_{\mathbb{H}_k} \leq 1} \frac{\left|\|h\|_{L^2_{\rho_{\mathbf{x}},n}}^2 - \|h\|_{L^2_{\rho_{\mathbf{x}}}}^2\right|}{\|h\|_{L^2_{\rho_{\mathbf{x}}}}^2 + t^2} \leq \frac{1}{2}\right\}^c, \quad \mathcal{A} := \left\{Z_n(t) \geq \frac{t^2}{2}\right\}, \quad \tilde{\mathcal{A}} := \left\{Z_n(t) \geq \frac{t^{1+2r}}{2}\right\}.$$

We first show that $\mathcal{E} \subset \mathcal{A}$. On the event $\mathcal{E}$, there exists $h \in \mathbb{H}_k$ such that $\|h\|_{\mathbb{H}_k} \leq 1$ and

$$\frac{\left|\|h\|_{L^2_{\rho_{\mathbf{x}},n}}^2 - \|h\|_{L^2_{\rho_{\mathbf{x}}}}^2\right|}{\|h\|_{L^2_{\rho_{\mathbf{x}}}}^2 + t^2} > \frac{1}{2}.$$

If $\|h\|_{L^2_{\rho_{\mathbf{x}}}} \leq t$, we have $\left|\|h\|_{L^2_{\rho_{\mathbf{x}},n}}^2 - \|h\|_{L^2_{\rho_{\mathbf{x}}}}^2\right| > \frac{1}{2}\|h\|_{L^2_{\rho_{\mathbf{x}}}}^2 + \frac{1}{2}t^2 \geq \frac{1}{2}t^2$. Otherwise, set $\tilde{h} = \frac{t}{\|h\|_{L^2_{\rho_{\mathbf{x}}}}}h$ then $\|\tilde{h}\|_{L^2_{\rho_{\mathbf{x}}}} = t$ and $\left|\|\tilde{h}\|_{L^2_{\rho_{\mathbf{x}},n}}^2 - \|\tilde{h}\|_{L^2_{\rho_{\mathbf{x}}}}^2\right| > \frac{t^2}{\|h\|_{L^2_{\rho_{\mathbf{x}}}}^2} \cdot \left(\frac{1}{2}\|h\|_{L^2_{\rho_{\mathbf{x}}}}^2 + \frac{1}{2}t^2\right) \geq \frac{1}{2}t^2$. Therefore, $\mathcal{E} \subset \mathcal{A}$.

Since $\frac{t^2}{2} \geq \frac{t^{1+2r}}{2}$ for $t \in (0,1]$, we have $\mathcal{E} \subset \mathcal{A} \subset \tilde{\mathcal{A}}$. To find an upper bound of $\mathbb{E}Z_n(t)$, we use the symmetrization argument as follows:

$$\mathbb{E}Z_n(t) = \mathbb{E}\left[\sup_{\|h\|_{\mathbb{H}_k} \leq 1, \|h\|_{L^2_{\rho_{\mathbf{x}}}} \leq t} \left|\frac{1}{n}\sum_{i=1}^n h(\mathbf{x}^i)^2 - \mathbb{E}h(\mathbf{x})^2\right|\right]$$

$$= \mathbb{E}\left[\sup_{\|h\|_{\mathbb{H}_k} \leq 1, \|h\|_{L^2_{\rho_{\mathbf{x}}}} \leq t} \left|\mathbb{E}\left[\frac{1}{n}\sum_{i=1}^n h(\mathbf{x}^i)^2 - \frac{1}{n}\sum_{i=1}^n h(\tilde{\mathbf{x}}^i)^2 \,\bigg|\, X\right]\right|\right]$$

$$\leq \mathbb{E}\left[\sup_{\|h\|_{\mathbb{H}_k} \leq 1, \|h\|_{L^2_{\rho_{\mathbf{x}}}} \leq t} \left|\frac{1}{n}\sum_{i=1}^n w_i\left(h(\mathbf{x}^i)^2 - h(\tilde{\mathbf{x}}^i)^2\right)\right|\right]$$

$$\leq 2\mathbb{E}\left[\sup_{\|h\|_{\mathbb{H}_k} \leq 1, \|h\|_{L^2_{\rho_{\mathbf{x}}}} \leq t} \left|\frac{1}{n}\sum_{i=1}^n w_i h(\mathbf{x}^i)^2\right|\right]$$

where $w_1, \cdots, w_n$ are i.i.d. Rademacher variables. By Lemma A.15,

$$2\mathbb{E}\left[\sup_{\|h\|_{\mathbb{H}_k} \leq 1, \|h\|_{L^2_{\rho_{\mathbf{x}}}} \leq t} \left|\frac{1}{n}\sum_{i=1}^n w_i h(\mathbf{x}^i)^2\right|\right] \leq 4\kappa \cdot \overline{Q}_n(t).$$

For $t \geq \epsilon_n$, we have $4\kappa \cdot \overline{Q}_n(t) \leq \frac{t^{1+2r}}{4}$ since

$$\overline{Q}_n(\epsilon_n) = \frac{\epsilon_n^{1+2r}}{16\kappa} \quad \text{and} \quad \overline{Q}_n(\epsilon) \leq \frac{\epsilon^{1+2r}}{16\kappa}, \quad \forall \epsilon \geq \epsilon_n. \tag{13}$$

Since

$$\frac{1}{n}\sum_{i=1}^{n}\sup_{\|h\|_{\mathbb{H}_k}\leq 1,\|h\|_{L^2_{\rho_\mathbf{x}}}\leq t}\mathbb{E}\left[(h(\mathbf{x}^i)^2-\mathbb{E}h(\mathbf{x})^2)^2\right]\leq \sup_{\|h\|_{\mathbb{H}_k}\leq 1,\|h\|_{L^2_{\rho_\mathbf{x}}}\leq t}\mathbb{E}\left[h(\mathbf{x})^4\right]\leq \kappa^2 t^2,$$

Lemma A.16 gives

$$\mathbb{P}\left(Z_n(t)\geq \mathbb{E}Z_n(t)+\frac{u^{1+2r}}{4}\right)\leq \exp\left(-\frac{\frac{1}{16}n^2 u^{2+4r}}{4\kappa^2\cdot n\mathbb{E}Z_n(t)+2n\kappa^2 t^2+\frac{2}{3}\kappa^2\cdot\frac{1}{4}nu^{1+2r}}\right)$$

$$\leq \exp\left(-c_1 n\left(\frac{u^{2+4r}}{t^{1+2r}}\wedge\frac{u^{2+4r}}{t^2}\wedge u^{1+2r}\right)\right)$$

where $c_1=\frac{1}{96\kappa^2}$. Putting $u=t$ gives

$$\mathbb{P}(\tilde{\mathcal{A}})\leq \mathbb{P}\left(Z_n(t)\geq \mathbb{E}Z_n(t)+\frac{t^{1+2r}}{4}\right)\leq \exp\left(-c_1 n\left(t^{1+2r}\wedge t^{4r}\right)\right)\leq \exp(-c_1 nt^{4r}),$$

i.e., $\mathbb{P}(\mathcal{E}^c)\geq \mathbb{P}(\tilde{\mathcal{A}}^c)\geq 1-\exp(-c_1 nt^{4r})$.  □

Let us return to our problem. Consider the event

$$\mathcal{E}^c=\left\{\sup_{\|h\|_{\mathbb{H}_k}\leq 1}\frac{\left|\|h\|^2_{L^2_{\rho_\mathbf{x},n}}-\|h\|^2_{L^2_{\rho_\mathbf{x}}}\right|}{\|h\|^2_{L^2_{\rho_\mathbf{x}}}+\tilde{\epsilon}_n^2}\leq\frac{1}{2}\right\}.$$

By Lemma A.6, we have $\mathbb{P}(\mathcal{E}^c)\geq 1-\exp(-c_1 n\tilde{\epsilon}_n^{4r})$ where $c_1$ is a constant that does not depend on $n$. Define

$$T:=\min\left\{t\in\mathbb{N}:\frac{1}{\sqrt{\eta Et}}\leq\tilde{\epsilon}_n\right\}.$$

By the definition, we can easily obtain the upper bound of $T$ as $T<1+\frac{1}{\eta E\tilde{\epsilon}_n^2}$. Since $\mathcal{R}(\cdot)$ is non-decreasing,

$$\mathcal{R}\left(\frac{1}{\sqrt{\eta ET}}\right)\leq\mathcal{R}(\tilde{\epsilon}_n)\leq\frac{1}{c_l}Q_n(\tilde{\epsilon}_n)$$

for some absolute constant $c_l$ where the second inequality follows from Lemma A.2. Note that

$$\|h\|^2_{L^2_{\rho_\mathbf{x},n}}-\|h\|^2_{L^2_{\rho_\mathbf{x}}}\geq-\frac{1}{2}\left(\|h\|^2_{L^2_{\rho_\mathbf{x}}}+\tilde{\epsilon}_n^2\right)\Rightarrow\|h\|^2_{L^2_{\rho_\mathbf{x},n}}\geq\frac{1}{2}\|h\|^2_{L^2_{\rho_\mathbf{x}}}-\frac{1}{2}\tilde{\epsilon}_n^2$$

for all $h$ such that $\|h\|_{\mathbb{H}_k}\leq 1$ on the event $\mathcal{E}^c$. Thus,

$$Q_n(\tilde{\epsilon}_n)=\mathbb{E}\left[\sup_{\|g\|_{\mathbb{H}_k}\leq 1,\|g\|_{L^2_{\rho_\mathbf{x},n}}\leq\tilde{\epsilon}_n}\left|\frac{1}{n}\sum_{i=1}^{n}w_i g(\mathbf{x}^i)\right|\ \middle|\ X\right]\leq\mathbb{E}\left[\sup_{\|g\|_{\mathbb{H}_k}\leq 1,\|g\|_{L^2_{\rho_\mathbf{x}}}\leq 2\tilde{\epsilon}_n}\left|\frac{1}{n}\sum_{i=1}^{n}w_i g(\mathbf{x}^i)\right|\ \middle|\ X\right]$$

on the event $\mathcal{E}^c$. Set $\mathcal{F}=\left\{g:\mathcal{X}\to\mathbb{R}:\|g\|_{\mathbb{H}_k}\leq 1,\|g\|_{L^2_{\rho_\mathbf{x}}}\leq 2\tilde{\epsilon}_n\right\}$. Then the ranges of functions in $\mathcal{F}$ are contained in $[-\kappa,\kappa]$ and $\mathcal{F}=-\mathcal{F}$. By Lemma A.17 we have

$$\mathbb{E}\left[\sup_{\|g\|_{\mathbb{H}_k}\leq 1,\|g\|_{L^2_{\rho_\mathbf{x}}}\leq 2\tilde{\epsilon}_n}\left|\frac{1}{n}\sum_{i=1}^{n}w_i g(\mathbf{x}^i)\right|\ \middle|\ X\right]\leq 2\mathbb{E}\left[\sup_{\|g\|_{\mathbb{H}_k}\leq 1,\|g\|_{L^2_{\rho_\mathbf{x}}}\leq 2\tilde{\epsilon}_n}\left|\frac{1}{n}\sum_{i=1}^{n}w_i g(\mathbf{x}^i)\right|\right]+c_1\kappa\tilde{\epsilon}_n^{1+2r}$$

with probability at least $1-\exp(-c_1 n\tilde{\epsilon}_n^{1+2r})\geq 1-\exp(-c_1 n\tilde{\epsilon}_n^{4r})$. Hence,

$$\mathcal{R}\left(\frac{1}{\sqrt{\eta ET}}\right)\leq\frac{2}{c_l}\cdot\overline{Q}_n(2\tilde{\epsilon}_n)+\frac{c_1\kappa}{c_l}\tilde{\epsilon}_n^{1+2r}\leq\left(\frac{1}{\kappa c_l}+\frac{c_1\kappa}{c_l}\right)\tilde{\epsilon}_n^{1+2r}$$

holds with probability at least $1-2\exp(-c_1 n\tilde{\epsilon}_n^{4r})$. Here, the second inequality follows from (13). Therefore,

$$\mathbb{E}\mathcal{R}\left(\frac{1}{\sqrt{\eta ET}}\right)^2\leq\left(\frac{8}{\kappa c_l}+\frac{\kappa c_1}{c_l}\right)^2\tilde{\epsilon}_n^{2+4r}\cdot(0\vee(1-2\exp(-c_1 n\tilde{\epsilon}_n^{4r})))+\kappa^2\cdot 2\exp(-c_1 n\tilde{\epsilon}_n^{4r})$$

$$\leq\left(\frac{8}{\kappa c_l}+\frac{\kappa c_1}{c_l}\right)^2\tilde{\epsilon}_n^{2+4r}+2\kappa^2\exp(-c_1 n\tilde{\epsilon}_n^{4r})$$

since $\mathcal{R}\left(\frac{1}{\sqrt{\eta ET}}\right)\leq\kappa$. From the fact that $\frac{\exp(-c_1 n\tilde{\epsilon}_n^{4r})}{\tilde{\epsilon}_n^{2+4r}}\lesssim n^{\frac{2r+1}{2r+s}}\exp(-c_1' n^{\frac{s}{2r+s}})\lesssim 1$, we have

$$\left(\mathbb{E}\mathcal{R}\left(\frac{1}{\sqrt{\eta ET}}\right)^2\right)^{1/2}\lesssim\tilde{\epsilon}_n^{1+2r}\lesssim n^{-\frac{2r+1}{4r+2s}}.$$

### A.2.6 Conclusion

Note that

$$\frac{1}{T} \lesssim \tilde{\epsilon}_n^2 \lesssim n^{-\frac{1}{2r+s}} \quad \text{and} \quad T \le 1 + \frac{1}{\eta E \tilde{\epsilon}_n^2} \lesssim n^{\frac{1}{2r+s}}.$$

Therefore, we bound the expected risk as

$$
\mathbb{E}\|\iota_{\rho_{\mathbf{x}}}(f_T - f_0^*)\|_{L^2_{\rho_{\mathbf{x}}}}
$$

$$
\lesssim B^{r-1/2}\left(\frac{1}{T} + n^{-\frac{1}{2r+s}}\right)^{1/2} n^{-\frac{r-1/2}{2r+s}} + \left(T^{1/2} + n^{-\frac{1/2}{2r+s}}T\right) \cdot \left(\mathbb{E}\mathcal{R}\left(\frac{1}{\sqrt{\eta T E}}\right)^2\right)^{1/2}
$$

$$
+ B^r(1 + \log T + T^{1/2}n^{-\frac{1/2}{2r+s}})n^{-\frac{r}{2r+s}}
$$

$$
\lesssim B^r n^{-\frac{r}{2r+s}} \log n.
$$

### A.3 Corollary of Theorem 3.4

As mentioned in Section 3.3, one can remove $B^r$ in the upper bound in Theorem 3.4 by using more public inputs. The precise statement is as follows:

**Corollary A.7.** *Under Assumption 3.1, 3.2, and 3.3, with $n_0 \ge B^{1+\epsilon}n^{\frac{1}{2r+s}}(\log(Bn))^3$ public inputs independently generated from $\tilde{\rho}_{\mathbf{x}}$ satisfying (2) DCL-KR gives the performance guarantee*

$$
\mathbb{E}\|\iota_{\rho_{\mathbf{x}}}(f_{j,T} - f_0^*)\|_{L^2_{\rho_{\mathbf{x}}}} \le C \cdot n^{-\frac{r}{2r+s}} \log n
$$

*for all $j = 1, \cdots, m$ where $\epsilon > 0$ is a fixed constant, $\eta \in (0, 1/\kappa^2)$ is a fixed learning rate, $T$ is an adequate stopping rule, and the prefactor $C$ does not depend on $B$, $m$, and $n$.*

*Proof.* In the proof of Theorem 3.4, there are two terms in the upper bound affected by $B$. One is

$$
B^{r-1/2}\left(\frac{(\log n_0)^3}{n_0}\right)^{r-1/2}
$$

in the norm bound of the first term in (7). The other is

$$
\mathbb{E}\|(T_{k,\tilde{\rho}_{\mathbf{x}}} + \lambda I)^{1/2}(I - P_{D_p})\|^{2r} \le 2^r\lambda^r + (\kappa^2 + 1) \cdot 4\exp\left(-\frac{1}{4(\kappa^2+1)\mathcal{B}_0}\right)
$$

in the norm bound of the third and fourth terms in (7). For the first part, $n_0 \ge B^{1+\epsilon}n^{\frac{1}{2r+s}}(\log(Bn))^3$ implies

$$
B^{r-1/2}\left(\frac{(\log n_0)^3}{n_0}\right)^{r-1/2}
$$

$$
\le B^{-\epsilon(r-1/2)}n^{-\frac{r-1/2}{2r+s}}(\log(Bn))^{-3r+3/2}\left((1+\epsilon)\log B + \frac{1}{2r+s}\log n + 3\log\log(Bn)\right)^{3r-3/2}
$$

$$
\lesssim n^{-\frac{r-1/2}{2r+s}}.
$$

For the latter part, set $\lambda = 128(\kappa^2+1)^2 n^{-\frac{1}{2r+s}}/B$. Then

$$
\mathcal{B}_0 \le \frac{\log\kappa^2 e + \log(1/\lambda)}{\lambda n_0} + \sqrt{\frac{\log\kappa^2 e + \log(1/\lambda)}{\lambda n_0}} \le \frac{1}{4(\kappa^2+1)\log(Bn)}
$$

and hence

$$
\mathbb{E}\|(T_{k,\tilde{\rho}_{\mathbf{x}}}+\lambda I)^{1/2}(I-P_{D_p})\|^{2r} \le 2^r\lambda^r + (\kappa^2+1)\cdot 4\exp\left(-\frac{1}{4(\kappa^2+1)\mathcal{B}_0}\right) \lesssim B^{-r}n^{-\frac{r}{2r+s}} + \frac{1}{Bn}.
$$

Since it eliminates $B^r$ in the upper bound, we are done. $\qquad\square$

## A.4 Useful Lemmas

Recall Cordes' inequality [16].

**Lemma A.8** (Cordes' Inequality). *Let $A, B$ be two bounded positive linear operators on a seperable Hilbert space. Then for any $s \in [0, 1]$*

$$\|A^s B^s\| \leq \|AB\|^s$$

*holds.*

We also recall a property of projection operators.

**Lemma A.9** ([53]). *Let $Z$ be a bounded linear operator and $P$ be a projection operator such that $\mathrm{ran}\, P = \overline{\mathrm{ran}\, Z^\top}$. Then for any bounded operator $X$ and $\lambda > 0$ we have*

$$\|(I - P)X\| \leq \lambda^{1/2}\|(Z^\top Z + \lambda I)^{-1/2}X\|.$$

There are some useful lemmas for PAC bounds.

**Lemma A.10** ([17]). *Let $X = \{\mathbf{x}^1, \cdots, \mathbf{x}^n\}$ be a dataset where data points are independently generated from $\nu$. Then*

$$\|(T_{k,\nu} + \lambda I)(T_{k,X} + \lambda I)^{-1}\| \leq 2 + 2\left(\left(\frac{2\kappa^2}{n\lambda} + \sqrt{\frac{4\kappa^2 \mathcal{N}_\nu(\lambda)}{n\lambda}}\right)\log(2/\delta)\right)^2$$

*holds with confidence at least $1 - \delta$ where $\delta \in (0, 1)$.*

**Lemma A.11** ([38]). *Let $X = \{\mathbf{x}^1, \cdots, \mathbf{x}^n\}$ be a dataset where data points are independently generated from $\nu$. Then*

$$\|(T_{k,\nu} + \lambda I)^{-1}(T_{k,X} + \lambda I)\| \leq 1 + \left(\frac{2\kappa^2}{n\lambda} + \sqrt{\frac{4\kappa^2 \mathcal{N}_\nu(\lambda)}{n\lambda}}\right)\log(2/\delta)$$

*holds with confidence at least $1 - \delta$ where $\delta \in (0, 1)$.*

**Lemma A.12** ([48]). *Let $X = \{\mathbf{x}^1, \cdots, \mathbf{x}^n\}$ be a dataset where data points are independently generated from $\nu$. For $\lambda \in (0, 1]$ such that $\mathcal{N}_\nu(\lambda) \geq 1$,*

$$\|(T_{k,\nu} + \lambda I)(T_{k,X} + \lambda I)^{-1}\| \leq 2$$

*holds with confidence at least $1 - \delta$ where*

$$4\exp\left(-\frac{1}{4(\kappa^2 + 1)} \cdot \left(\frac{1 + \log \mathcal{N}_\nu(\lambda)}{\lambda n} + \sqrt{\frac{1 + \log \mathcal{N}_\nu(\lambda)}{\lambda n}}\right)^{-1}\right) \leq \delta < 1.$$

To prove Lemma A.2 in Appendix A.2.5, we introduce a concentration inequality for Lipschitz functions.

**Lemma A.13** ([60]). *Let $X_1, \cdots, X_n$ be independent random variables whose supports are contained in $[a, b]$ and $f : \mathbb{R}^n \to \mathbb{R}$ be convex and $L$-Lipschitz with respect to the Euclidean norm. Then we have*

$$\mathbb{P}(f(X) \geq \mathbb{E}f(X) + t) \leq \exp\left(-\frac{t^2}{4L^2(b-a)^2}\right)$$

*where $X = [X_1, \cdots, X_n]$ and $t > 0$.*

Precisely, we use the following fact in Appendix A.2.5

*Remark* A.14 ([60]). Let $A \subset \mathbb{R}^n$ be a bounded set and

$$f(\mathbf{x}) = \sup_{\mathbf{a} \in A} \sum_{k=1}^n a_k \mathbf{x}_k$$

where $\mathbf{x} = [\mathbf{x}_1, \cdots, \mathbf{x}_n] \in [-1, 1]^n$ and $\mathbf{a} = [a_1, \cdots, a_n]$. Since

$$f(\mathbf{x}) - f(\mathbf{x}') = \sup_{\mathbf{a} \in A} \sum_{k=1}^n a_k \mathbf{x}_k - \sup_{\mathbf{a} \in A} \sum_{k=1}^n a_k \mathbf{x}'_k \leq \sup_{\mathbf{a} \in A} \langle \mathbf{a}, \mathbf{x} - \mathbf{x}' \rangle_{\mathbb{R}^n} \leq \sup_{\mathbf{a} \in A} \|\mathbf{a}\|_{\mathbb{R}^n} \|\mathbf{x} - \mathbf{x}'\|_{\mathbb{R}^n},$$

$f$ is a $\sup_{\mathbf{a} \in A} \|\mathbf{a}\|_{\mathbb{R}^n}$-Lipshitz function where $\|\cdot\|_{\mathbb{R}^n}$ is the Euclidean norm on $\mathbb{R}^n$. We can observe that $f$ is convex since $f$ is a supremum of convex functions defined on a convex compact set.

To prove Lemma A.6 in Appendix A.2.5, we recall the Ledoux-Talagrand contraction inequality [29, 56] and Talagrand's inequality [4, 56].

**Lemma A.15** (Ledoux-Talagrand Contraction Inequality). *If $\phi : \mathbb{R} \to \mathbb{R}$ is a L-Lipshitz function, then*

$$\mathbb{E}\left[\sup_{h \in \mathcal{F}} \frac{1}{n} \sum_{i=1}^{n} \epsilon_i \phi(h(\mathbf{x}_i))\right] \le L \cdot \mathbb{E}\left[\sup_{h \in \mathcal{F}} \frac{1}{n} \sum_{i=1}^{n} \epsilon_i h(\mathbf{x}_i)\right].$$

**Lemma A.16** (Talagrand's Inequality). *Let $X_1, \cdots, X_n$ be independent $\mathcal{X}$-valued random variables. Let $\mathcal{F}$ be a countably family of measurable real-valued functions on $\mathcal{X}$ such that $\|f\|_\infty \le U < \infty$ and $\mathbb{E}f(X_i) = 0$ for all $f \in \mathcal{F}$. Let*

$$Z := \sup_{f \in \mathcal{F}} \sum_{i=1}^{n} f(X_i), \quad \sigma^2 \ge \frac{1}{n} \sum_{i=1}^{n} \sup_{f \in \mathcal{F}} \mathbb{E}[f(X_i)^2], \quad \nu_n := 2U\mathbb{E}Z + n\sigma^2.$$

*Then*

$$\mathbb{P}(Z \ge \mathbb{E}Z + t) \le \exp\left(-\frac{t^2}{2\nu_n + \frac{2}{3}Ut}\right)$$

*for all $t \ge 0$.*

Lastly, we recall the following well-known property used in Appendix A.2.5.

**Lemma A.17** ([2]). *Let $\mathcal{F}$ be a class of functions with ranges in $[a, b]$ and $w_1, \cdots, w_n$ be i.i.d. Rademacher variables. Then*

$$\frac{1}{n}\mathbb{E}\left[\sup_{h \in \mathcal{F}} \sum_{i=1}^{n} w_i f(\mathbf{x}^i)\right] \le \inf_{\alpha \in (0,1)} \left(\frac{1}{1-\alpha}\frac{1}{n}\mathbb{E}\left[\sup_{h \in \mathcal{F}} \sum_{i=1}^{n} w_i f(\mathbf{x}^i) \,\bigg|\, X\right] + \frac{(b-a)\log(1/\delta)}{4n\alpha(1-\alpha)}\right)$$

*holds with probability at least $1 - \delta$. Also,*

$$\frac{1}{n}\mathbb{E}\left[\sup_{h \in \mathcal{F}} \sum_{i=1}^{n} w_i f(\mathbf{x}^i) \,\bigg|\, X\right]$$
$$\le \inf_{\alpha > 0} \left((1+\alpha)\frac{1}{n}\mathbb{E}\left[\sup_{h \in \mathcal{F}} \sum_{i=1}^{n} w_i f(\mathbf{x}^i)\right] + \frac{(b-a)\log(1/\delta)}{2n}\left(\frac{1}{2\alpha} + \frac{1}{3}\right)\right)$$

*holds with probability at least $1 - \delta$.*

## B  Details on DCL-NN Algorithm

As we mentioned before, DCL-NN considers the same problem as in Section 3 but local models are heterogeneous neural networks. That is, there are $m$ parties and $D_i = \{(\mathbf{x}_i^j, y_i^j) : j = 1, \cdots, n_i\}$ is the private dataset of the $i$th party ($i = 1, \cdots, m$) where $D = \bigcup_{i=1}^{m} D_i$ are i.i.d. whose distribution is $\rho_{\mathbf{x},y}$. One remark is that the local data distributions of parties are not the same in general. To communicate training information, we introduce an unlabeled public input dataset $Z = \{\mathbf{z}^1, \cdots, \mathbf{z}^{n_0}\} \subset \mathcal{X}$. The goal of parties is to find a minimizer of the population risk $\mathcal{E}$ defined in Section 3.

To extend DCL-KR to heterogeneous neural network settings, it is necessary to ensure that the assumptions of DCL-KR are satisfied as much as possible. Specifically, one important assumption in DCL-KR is the equality of kernels across local models. Indeed, public data predictions can vary in conflicting directions after the local training procedure, even when using the same local datasets, if the kernels differ.

For further explanation of this claim, we consider the simple case of $E = 1$ in DCL-KR where $E$ is the number of local iterations. After the consensus prediction $u$ is distributed to local parties, the server then receives the updated local prediction on $Z$:

$$(I - \frac{\eta}{n_i} K_{ZX_i} K_{X_i \bar{Z}} K_{\bar{Z}\bar{Z}}^{-1})u + \frac{\eta}{n_i} K_{ZX_i} \mathbf{y}_i \tag{14}$$

from the $i$th local party. (The notation is consistent with Appendix A) Suppose two parties have exactly the same dataset. If the same kernel is used in these two parties, the updated local predictions

---

**Algorithm 2** DCL-NN Algorithm

---

1: **Hyperparameters:** $D_i$: local dataset of party $i$ ($i = 1, \cdots, m$), $Z = \{\mathbf{z}^1, \cdots, \mathbf{z}^{n_0}\}$: public inputs, $E$: the number of local iterations at each communication round, $T$: total communication rounds, $T_k$: epochs of kernel distillation

2: **Pretrain:** Pretrain local AI models $f_i(\cdot) = \mathbf{w}_i^\top g_i(\cdot) + b_i$ ($i = 1, \cdots, m$).

3: *# Feature Kernel Distillation Procedure*

4: For party $i$ ($i = 1, \cdots, m$), compute the feature kernel values on public inputs $\{k_{f_i}(\mathbf{z}^{j_1}, \mathbf{z}^{j_2}) : 1 \leq j_1, j_2 \leq n_0\}$ via (3) and upload them to the server.

5: The server aggregates the local feature kernel values to the target kernel values $\{k(\mathbf{z}^{j_1}, \mathbf{z}^{j_2}) : 1 \leq j_1, j_2 \leq n_0\}$ with (4) and distributes them to all parties.

6: **for** party $i = 1, \cdots, m$ **do**

7:     **for** $t_k = 0, \cdots, T_k - 1$ **do**

8:         **for** mini-batch $Z_0 \subset Z$ **do**

9:             Update parameters of its model $f_i$ by maximizing $\widehat{\mathrm{CKA}}(k, k_{f_i})$ on $Z_0$ defined as (16) via gradient descent where $k$ is fixed.

10:         **end for**

11:     **end for**

12: **end for**

13: *# Collaborative Learning Procedure*

14: Initialize the consensus prediction $\mathbf{y}_{p,0} = 0$.

15: **for** $t = 0, \cdots, T - 1$ **do**

16:     **for** party $i = 1, \cdots, m$ **do**

17:         Update $\mathbf{w}_i$ and $b_i$ by maximizing MSE (Mean Squared Error) on the public inputs $Z$ with consensus prediction $\mathbf{y}_{p,t}$ via gradient descent with sufficiently many iterations.

18:         **for** $e = 1, \cdots, E$ **do**

19:             Update $\mathbf{w}_i$ and $b_i$ by maximizing MSE on $D_i$ via gradient descent.

20:         **end for**

21:         Upload the local prediction $\mathbf{y}_{p,t+1}^i$ on $Z$ to the server.

22:     **end for**

23:     The server aggregates the local predictions to compute the consensus prediction $\mathbf{y}_{p,t+1} = \sum_{i=1}^m \frac{n_i}{n} \mathbf{y}_{p,t+1}^i$ and distributes $\mathbf{y}_{p,t+1}$ to all parties.

24: **end for**

---

will be identical. However, if the kernels are different, this will not be the case. For kernels like the Gaussian kernel, which have high correlation between close inputs, the updated local predictions will be strongly influenced by data points close to each input. On the other hand, for kernels like the linear kernel, which have high correlation between distant inputs, the updated local prediction on $Z$ will be influenced more by data points farther from each input. We can observe this fact from the above formula (14). This observation implies that aggregating local learning information becomes very challenging when the kernels differ. In short, using the same kernel ensures that the shift mechanisms of predictions on $Z$ at the edges are identical, making it possible for the aggregation through simple weighted averaging to work well. This is a key element of the strong theoretical results of DCL-KR and explains why kernel matching between neural networks is necessary in DCL-NN.

Let $f_i$ be a local model of the $i$th party such that $f_i(\cdot) = \mathbf{w}_i^\top g_i(\cdot) + b_i$, $g_i : \mathcal{X} \to \mathbb{R}^{c_i}$, $\mathbf{w}_i \in \mathbb{R}^{c_i}$, $c_i \in \mathbb{N}$, and $b_i \in \mathbb{R}$ for $i = 1, \cdots, m$. Since most modern neural network architectures have a linear layer as the last layer, this setting is general enough. As (3), we set the feature kernel of $f_i$ ($i = 1, \cdots, m$) to be

$$k_{f_i}(\mathbf{x}^1, \mathbf{x}^2) = g_i(\mathbf{x}^1)^\top g_i(\mathbf{x}^2), \quad \mathbf{x}^1, \mathbf{x}^2 \in \mathcal{X}.$$

To bring the setting to the DCL-KR scheme, DCL-NN matches $k_{f_1}, \cdots, k_{f_m}$ via kernel distillation procedure. Obviously, the target kernel in this procedure is a key factor in enhancing performance.

Theoretically, using the ensemble kernel

$$k = \sum_{i=1}^m \frac{n_i}{n} k_{f_i}. \tag{15}$$

can be a good way to construct a good kernel derived from local feature kernels. The reason is that this ensemble kernel is identical to the kernel induced by the (scaled) concatenation of the local feature maps, i.e.,

$$k(\mathbf{x}^1, \mathbf{x}^2) = \sum_{i=1}^{m} \frac{n_i}{n} k_{f_i}(\mathbf{x}^1, \mathbf{x}^2) = \sum_{i=1}^{m} \frac{n_i}{n} g_i(\mathbf{x}^1)^\top g_i(\mathbf{x}^2)$$

$$= \begin{bmatrix} \sqrt{\frac{n_1}{n}} g_1(\mathbf{x}^1)^\top & \sqrt{\frac{n_2}{n}} g_2(\mathbf{x}^1)^\top & \cdots & \sqrt{\frac{n_m}{n}} g_m(\mathbf{x}^1)^\top \end{bmatrix} \begin{bmatrix} \sqrt{\frac{n_1}{n}} g_1(\mathbf{x}^2) \\ \sqrt{\frac{n_2}{n}} g_2(\mathbf{x}^2) \\ \cdots \\ \sqrt{\frac{n_m}{n}} g_m(\mathbf{x}^2) \end{bmatrix}.$$

In other words, the ensemble kernel has greater expressive power than individual feature kernels, and with a sufficient amount of data, it leads to better performance. We empirically verify that the performance of this ensemble kernel surpasses that of individual feature kernels in Figure 4.

DCL-NN sets this ensemble kernel $k$ as the target kernel and local parties match their local feature kernels $k_{f_1}, \cdots, k_{f_m}$ with the kernel $k$ using the public dataset $Z$. For this purpose, we introduce Centered Kernel Alignment (CKA) [8] as a kernel similarity measure. The CKA between two kernels $k_1$ and $k_2$ on the public input distribution $\tilde{\rho}_\mathbf{x}$ is given by

$$\mathrm{CKA}(k_1, k_2) = \frac{\mathrm{HSIC}(k_1, k_2)}{\sqrt{\mathrm{HSIC}(k_1, k_1)\mathrm{HSIC}(k_2, k_2)}}$$

where $\mathrm{HSIC}(\cdot, \cdot)$ is a Hilbert-Schmidt Independence Criterion (HSIC) defined as

$$\mathrm{HSIC}(k_i, k_j) = \mathbb{E}_{\mathbf{x}^1, \mathbf{x}^2 \sim \tilde{\rho}_\mathbf{x}} [k_i^c(\mathbf{x}^1, \mathbf{x}^2) k_j^c(\mathbf{x}^1, \mathbf{x}^2)]$$

and the centered kernel $k_i^c$ is given by

$$k_i^c(\mathbf{x}^1, \mathbf{x}^2) = k_i(\mathbf{x}^1, \mathbf{x}^2) - \mathbb{E}_{\tilde{\mathbf{x}}^2 \sim \tilde{\rho}_\mathbf{x}}[k_i(\mathbf{x}^1, \tilde{\mathbf{x}}^2)] - \mathbb{E}_{\tilde{\mathbf{x}}^1 \sim \tilde{\rho}_\mathbf{x}}[k_i(\tilde{\mathbf{x}}^1, \mathbf{x}^2)] + \mathbb{E}_{\tilde{\mathbf{x}}^1, \tilde{\mathbf{x}}^2 \sim \tilde{\rho}_\mathbf{x}}[k_i(\tilde{\mathbf{x}}^1, \tilde{\mathbf{x}}^2)],$$

$\mathbf{x}^1, \mathbf{x}^2 \in \mathcal{X}$ ($i = 1, 2$). However, since we have a finite number of samples, we employ the empirical CKA. The empirical CKA between two kernels $k_1$ and $k_2$ on inputs $\{\mathbf{c}^1, \cdots, \mathbf{c}^p\}$ is given by

$$\widehat{\mathrm{CKA}}(k_1, k_2) = \frac{\widehat{\mathrm{HSIC}}(K_1, K_2)}{\sqrt{\widehat{\mathrm{HSIC}}(K_1, K_1)\widehat{\mathrm{HSIC}}(K_2, K_2)}} \tag{16}$$

where

$$K_1 = \begin{bmatrix} k_1(\mathbf{c}^1, \mathbf{c}^1) & \cdots & k_1(\mathbf{c}^1, \mathbf{c}^p) \\ \vdots & \ddots & \vdots \\ k_1(\mathbf{c}^p, \mathbf{c}^1) & \cdots & k_1(\mathbf{c}^p, \mathbf{c}^p) \end{bmatrix} \text{ and } K_2 = \begin{bmatrix} k_2(\mathbf{c}^1, \mathbf{c}^1) & \cdots & k_2(\mathbf{c}^1, \mathbf{c}^p) \\ \vdots & \ddots & \vdots \\ k_2(\mathbf{c}^p, \mathbf{c}^1) & \cdots & k_2(\mathbf{c}^p, \mathbf{c}^p) \end{bmatrix}$$

are Gram matrices and $\widehat{\mathrm{HSIC}}$ is an estimator of HSIC defined as

$$\widehat{\mathrm{HSIC}}(L, M) = \frac{1}{(p-1)^2}\mathrm{tr}(LHMH), \quad L, M \in \mathbb{R}^{p \times p}$$

where $H := I_p - \frac{1}{p}\mathbf{1}\mathbf{1}^\top$ is the centering matrix, $I_p$ is a $p \times p$ identity matrix, and $\mathbf{1} = [1, 1, \cdots, 1]^\top$ is a $p$-dimensional one vector. During the kernel distillation procedure, the $i$th local party maximizes $\widehat{\mathrm{CKA}}(k_{f_i}, k)$ on $Z$ where $k$ is a fixed target kernel given by (4). In practice, we use batching to perform the kernel distillation to reduce computational costs.

Due to the definition of the empirical CKA, it is necessary to calculate the Gram matrix of $k$ over $Z$. To this end, the $i$th local party calculates the Gram matrix of $k_{f_i}$ over $Z$ and uploads it to the server for $i = 1, \cdots, m$. Then the server computes the Gram matrix of $k$ by weighted averaging the Gram matrices of local feature kernels $k_{f_1}, \cdots, k_{f_m}$. Since this process only requires communication of feature kernel values, DCL-NN still preserves the privacy of local model information.

While (empirical) CKA is a good metric for kernel matching, it is invariant to scaling, and therefore the local feature kernels resulting from the kernel distillation may have different scales. This affects the degree to which each local training influences during the DCL-KR-like follow-up procedure. To

illustrate this point, consider the following example: Let the feature kernels of two local models $f_1$ and $f_2$ be as follows:

$$k_{f_1}(\mathbf{x}, \mathbf{y}) = \phi(\mathbf{x})^\top \phi(\mathbf{y}), \qquad k_{f_2}(\mathbf{x}, \mathbf{y}) = (\alpha\phi(\mathbf{x}))^\top (\alpha\phi(\mathbf{y})).$$

After distilling on the public data with consensus predictions, these two models become like

$$f_1(\cdot) = \mathbf{w}^\top \phi(\cdot) + b, \qquad f_2(\cdot) = \left(\frac{1}{\alpha}\mathbf{w}\right)^\top (\alpha\phi(\cdot)) + b = \mathbf{w}^\top \phi(\cdot) + b,$$

i.e., two models are the same. Nevertheless, a gradient descent update on a data point $(\mathbf{x}_0, y_0)$ with a learning rate $\eta$ is

$$\mathbf{w}_1 \leftarrow \mathbf{w} - \eta(\mathbf{w}^\top \phi(\mathbf{x}_0) + b - y_0)$$

for $f_1$ and

$$\mathbf{w}_2 \leftarrow \frac{1}{\alpha}\mathbf{w} - \eta(\mathbf{w}^\top \phi(\mathbf{x}_0) + b - y_0)$$

for $f_2$. Thus, after the gradient descent step we have

$$f_1(\cdot) = (\mathbf{w} - \eta(\mathbf{w}^\top \phi(\mathbf{x}_0) + b - y_0))^\top \phi(\cdot) + b$$

and

$$f_2(\cdot) = \left(\frac{1}{\alpha}\mathbf{w} - \eta(\mathbf{w}^\top \phi(\mathbf{x}_0) + b - y_0)\right)^\top (\alpha\phi(\cdot)) + b = (\mathbf{w} - \alpha\eta(\mathbf{w}^\top \phi(\mathbf{x}_0) + b - y_0))^\top \phi(\cdot) + b.$$

Hence the scale $\alpha$ affects the collaborative learning procedure. To address this issue, we compute the scale $\alpha$ using the estimator $\widehat{\mathrm{HSIC}}$. Specifically, at the beginning of the collaborative learning phase, we compute $\alpha_i = \widehat{\mathrm{HSIC}}(K_i, K_i)$ where $K_i$ is the Gram matrix with respect to the feature kernel of the $i$th party on $Z$ $(i = 1, \cdots, m)$. Then we set the learning rate for the $i$th party as

$$\eta_0 \cdot \frac{\max_{1 \leq j \leq m} \alpha_j^{1/2}}{\alpha_i^{1/2}}$$

where $\eta_0$ is a base learning rate. In practice, computing HSIC over all public inputs is costly and unnecessary. Using only a small subset of public inputs is sufficient.

We present DCL-NN in Algorithm 2. Here are some remarks.

(1) The feature kernel distillation procedure requires only one round of two-way communication between the server and the parties.

(2) The collaborative learning procedure follows the same process as in DCL-KR with $g_1, \cdots, g_m$ fixed. Note that kernel gradient descent reduces to standard gradient descent since the kernels have finite rank. In this process, if possible, optimization on the public dataset can be performed using the closed-form solution of kernel linear regression instead of gradient descent.

(3) In this work, we apply FedMD for the pretraining of DCL-NN in the experiment. However, DCL-NN is a general algorithm that can use any algorithm for pretraining to obtain good local feature kernels. For example, kernel learning techniques [63] may be applied for pretraining.

(4) Our algorithm can be naturally extended to regression problems with multi-dimensional outputs.

## C  Details and Further Discussion on Experiments

### C.1  Dataset Description

For all datasets, we follow Algorithm 3 to construct non-i.i.d. settings. Note that this procedure is similar to the non-i.i.d. data generation procedure in classification tasks [44, 69].

**Algorithm 3** Data Generating Procedure

---

1: **Inputs:** the number of parties $m$, the total number of private data $n$, the number of public inputs $n_0$, the partition of input space $\mathcal{A} = \{\mathcal{A}_1, \cdots, \mathcal{A}_c\}$
2: Generate the whole private dataset size of $n$ from $\rho_{\mathbf{x},y}$ and the public inputs size of $n_0$ from $\tilde{\rho}_{\mathbf{x}}$.
3: Sample a base private data ratio $(\alpha_1, \cdots, \alpha_m)$ from $\mathrm{Dir}([10, 10, \cdots, 10])$.
4: **while** $\sum_{j=1}^{m} C_{ij} \neq 0$ for all $i = 1, \cdots, c$ **do**
5:      **for** party $j = 1, \cdots, m$ **do**
6:          Sample two elements uniformly from the partition $\mathcal{A}$.
7:      **end for**
8:      Set $C_{ij} = 1$ if $\mathcal{A}_i$ is chosen at the $j$th party and $C_{ij} = 0$ otherwise.
9: **end while**
10: **for** $i = 1, \cdots, c$ **do**
11:      Put the private data points where inputs are in $\mathcal{A}_i$ into the datasets of parties with ratio

$$
\left[ \frac{\alpha_k C_{ik}}{\sum_{j=1}^{m} \alpha_j C_{ij}} \right]_{k=1,\cdots,m}.
$$

12: **end for**

---

### C.1.1 Toy-1D

Let $\mathcal{X} = [0, 1] \subset \mathbb{R}$ and $\rho_x$ be the uniform distribution on $\mathcal{X}$. The space

$$
H^1 := \left\{ f \in \mathrm{AC}[0,1] \;\middle|\; f(0) = 0, \int f'(x)^2 \, d\rho_x(x) < \infty \right\}
$$

is the reproducing kernel Hilbert space associated to the kernel $k(x, y) = \min(x, y)$ where $\mathrm{AC}[0,1]$ is the collection of all absolutely continuous functions on $[0, 1]$ [33, 60]. As mentioned in [33], the covariance operator $T_{k,\rho_x}$ has eigenpairs $\{(\lambda_i, e_i)\}_{i \in \mathbb{N}}$ where

$$
\lambda_i = \left( \frac{2i - 1}{2} \pi \right)^{-2}, \qquad e_i(x) = \sqrt{2} \sin\left( \frac{2i - 1}{2} \pi x \right).
$$

Thus, the eigenvalue decay rate $s$ is $\frac{1}{2}$. Set a target function

$$
f_0^*(x) = \sum_{i=1}^{\infty} \frac{e_i(x)}{i^3} = \sum_{i=1}^{\infty} \frac{\sqrt{2}}{i^3} \sin\left( \frac{2i - 1}{2} \pi x \right).
$$

From the fact that

$$
\left\| T_{k,\rho_x}^{1/2 - r} \sum_{i=1}^{\infty} h_i e_i \right\|_{\mathbb{H}_k} \leq R \quad \Leftrightarrow \quad \sum_{i=1}^{\infty} \frac{h_i^2}{\lambda_i^{2r}} \leq R^2,
$$

we have $f_0^* = T_{k,\rho_x}^{1/2} g_0^*$ such that

$$
\|g_0^*\|_{\mathbb{H}_k} = \left( \sum_{i=1}^{\infty} \frac{1}{i^6} \left( \frac{2i - 1}{2} \pi \right)^4 \right)^{1/2} =: R < \infty.
$$

Then $r = 1$. We generate data points from $\rho_x \cdot \rho_{y|x}$ such that $\rho_x$ is the uniform distribution on $\mathcal{X}$ as above and

$$
\rho_{y|x} = \mathcal{N}(y | f_0^*(x), 0.44^2).
$$

We divide $\mathcal{X}$ into a partition $\mathcal{A} := \left\{ \left[ \frac{i}{8}, \frac{i+1}{8} \right] : i = 0, \cdots, 7 \right\}$. We follow Algorithm 3 with $m \in [10, 20, \cdots, 100]$, $n = 50m$, and $n_0 \approx n^{\frac{1}{2r+s}} (\log_{10} n)^3$. With $n_0 \approx n^{\frac{1}{2r+s}} (\log_{10} n)^3$, we also achieve the same bound (with different prefactors) in Theorem 3.4. In the main experiments, we set $\rho_{\mathbf{x}} = \tilde{\rho}_{\mathbf{x}}$.

## C.1.2 Toy-3D

Let $\mathcal{X} = [0,1]^3 \subset \mathbb{R}^3$ and $\rho_\mathbf{x}$ be the uniform distribution on $\mathcal{X}$. Define a kernel $k(\mathbf{x}, \mathbf{y}) = (1 - \|\mathbf{x} - \mathbf{y}\|_{\mathbb{R}^3})^2_+$. The reproducing kernel Hilbert space $\mathbb{H}_k$ associated to the kernel $k$ is norm-equivalent to Sobolev space $H^2(\mathcal{X})$ [55] and the eigenvalue decay of $T_{k,\rho_\mathbf{x}}$ is $s = \frac{3}{4}$ [68]. Note that $k'(\mathbf{x}, \mathbf{y}) = (1 - \|\mathbf{x} - \mathbf{y}\|_{\mathbb{R}^3})^6_+ (35\|\mathbf{x} - \mathbf{y}\|^2_{\mathbb{R}^3} + 18\|\mathbf{x} - \mathbf{y}\|_{\mathbb{R}^3} + 3)$ is a kernel and its reproducing kernel Hilbert space is norm-equivalent to Sobolev space $H^4(\mathcal{X})$. Using the interpolation relation between $H^2(\mathcal{X})$ and $H^4(\mathcal{X})$ gives that

$$f_0^*(\mathbf{x}) = (1 - \|\mathbf{x}\|_{\mathbb{R}^3})^6_+ (35\|\mathbf{x}\|^2_{\mathbb{R}^3} + 18\|\mathbf{x}\|_{\mathbb{R}^3} + 3)$$

is in $\mathbb{H}_k$ and the regularity $r$ is 1. Similarly as before, we generate data points from $\rho_\mathbf{x} \cdot \rho_{y|\mathbf{x}}$ such that $\rho_\mathbf{x}$ is the uniform distribution on $\mathcal{X}$ and

$$\rho_{y|\mathbf{x}} = \mathcal{N}(y|f_0^*(\mathbf{x}), 0.44^2).$$

We divide $\mathcal{X}$ into a partition $\mathcal{A} := \{[\frac{k_1}{2}, \frac{k_1+1}{2}] \times [\frac{k_2}{2}, \frac{k_2+1}{2}] \times [\frac{k_3}{2}, \frac{k_3+1}{2}] \subset \mathbb{R}^3 : (k_1, k_2, k_3) \in \{0,1\}^3\}$. Again, we follow Algorithm 3 with $m \in [10, 20, \cdots, 100]$, $n = 50m$, $n_0 \approx n^{\frac{1}{2r+s}}(\log_{10} n)^3$, and $\rho_\mathbf{x} = \tilde{\rho}_\mathbf{x}$ for kernel machine-based algorithms. For neural network-based algorithms, $m = 50$, $n_0 = 2500$, and the other configurations are the same.

## C.1.3 Real World Datasets

**Energy** Energy dataset is a real-world tabular dataset from the UCI database [12]. It has 28 input features including measurement time, temperature and humidity of each room, outside temperature, and wind speed. The output is the appliances energy use. We normalize all features, including the output, using MinMaxScaler. There are 12,000 training data points distributed across the parties. We use 6,000 samples as public inputs and 1,000 samples for testing. To construct a non-i.i.d. setting, we set a partition $\mathcal{A}$ consisting of 8 subsets, each formed by splitting three normalized variables (measurement time, visibility, and dewpoint) at their midpoints. We apply Algorithm 3 with $m = 50$.

**RotatedMNIST** RotatedMNIST is a dataset derived from MNIST [11]. The task is to predict the rotated angle of a given rotated MNIST image. Each image is $1 \times 28 \times 28$ image, and we normalize all images by their mean and variance. To generate RotatedMNIST, we rotate MNIST images by a random angle between $-\frac{\pi}{2}$ and $\frac{\pi}{2}$ and use the angle as the label. To construct a large-scale dataset, each image is rotated at multiple angles to generate multiple data instances. For training data, we additionally inject Gaussian noise with a standard deviation of $0.2$ to each label. We use 200,000 images as the entire training data, 50,000 images as public inputs, and 50,000 images as test data. The whole training input distribution, public input distribution, and test input distribution are uniformly distributed across digits 0 to 9. For example, there are 20,000 rotated images of the digit '4' in the training set. For a non-i.i.d. setting, we partition the data into $\mathcal{A}$ where $|\mathcal{A}| = 10$, based on the digit $(0 \sim 9)$ and follow Algorithm 3 with $m = 50$.

**UTKFace** UTKFace dataset [71] is an image dataset used for age estimation. Since the image sizes vary, we resize all images to $3 \times 128 \times 128$ and normalize them by their mean and variance for each channel. The labels are normalized to the range $[0, 1]$ using MinMaxScaler. For training data, we inject Gaussian noise with a standard deviation of $0.5$ to each label before normalization. We use 12,544 samples for training and 1,039 samples for testing. We have 6,234 public inputs. These three datasets have the same distribution for metadata (gender and race). Based on this metadata, we construct a partition $\mathcal{A}$ with $|\mathcal{A}| = 10$ and distribute the training data among 50 parties according to Algorithm 3.

**IMDB-WIKI** IMDB-WIKI dataset [52] is also an image dataset for age estimation. In experiments, we utilize a clean version [42]. We further resize all images to $3 \times 64 \times 64$ and normalize them as UTKFace. The labels are also normalized to the range $[0, 1]$ using MinMaxScaler. We use 147,107 images as the entire training data, 36,780 images as public inputs, and 56,087 images as test data. In this dataset, we utilize the triplet (head_roll, head_yaw, head_pitch) as metadata. Both training inputs and public inputs have the same distribution for metadata. We construct a decentralized setting among 50 parties using a partition $\mathcal{A}$ based on the metadata, following Algorithm 3. The partitioning is performed by dividing the dataset into regions based on the median values of each metadata variable.

Table 4: Hyperparameters $C$ and $D$ on kernel machine-based algorithms in main results

|        | centralKRR  | centralKRGD | DC-NY       | DKRR-NY-CM  | IED         | DCL-KR     |
|--------|-------------|-------------|-------------|-------------|-------------|------------|
| Toy-1D | $C = 0.055$ | $D = 15$    | $C = 0.006$ | $C = 0.008$ | $C = 0.025$ | $D = 2.5$  |
| Toy-3D | $C = 0.016$ | $D = 50$    | $C = 0.002$ | $C = 0.005$ | $C = 0.007$ | $D = 12.5$ |

## C.2 Implementation

The experiments are implemented in PyTorch. We simulate a decentralized setting on a single deep learning workstation (Intel(R) Xeon(R) Gold 6430 with one NVIDIA GeForce RTX 4090 GPU and 189GB RAM). All DCL-KR implementations take less than 10 minutes per simulation. The non-parallel implementation of DCL-NN for large-scale datasets is completed within 48 hours. With parallel computing, the execution time for the same setup is expected to be reduced to within 2 hours.

## C.3 Experimental Setup and Results on Kernel Machine-based Algorithms

### C.3.1 Experimental Setup Details and Main Results

In this experiment, we evaluate DCL-KR by comparing its performance against two central models and three baselines. Specifically, we employ DC-NY [66], DKRR-NY-CM [67], and IED [48] as baselines. We also compare DCL-KR with central Kernel Ridge Regression (centralKRR) and central Kernel Regression with Gradient Descent (centralKRGD). As mentioned earlier, we evaluate the performance of these algorithms on Toy-1D and Toy-3D datasets.

There are several hyperparameters for kernel machine-based algorithms: the ridge regularization hyperparameter $\lambda$ in the ridge regressions and the number of iterations (or communication rounds) in the gradient descent-based regressions. We set $\lambda = C \cdot n^{-\frac{1}{2r+s}}$ and $T = \text{int}(D \cdot n^{\frac{1}{2r+s}})$ which are the optimal choices from theory. We determine the best values for $C$ and $D$ by grid search in our experiments (see Table 4 for the selected hyperparameter values). For centralKRGD and DCL-KR, we set the learning rate $\eta = 0.5$ which satisfies $\eta \in (0, 1/\kappa^2)$ in Theorem 3.4. The number of local iterations $E$ for DCL-KR is set to 5. For DKRR-NY-CM, we modify its Newton-Raphson iteration as shown below due to an instability issue, and we set the learning rate $\eta = 0.01$:

$$u \leftarrow u - \eta \cdot \sum_{j=1}^{m} \frac{n_j}{n} (P_Z T_{k,X_i} P_Z + \lambda I)^{-1} ((P_Z T_{k,X} P_Z + \lambda I)u - P_Z S_D^\top \mathbf{y}).$$

The communication round $T$ for DKRR-NY-CM is set to 10.

To measure performance, we sample data from the distribution presented in Appendix C.1 for each simulation and compute $\mathbb{E}\|\iota_{\rho_{\mathbf{x}}}(f_{i,T} - f_0^*)\|_{L_{\rho_{\mathbf{x}}}^2}$ by averaging the Root Mean Squared Errors (RMSEs) on the test dataset. We conduct 500 simulations for each setting, and the results are summarized in Figure 1.

As shown in Table 1, DC-NY and DKRR-NY-CM have theoretical performance guarantees under the statistically homogeneous condition for a limited number of parties. However, they do not exhibit sufficiently good performance in massively distributed statistically heterogeneous settings. The performance degradation of DKRR-NY-CM appears to be linked to its second-order optimization scheme, which leads to ineffective batching in statistically heterogeneous settings. The performance degradation of DC-NY is expected, given the inherent limitations of divide-and-conquer algorithms.

In contrast, IED demonstrates relatively better performance, despite the strong assumptions underlying its theory. Nevertheless, it still exhibits performance degradation compared to centralized models. DCL-KR, on the other hand, achieves performance comparable to centralized models, validating both the theoretical results and its practical feasibility.

### C.3.2 Effect of $n_0$

As public inputs directly affect the training information sharing, we anticipate that the performance of DCL-KR will vary depending on the number of public inputs $n_0$. To examine this effect, we

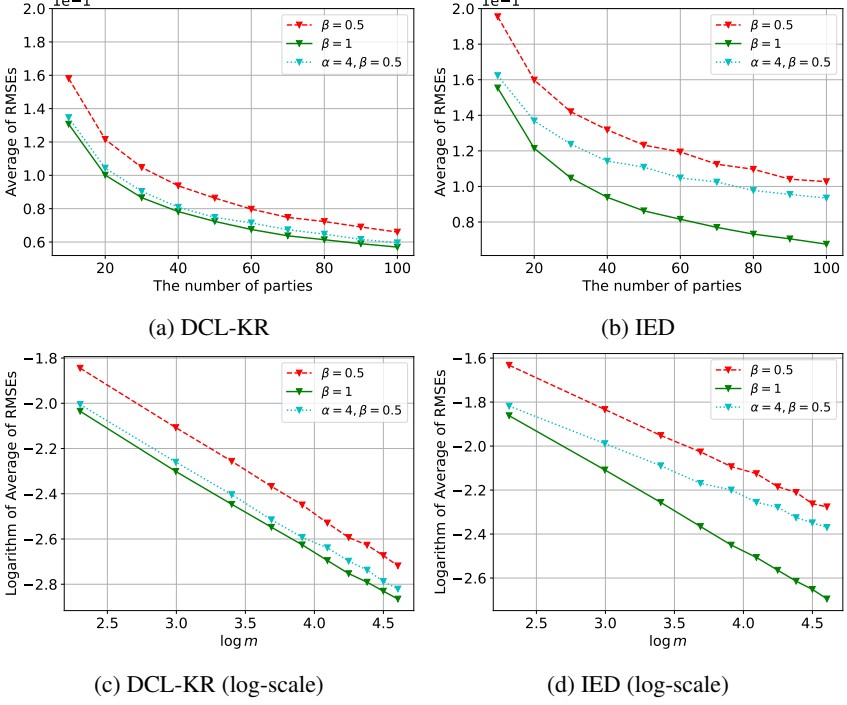

(a) DCL-KR

(b) IED

(c) DCL-KR (log-scale)

(d) IED (log-scale)

Figure 5: Performance of IED and DCL-KR with $\tilde{\rho}_{\mathbf{x}} \neq \rho_{\mathbf{x}}$ on Toy-3D

measure the performance of DCL-KR on Toy-3D for $n_0 \approx \alpha \cdot n^{\frac{1}{2r+s}}(\log_{10} n)^3$ with various $\alpha \in \{0.1, 0.3, 0.5, 1, 2\}$. Additionally, we conduct the same experiment with IED, the most competitive baseline, for comparison.

The results are summarized in Figure 2. Consistent with theoretical results, the value of $\alpha$ does not affect the convergence rate of IED and DCL-KR. However, IED displays significant performance variations across different $\alpha$ values. In contrast, DCL-KR achieves its maximum performance when $\alpha$ is not too small (i.e., $\alpha \geq 0.3$ in Figure 2). This implies that DCL-KR requires fewer public inputs to achieve its maximal performance compared to IED, as predicted by theoretical results.

### C.3.3 Effect of $\tilde{\rho}_{\mathbf{x}}$

So far, we consider the settings with $\rho_{\mathbf{x}} = \tilde{\rho}_{\mathbf{x}}$. However, Theorem 3.4 covers the general case where $\tilde{\rho}_{\mathbf{x}} \neq \rho_{\mathbf{x}}$. To verify this, we define the public input distribution in Toy-3D with the following density function (parametrized by $\beta$):

$$p(x_1, x_2, x_3 | \beta) = \prod_{i=1}^{3}((2 - 2\beta)x_i + \beta), \quad (x_1, x_2, x_3) \in [0, 1]^3, \quad \beta \in (0, 1].$$

The Radon-Nikodym derivative $\frac{d\rho_{\mathbf{x}}}{d\tilde{\rho}_{\mathbf{x}}}$ satisfies $0 \leq \frac{d\rho_{\mathbf{x}}}{d\tilde{\rho}_{\mathbf{x}}} \leq (\frac{1}{\beta})^3$. We conduct additional experiments to verify Theorem 3.4 and Corollary A.7, considering the case where $\beta = 0.5$ (with $\alpha = 1$) and the case where $\beta = 0.5$ but $\alpha = 4$ to compensate. The results are provided in Figure 5. In the log-scale plot, the slope represents the convergence rate.

First, we observe that the convergence rate of DCL-KR remains unchanged when $\beta$ is changed from 1 (i.e., $\rho_{\mathbf{x}} = \tilde{\rho}_{\mathbf{x}}$) to 0.5. This observation is consistent with Theorem 3.4. Additionally, regarding Corollary A.7, we confirm that DCL-KR achieves performance almost identical to the case of $\rho_{\mathbf{x}} = \tilde{\rho}_{\mathbf{x}}$ by increasing $n_0$. In contrast, the convergence rate of IED worsens when $\tilde{\rho}_{\mathbf{x}}$ changes, even when $n_0$ is increased. These experimental results highlight the advantages of DCL-KR in statistically heterogeneous environments.

Table 5: Hyperparameters for Standalone

|  | Toy-3D | Energy | MNIST | UTKFace | IMDB |
|---|---|---|---|---|---|
| batch size | 10 | 16 | 128 | 16 | 32 |
| learning rate | 1e-2 | 2e-2 | 5e-3 | 1e-4 | 5e-3 |

Table 6: Hyperparameters for FedMD

|  | Toy-3D | Energy | MNIST | UTKFace | IMDB |
|---|---|---|---|---|---|
| communication rounds | 500 | 100 | 50 | 50 | 50 |
| sample size of public data | 500 | 1000 | 5000 | 2000 | 5000 |
| learning rate | 2e-4 | 1e-4 | 1e-4 | 5e-5 | 2e-4 |
| local epochs | 10 | 10 | 5 | 5 | 5 |
| distillation epochs | 5 | 1 | 20 | 20 | 20 |
| batch size (local) | 10 | 10 | 32 | 16 | 32 |
| batch size (public) | 32 | 32 | 16 | 16 | 32 |

Table 7: Hyperparameters for FedHeNN

|  | Toy-3D | Energy | MNIST | UTKFace | IMDB |
|---|---|---|---|---|---|
| communication rounds | $\leq 100$ | $\leq 100$ | 50 | 100 | $\leq 50$ |
| sample size of public data | 500 | 500 | 5000 | 1000 | 5000 |
| learning rate | 2e-4 | 2e-4 | 1e-4 | 1e-4 | 5e-4 |
| distillation coefficient | 1 | 1 | 1 | 0.1 | 1 |
| batch size (local) | 10 | 10 | 32 | 10 | 16 |
| batch size (public) | 32 | 32 | 16 | 32 | 16 |

## C.4 Experimental Setup and Results on Neural Network-based Algorithms

### C.4.1 Experimental Setup Details

In the experiments on neural network-based collaborative learning algorithms, we evaluate three baselines (FedMD, FedHeNN, KT-pFL) and our algorithm DCL-NN. Note that while KT-pFL is a personalized collaborative learning algorithm, it also performs well in non-personalized settings, so we include it for comparison. Additionally, we evaluate centralized models as ideal cases and standalone models as worst cases. Centralized models are trained using all local data.

We use two tabular datasets, Toy-3D and Energy, and three image datasets, RotatedMNIST, UTKFace, and IMDB-WIKI. The number of parties is set to 50 for all settings. For the tabular datasets, we employ four different fully connected neural networks (FNNs) with a ratio of 30%, 30%, 20%, and 20%. Specifically, There are fifteen 4-layer FNNs with 32 hidden units, fifteen 4-layer FNNs with 64 hidden units, ten 5-layer FNNs with 32 hidden units, and ten 3-layer FNNs with 64 hidden units. Similarly, for the image datasets, we use four different convolutional neural networks (CNNs) with the same ratio of 30%, 30%, 20%, and 20%. For the large-scale image datasets (RotatedMNIST and IMDB-WIKI), we construct 50 local parties using fifteen ResNet-18, fifteen ResNet-34, ten ResNet-50 [20], and ten MobileNetv2 [54]. For UTKFace, which is an image dataset with limited data, we utilize four simpler CNN architectures due to the ineffectiveness of knowledge distillation with large underperforming models. The first and third CNNs share a similar architecture, featuring two convolutional layers with batch normalization [22], two max pooling layers, and two fully-connected layers at the end. They differ only in the number of channels. In contrast, the second and fourth CNNs are more complex, featuring four convolutional layers with batch normalization, two max pooling layers, and two fully-connected layers at the end. They also differ in the number of channels. We use the ReLU activation function throughout.

Table 8: Hyperparameters for KT-pFL

|  | Toy-3D | Energy | MNIST | UTKFace | IMDB |
|---|---|---|---|---|---|
| communication rounds | 50 | $\leq$50 | 100 | 50 | $\leq$50 |
| sample size of public data | 500 | 500 | 5000 | 2000 | 1000 |
| learning rate | 2e-4 | 1e-4 | 1e-4 | 1e-4 | 2e-4 |
| distillation epochs | 2 | 1 | 10 | 2 | 10 |
| batch size (local) | 10 | 10 | 32 | 16 | 16 |
| batch size (public) | 32 | 16 | 32 | 16 | 16 |

Table 9: Hyperparameters for DCL-NN

|  | Toy-3D | Energy | MNIST | UTKFace | IMDB |
|---|---|---|---|---|---|
| epochs (kernel matching) | 100 | 100 | 200 | 200 | 200 |
| base learning rate (local) | 5e-2 | 1e-2 | 8e-3 | 8e-3 | 8e-3 |
| local epochs | 50 | 50 | 25 | 25 | 25 |

All optimizers used are Adam [26].[2] One remark is that baseline algorithms only utilize a subset of public inputs through random sampling in each communication round, as performance tends to deteriorate due to overfitting when all public inputs are used in every round. Hyperparameters are tuned via grid search.

Standalone models are trained with cross-validation and early stopping to prevent overfitting. To evaluate centralized models, we first compute the averaged test Root Mean Squared Error (RMSE) from at least 10 simulations for each neural network architecture and then calculate the weighted average of the performances of all architectures according to their ratio. For standalone models, we use the average of the test RMSEs of local models with the hyperparameters listed in Table 5. In the table, RotatedMNIST is abbreviated as MNIST, and IMDB-WIKI is abbreviated as IMDB.

For FedHeNN, we set the number of local epochs to 30 in all experiments. For KT-pFL, we set the number of local epochs to 10, the distillation coefficient to 0.5, and the learning rate of knowledge coefficient to 1e-3. Lastly, for DCL-NN in the main experiment, we set the learning rate for kernel matching to 1e-4 and the number of communication rounds in the collaborative learning phase to 50. We use the closed-form solution to train public data and full-batch gradient descent to train local data in the collaborative learning phase of DCL-NN and utilize FedMD for pretraining in DCL-NN. In the pretraining phase, the hyperparameters are the same as in the FedMD setting, except that the number of communication rounds is 100 for tabular data and 50 for image data. The remaining hyperparameters are presented in Table 6, 7, 8, and 9. For all distillation-based collaborative learning algorithms, we simulate each setting at least 5 times with different initializations.

**Communication Efficiency**  Compared with FedMD and KT-pFL, DCL-NN incurs higher communication costs of $O(n_0^2)$ due to the transmission of the Gram matrix. FedHeNN also utilizes kernel matching but performs it in batches for each communication round. Thus, in scenarios requiring many communication rounds, DCL-NN is more efficient than FedHeNN. However, pretraining also demands more communication cost. We leave the study of communication-efficient methods in DCL-NN for future work.

### C.4.2  Effect of Public Inputs

In practice, the public inputs can be sampled from a distribution whose support is disjoint from that of the whole local input distribution. In this case, the assumption of DCL-KR does not hold; however, DCL-NN can still be applicable. To evaluate the performance of DCL-NN under these conditions, we compare the performance of DCL-NN and FedMD when the distribution of public inputs differs, as in Table 10. We use CIFAR10 [28] for public inputs on UTKFace. As shown in Table 10, DCL-NN

---

[2]Note that while DCL-NN should use vanilla gradient descent according to DCL-KR, Adam performs better in practice.

Table 10: Performance comparison of FedMD and DCL-NN on UTKFace with different public datasets. In addition to performance, the kernel performance of local feature kernels, computed in the same way as before, is shown in parantheses.

|  | FedMD | DCL-NN |
|---|---|---|
| w/ UTKFace | $0.151 \pm 0.004$ (0.149) | **0.148 $\pm$ 0.001** (**0.146**) |
| w/ CIFAR10 | **0.160 $\pm$ 0.000** (0.159) | $0.162 \pm 0.001$ (**0.158**) |

does not yield better results in this case. Note that the kernel performance of local feature kernels is improved compared to FedMD, leading us to conclude that the performance degradation of DCL-NN with CIFAR10 is due to the violation of the DCL-KR assumption rather than the ineffectiveness of the kernel distillation procedure.

## D   Limitations and Future Works

**Privacy Benefits of Distillation-based Collaborative Learning**   Due to its black-box nature, distillation-based information interaction is expected to offer privacy preservation benefits compared to parameter exchange (mainly done in FL) as mentioned in [19]. To the best of our knowledge, there is no rigorous study that discusses the privacy preservation advantages of distillation-based collaborative learning. We hope to see further discussion on this as well.

**Public Input Distribution**   Theorem 3.4 covers the case where the public input distribution $\tilde{\rho}_{\mathbf{x}}$ differs from that of local data inputs $\rho_{\mathbf{x}}$, but at least the support of $\tilde{\rho}_{\mathbf{x}}$ must include the support of $\rho_{\mathbf{x}}$. Therefore, we experimentally observe a performance drop in DCL-NN, a practical extension of DCL-KR, when $\tilde{\rho}_{\mathbf{x}}$ and $\rho_{\mathbf{x}}$ have different supports. Enhancing the robustness of DCL-NN in this scenario is considered a promising direction for future work. Our theory does not cover situations where collecting public inputs is difficult. In such cases, a seperate generative model is usually trained to generate public inputs [72]. We leave the theoretical discussion that includes these cases for future work.

