# OpenReview forum: "A Kernel Perspective on Distillation-based Collaborative Learning"
_NeurIPS.cc/2024/Conference — NeurIPS 2024 poster_

### Official Review · Reviewer_Zad8 · 2024-07-08

**Soundness:** 2
**Presentation:** 1
**Contribution:** 3
**Rating:** 5
**Confidence:** 4

**Summary:**

This work presents an analysis of a distillation-based collaborative learning algorithm called FedMD from a nonparametric perspective. The method is to adopt an operator theoretic approach to obtain an upper rate for the expected generalization error of local models. Then, the authors propose DCL-KR to achieve a privacy-preserving nearly minimax optimal collaborative learning algorithm with kernel regression in massively distributed statistically heterogeneous environments.

**Strengths:**

+ The idea seems to be interesting.
+ The first work tries to prove the (nearly) minimax optimality of a privacy-preserving collaborative learning algorithm.

**Weaknesses:**

- The paper's organization and structure are not clear.
- The writing looks like combining different parts of different papers without a clear clue.
- The paper suddenly gives many assumptions in Section 3 for analysis but does not give any intuition or insight into them.
- Some important experimental results are in the appendix. I would suggest that the authors extract insight into the experimental findings in the main body.
- The language used in the paper is not concise.

**Questions:**

The related work only gives a rough summary of the previous works. When the authors focus on the theoretical analysis, could the authors compare their theoretical bounds to other works?

There are many important experiments in the appendix, such as C.3.1. Can the authors summarize empirical findings with a better paper structure?

In Section 3.3, the paper starts with "To derive theoretical results" and lists a lot of assumptions without any intuitive understanding behind them. Could the authors give an overview and intuition of theoretical results?

Minor: I feel very confused when reading the paper as it introduces many conclusions/results without too much explanation. I suggest the authors elaborate on the intuitions before going into the details.

**Limitations:**

The paper presents a discussion of limitations, which is not convincing. For example, "there is no rigorous study that discusses the privacy preservation advantages of distillation-based collaborative learning". Actually, the standard technique for privacy is not distillation.

---

> ### Author Rebuttal · Authors · 2024-08-05
>
> We appreciate the reviewer for the detailed comments and hope to address all of the questions and concerns raised by the reviewer.
>
> > The paper's organization and structure are not clear.
> > The writing looks like combining different parts of different papers without a clear clue.
>
> We attempted to write the manuscript well, but it seems that our manuscript did not effectively convey its content to the reviewer. Please refer to the general response. Our general response contains the outline of the overall flow of our manuscript and an additional explanation on the connection between DCL-KR and DCL-NN. Through this general response, we hope that the reviewer finds our manuscript to have a natural flow.
>
> > In Section 3.3, the paper starts with “To derive theoretical results” and lists a lot of assumptions without any intuitive understanding behind them. Could the author give an overview and intuition of theoretical results?
>
> We are sorry to hear that the reviewer had a difficulty to read our theoretical analysis. In the future version, we will provide overview and intuition of theoretical results at the beginning of Section 3.3 as follows:
> * In this subsection, we prove the nearly minimax optimality of DCL-KR;
> * This result implies that DCL-KR has an almost same convergence rate as the minimax optimal central training when there are sufficiently many public inputs;
> * To the best of our knowledge, this is the first work to prove the nearly minimax optimality of a privacy-preserving collaborative learning algorithm with kernel regression in massively distributed statistically heterogeneous environments (Lines 179 – 181 in our manuscript).
>
> We also provide a detailed explanation of assumptions from Section 3 in the general response. Please refer to the general response for details on this matter.
>
> > There are many important experiments in the appendix, such as C.3.1. Can the authors summarize empirical findings with a better paper structure?
>
> Due to the page limitation, we only included some of the experimental results and discussions in the main body. However, we agree with the reviewer that some important experimental discussions should be included in the main text to convey meaningful content and improve its structure. If permitted, we will utilize an additional page to incorporate the following results from the appendix into the main body.
> * We will include Figure 3, 4, and 5(c)-(d) in the main body.
>
> We will include the following discussions in the main text:
> * As shown in Figure 3, DCL-KR outperforms the baselines in all experimental settings and achieves comparable performance to the central models. In contrast, DC-NY and DKRR-NY-CM exhibit significantly lower performance compared with DCL-KR in massively distributed environments where their theory does not cover.
> * IED does not show a significant performance drop in massively distributed environments. However, as predicted by the theoretical results, to perform well, IED requires more public inputs compared with DCL-KR. Moreover, when there is a public distribution shift, the convergence rate of DCL-KR is maintained, whereas that of IED worsens. We can also confirm that DCL-KR can address this distribution shift effectively with a large amount of public inputs.
> * Overall, our experiments validate the theoretical results of DCL-KR and demonstrate its superiority over previous results.
>
> > The language used in the paper is not concise.
>
> If allowed, we will make effort to revise the manuscript to ensure it is concise and clear.
>
> > The related work only gives a rough summary of the previous works. When the authors focus on the theoretical analysis, could the authors compare their theoretical bounds to other works?
>
> Indeed, the previous works similarly aim to achieve (nearly) minimax optimality (i.e., prove inequalities that are similar to (2) in our manuscript). Thus, there is no difference in the bound itself except for logarithm rates and prefactors. Note that the prefactors are not considered in the analysis of minimax optimality. In addition, $\log n$ grows slower than any polynomial.
>
> The main differences lie not in the theoretical bound itself but in the decentralized environment settings that guarantee the bound, as summarized in Table 1. Some prior works have limitations on the number of local parties. For example, DKRR-NY-CM assumes $m\leq O(n^{(2r+s-1)/(2r+s)})$ (with the same notation in our manuscript). In contrast, nFedAvg directly communicates local data and IED does not consider statistically heterogeneous cases. We elaborate on these points in lines 105 – 117 and lines 179 – 209. Our experimental results also demonstrate performance degradation of baselines when they are applied outside their assumed settings.
>
> > The paper presents a discussion of limitations, which is not convincing. For example, "there is no rigorous study that discusses the privacy preservation advantages of distillation-based collaborative learning". Actually, the standard technique for privacy is not distillation.
>
> As the reviewer pointed out, distillation is not a standard technique for privacy. However, as mentioned in Section 1, Distillation-based Collaborative Learning (DCL) algorithms have been proposed in the context of Federated Learning (FL) where local data privacy preservation is crucial. This is because, as mentioned in Lines 1383 – 1385, the communicated information is predictions on public data, which is expected to preserve local data privacy due to its black box nature. Initially, traditional FL algorithms that exploit parameter exchange also do not provide clear evidence for local data privacy concerns, but recent discussions address these issues extensively. In summary, we emphasize in Appendix D that DCL has the potential to preserve local privacy and that there is a need to study privacy concerns in DCL, similar to the research flow in traditional FL algorithms.

---

> > ### Comment · Reviewer_Zad8 · 2024-08-09
> >
> > Thanks for the detailed explanation and response. After reading the whole page, I am willing to improve my rating based on trusting the authors to modify the paper.
> >
> > Thank you very much.

---

### Official Review · Reviewer_UpcW · 2024-07-09

**Soundness:** 3
**Presentation:** 2
**Contribution:** 3
**Rating:** 6
**Confidence:** 3

**Summary:**

The authors propose a nonparametric version of FedMD, a distillation-based collaborative learning methodology. They also propose a neural network variant of the nonparametric approach as an extension for heterogeneous local neural network models. They provide both theoretical results on the nonparametric approach and experimental results demonstrating efficacy of the neural network extension. The theoretical results prove the nearly minimax optimality of this nonparametric collaborative learning algorithm in massively distributed statistically heterogeneous environments.

**Strengths:**

(1) The authors prove the (nearly) minimax optimality of a privacy-preserving collaborative learning algorithm using kernel regression in massively distributed, statistically heterogeneous environments. (2) Theorem 3.4 addresses a more general setting than prior works [48, 58]. Su et al. [58] only cover r = \frac{1}{2}​ of Assumption 3.3. Park et al. [48] do not consider Assumption 3.2, which provides a finer result. Furthermore, compared to [48], the required size of public inputs is reduced and the statistical homogeneity condition is dropped. (3) The rate n^{-\frac{r}{2r+s}} is the minimax lower rate under Assumptions 3.1, 3.2, and 3.3, making DCL-KR nearly optimal in a minimax sense. (4) On the difference between public and local distributions: Theorem 3.4 allows the public input distribution \tilde{\rho}_x to differ from the local input distribution \rho_x​. (5) Finally, The difference between \rho_x​ and \tilde{\rho}_x​ impacts the expected risk upper bound as a multiplication of B_r​, which can be removed by increasing public inputs.

**Weaknesses:**

(1) The need to perform learning rate scaling to ensure the impact of local iterations is consistent across models introduces additional complexity. The exact method for scaling may vary and might not be trivial to determine or implement. (2) The computation of Gram matrices for CKA involves O(p^2) operations, where p is the number of public data points. This quadratic complexity can become prohibitive as the size of the public dataset increases. (3) The method assumes that the ensemble kernel k = \sum_{i=1}^{m} \frac{n_i}{n} k_{f_i} will perform better than individual feature kernels. This assumption might not hold in cases where local models are highly heterogeneous, as the averaged kernel could lose important characteristics of individual models, leading to suboptimal performance. The authors empirically verify the superiority of the ensemble in the appendix.

**Questions:**

(1) How do you handle the computational burden associated with the calculation of Gram matrices for large datasets? (2) How do you handle cases where these representations before the last layers of the neural networks are suitable for kernel matching?

---

> ### Author Rebuttal · Authors · 2024-08-05
>
> We appreciate the reviewer for the detailed comments and hope to address all of the questions and concerns raised by the reviewer.
>
> > (1) The need to perform learning rate scaling to ensure the impact of local iterations is consistent across models introduces additional complexity. The exact method for scaling may vary and might not be trivial to determine or implement.
>
> As the reviewer pointed out, we calculate and utilize the empirical CKA on the whole public inputs of local feature kernels for learning rate scaling. We use this approach for precise calculation, but we agree that it can be computationally intensive. Therefore, we additionally conduct experiments on RotatedMNIST by sampling a few public inputs and calculating the empirical CKA based on them. The experimental results are as follows:
>
> |GD, 50 data points used|GD, full data points used|
> |---|---|
> |$0.243\pm 0.006$|$0.243\pm 0.006$|
>
> We can observe that when using vanilla gradient descent, calculating empirical CKA with 50 public inputs is sufficient for learning rate scaling. When using Adam, we obtain the following results, also indicating that calculating empirical CKA with 50 samples is sufficient.
>
> |Adam, 50 data points used|Adam, full data points used|
> |---|---|
> |$0.227\pm 0.003$|$0.227\pm 0.003$|
>
> Consequently, calculating empirical CKA for learning rate scaling can be done with a small number of public inputs and then it does not require extensive computational resources. We will include these results in the manuscript.
>
> > (2) The computation of Gram matrices for CKA involves O(p^2) operations, where p is the number of public data points. This quadratic complexity can become prohibitive as the size of the public dataset increases.
>
> We think this is a good point. The kernel distillation procedure is a crucial component in enhancing performance by maintaining the theoretical assumptions of DCL-KR in the neural network setting, so calculating the gram matrix is unavoidable. However, we can apply some tricks to compute it more efficiently.
>
> For example, we can perform this calculation on the server to reduce the computational burden imposed on local parties. First, the local parties directly upload the raw features of public inputs (whose dimension is $n_0 \times d$ where $n_0$ is the number of public inputs and $d$ is the dimension of the local feature) to the server. Then the server can compute the feature kernels. Since this approach only requires the local party to perform forward propagation for the public inputs, it significantly reduces the computational cost on the local parties. Additionally, this method can also reduce communication costs when dealing with a large amount of public data. For reducing the overall computational burden (including the server), we believe that an additional study is needed and we will strive to address this issue in future work.
>
> > (3) The method assumes that the ensemble kernel k = \sum_{i=1}^{m} \frac{n_i}{n} k_{f_i} will perform better than individual feature kernels. This assumption might not hold in cases where local models are highly heterogeneous, as the averaged kernel could lose important characteristics of individual models, leading to suboptimal performance. The authors empirically verify the superiority of the ensemble in the appendix.
>
> In fact, the ensemble kernel theoretically has stronger expressivity than each individual local feature kernel, so with a sufficiently large amount of data, a better regressor will be trained when we use the ensemble kernel. We demonstrate this in Appendix B (lines 1142 - 1145). Therefore, with enough data, it is not possible for the ensemble kernel to perform worse than individual local feature kernels.
> However, fully distilling the ensemble kernel is challenging. Thus, there is a possibility of losing important characteristics (that the reviewer concerned with) during the kernel distillation process. Precisely, some important features may have a lower portion in the ensemble kernel and so they can be potentially missed during the distillation process. One possible solution is for each local party to construct local feature kernels by selecting important features and adjusting weights accordingly.
>
> > (1) How do you handle the computational burden associated with the calculation of Gram matrices for large datasets?
>
> Please refer to our responses above.
>
> > (2) How do you handle cases where these representations before the last layers of the neural networks are suitable for kernel matching?
>
> In our work, we assume that the local model is of the form $f(\cdot) = w^\top g(\cdot) + b$ and design our algorithm accordingly. All neural network architectures with this form (that is, the last layer is a fully connected layer) are suitable for kernel matching. Note that most modern neural networks have this structure. Therefore, in most cases, our algorithm is applicable.
>
> If other forms of neural networks need to be utilized, we anticipate that significant changes to the algorithm would be required. Fundamentally, DCL-NN is inspired by DCL-KR. DCL-KR is based on kernel regression (a type of linear model) and this nature plays a crucial role in its theoretical results. Exploring the ways to generalize the extension from DCL-KR to neural network settings would be a valuable direction for future work.

---

> > ### Comment · Reviewer_UpcW · 2024-08-13
> >
> > Thanks for the response. This has cleared up most concerns for me.
> >
> > On this comment,
> > >Therefore, with enough data, it is not possible for the ensemble kernel to perform worse than individual local feature kernels. However, fully distilling the ensemble kernel is challenging. Thus, there is a possibility of losing important characteristics (that the reviewer concerned with) during the kernel distillation process. Precisely, some important features may have a lower portion in the ensemble kernel and so they can be potentially missed during the distillation process. One possible solution is for each local party to construct local feature kernels by selecting important features and adjusting weights accordingly.
> >
> > This is particularly what I was aiming at. On re-reading my review, I realise I was not clear.
> > I agree with enough data the ensembled kernel cannot be worse than the local kernels. I agree the problem here is the distillation process and that piece should be explored in future work. There is work in leveraging bochner's theorem for learning kernels from data in the gaussian process literature. Perhaps these approaches could provide more optimal kernels at the local level.
> >
> > I've updated my score given this response.

---

### Official Review · Reviewer_UU4M · 2024-07-11

**Soundness:** 3
**Presentation:** 3
**Contribution:** 3
**Rating:** 6
**Confidence:** 3

**Summary:**

The paper investigates the theoretical underpinnings and practical implementation of distillation-based collaborative learning (DCL) from a kernel regression perspective. The authors propose DCL-KR, a nonparametric version of the FedMD algorithm, which achieves nearly minimax optimal convergence rates in massively distributed and statistically heterogeneous environments. Building on these theoretical insights, the authors introduce DCL-NN, a practical DCL algorithm designed for neural networks, which incorporates feature kernel matching to align local models. Extensive experiments on various regression tasks demonstrate the superiority of DCL-KR and DCL-NN over existing methods.

**Strengths:**

1. The paper provides a rigorous theoretical analysis of DCL-KR, proving its nearly minimax optimality in distributed and heterogeneous settings.

2. Experiments are comprehensive. Six datasets are included.

3. The improvement of DCL-NN is significant over the other baselines in some datasets.

**Weaknesses:**

1. The paper lacks a discussion on the privacy risk and communication efficiency of DCL-KR and DCL-NN.

2. The paper is based on distillation-based collaborative learning method that requires public data, which limits the applications on private domains.

**Questions:**

1. How does the choice of kernel affect the performance/theoretical guarantee of the proposed approach?

2. Is transferring predictions a common practice in collaborative learning for regression? If not, how does it compare with the other approaches in terms of privacy and efficiency?

**Limitations:**

Yes.

---

> ### Author Rebuttal · Authors · 2024-08-05
>
> We appreciate the reviewer for the detailed comments and hope to address all of the questions and concerns raised by the reviewer.
>
> > The paper lacks a discussion on the privacy risk and communication efficiency of DCL-KR and DCL-NN.
>
> Basically, our work shares the typical (widely used) assumption of distillation-based federated learning (that the collaborative distillation method, interacting through the predictions on public inputs, can preserve privacy due to its black-box nature). However, as explained in Appendix D, privacy concerns for DCL algorithms have not been sufficiently explored. This issue is not straightforward because it requires fundamental analysis on privacy preservation of functional information transfer through knowledge distillation. This topic seems to be a very interesting topic, but it appears to be beyond the scope of our work and left for a further study.
>
> Regarding communication efficiency, both DCL-KR and kernel regression-based baselines require a communication cost of $O(n_0)$ per communication round (except the download of public inputs) where $n_0$ is the number of public inputs. DCL-NN communicates the gram matrix (with communication cost $O(n_0^2)$) and requires $O(n_0)$ communication cost per communication round in collaborative learning phase. Compared with FedMD and KT-pFL, DCL-NN has higher communication costs due to transmitting the gram matrix. Since FedHeNN also uses kernel matching but performs it in batches for each communication round, in cases that many communication rounds are needed, DCL-NN is more efficient than FedHeNN. However, when we use DCL algorithms for pretraining of DCL-NN, more communication cost is required for pretraining. Therefore, there is a trade-off between performance and communication cost.
>
> To reduce the communication cost of DCL-NN, we can use the following idea: If the feature dimension $d$ of the local model is smaller than $n_0$ (# of public inputs), a direct transmission of the raw feature (whose dimension $n_0 \times d$) or the decomposition of the gram matrix can reduce the communication cost.
>
> > The paper is based on distillation-based collaborative learning method that requires public data, which limits the applications on private domains.
>
> As the reviewer pointed out, there may be cases where it is difficult to collect public data. However, as explained in Section 1, we would like to emphasize that our problem involves prohibiting the direct exchange of not only local data but also model information. These restrictions necessitate an additional information sharing medium.
>
> In this context, our theoretical findings highlight a strength of our algorithm. According to Theorem 3.4 and Corollary A.7, as long as the public data distribution covers the support of the local data distribution, theoretical performance guarantees are provided. Moreover, when the public data has a different distribution from the local data but there is a large amount of public inputs, it can compensate for the distribution gap and ultimately achieve the same performance as central training.
>
> > How does the choice of kernel affect the performance/theoretical guarantee of the proposed approach?
>
> In our theoretical analysis, the kernel is related to the quantity $r$ and $s$ when the target function is given. These quantities determine the minimax lower rate $O(n^{–r/(2r+s)})$ in Theorem 3.4, meaning that the convergence rate varies depending on the choice of kernel. Hence, the faster eigenvalue decay of the kernel (smaller $s$) and the better regularity of the target function (larger $r$) in the RKHS induced by this kernel (which means the RKHS represents the target function well in some sense) give a better performance (faster convergence rate). We can also see that a good choice of the kernel reduces the number of the required public data $(n^{1/(2r+s)}log^3 n)$ in Theorem 3.4.
>
> > Is transferring predictions a common practice in collaborative learning for regression? If not, how does it compare with the other approaches in terms of privacy and efficiency?
>
> In terms of not directly sharing local data, Federated Learning (FL) with parameter exchange can be considered as a similar approach to Distillation-based Collaborative Learning (DCL). However, to the best of our knowledge, in scenarios where neither model nor local data can be directly shared, DCL methods are the only option. As stated in Section 1, our setting is distinguished from traditional FL (that exploits parameter exchange) by the inability to directly share the local model. Therefore, a direct comparison between DCL and traditional FL may not be appropriate in general.
>
> Of course, putting these points aside, we can simply compare DCL with traditional FL in terms of privacy and efficiency. As noted in Appendix D of our manuscript, there is an intuition that DCL may be better than parameter exchange-based FL in terms of privacy. In more detail, parameter exchange has a white-box nature in which the internal information of the model is shared, while the predictions of public data have a black-box nature because the structure of each local model is not known externally. However, we acknowledge that more rigorous discussions on privacy for DCL is required as mentioned earlier.
>
> From an efficiency standpoint, a simple comparison of communication cost is possible. When using large models, DCL-NN (that communicates public inputs and gram matrix only once and shares predictions in each communication round) can be more efficient than communicating model parameters in each communication round.
>
> However, DCL algorithms still exhibit worse performance compared with traditional FL in general. This is because DCL addresses a more challenging problem. We believe that DCL needs to improve its effectiveness before focusing on efficiency improvements, and we expect our work to play a significant role in this aspect.

---

> > ### Comment · Reviewer_UU4M · 2024-08-09
> >
> > Thanks for the author's response. The response has addressed my questions and I'll keep my positive score.

---

### Official Review · Reviewer_1rrx · 2024-07-12

**Soundness:** 2
**Presentation:** 2
**Contribution:** 2
**Rating:** 5
**Confidence:** 1

**Summary:**

This paper theoretically proves a nonparametric version of the most standard distillation based collaborative learning algorithm (named DCL-KR) is nearly minimax optimal in massively distributed statistically heterogeneous environments.
Extensive experiments demonstrate their theoretical results and show the practical feasibility of DCL-NN.

**Strengths:**

1. The paper is well-written.
2. The authors conducted extensive experiments to verify the effectiveness of the proposed method.

**Weaknesses:**

Please see Limitations

**Questions:**

Please see Limitations

**Limitations:**

This is not my field of study and I can only give a Borderline Reject. I will adjust my score based on the scores of other more specialised judges.

---

> ### Author Rebuttal · Authors · 2024-08-05
>
> We appreciate the reviewer for the detailed comments. We briefly describe the main contribution of our work as below:
> * Applying kernel regression theory, we analyze the most representative DCL algorithm, FedMD, from a non-parametric perspective (called DCL-KR). Compared with the existing studies, our work is the first to theoretically demonstrate the effectiveness of DCL algorithm on statistically heterogeneous and massively distributed environments.
> * Based on the theoretical results, we propose a novel neural network-based DCL algorithm (called DCL-NN) using kernel matching.
> * Through experiments, we validate the theoretical results and demonstrate the superiority of our algorithms (DCL-KR and DCL-NN) by comparing its practical performance with baselines.
>
> Kindly let me know if you require any further clarification or have additional comments.

---

> > ### Comment · Reviewer_1rrx · 2024-08-09
> >
> > Thank you for the author's response. I will take into account the feedback from other reviewers to adjust my scoring subsequently.

---

> ### Comment · Reviewer_1rrx · 2024-08-13
>
> After reading feedback from other reviewers, I am willing to improve my rating based on trusting the authors to modify the paper.

---

### Official Review · Reviewer_Dtec · 2024-07-15

**Soundness:** 4
**Presentation:** 4
**Contribution:** 4
**Rating:** 7
**Confidence:** 1

**Summary:**

In this paper, the authors perform a study of distillation-based collaborative learning (DCL) in massively distributed statistically heterogeneous environments. In particular, the study focuses on analyzing DCL-KR, a non-parametric version of FedMD, and proves its near-minimax optimality. The authors proposed DCL-NN, a method that leverages kernel matching to implement DCL with heterogeneous neural networks. Experimental results support the theoretical analysis of DCL-KR and demonstrate the effectiveness of DCL-NN.

**Strengths:**

The paper provides a theoretical analysis of DCL-KR and proposes a practical algorithm for using neural networks in massively distributed statistically heterogeneous environments. The authors perform experiments with both synthetic and real-world datasets to support the theoretical results and evaluate the performance of DCL-NN.

**Weaknesses:**

The paper does not perform any formal privacy analysis. As such, I believe using terms like "privacy-preserving" can be misleading in this context.

**Questions:**

Could the authors avoid mentioning privacy as no formal privacy study is conducted?

**Limitations:**

The paper unnecessarily emphasizes privacy while admitting that privacy is not formally defined (cf. question). It would be nice to provide more discussion on this aspect or to present some empirical evidence of the claimed benefit (e.g., performing privacy attacks such as membership inference)

---

> ### Author Rebuttal · Authors · 2024-08-05
>
> We appreciate the reviewer for the detailed comments and hope to address all of the questions and concerns raised by the reviewer.
>
> > The paper does not perform any formal privacy analysis. As such, I believe using terms like "privacy-preserving" can be misleading in this context.
>
> > Could the authors avoid mentioning privacy as no formal privacy study is conducted?
>
> In our manuscript, we use the term “privacy-preserving” to describe distillation-based collaborative learning (DCL) in the sense that it exchanges the predictions on public data (functional information) rather than directly exchanging local data. As we mentioned in lines 1383 - 1385, we believe that this black-box nature allows DCL to protect local data privacy.
>
> In a similar vein, as mentioned in Section 1 and 2, DCL has been studied mainly in the context of Federated Learning (FL) that aims to preserve local data privacy on decentralized learning. Therefore, we can see that previous studies have also been conducted based on this intuitive (though not rigorous) reasoning.
>
> However, as the reviewer pointed out (and as mentioned in Appendix D of our manuscript), there has not been a rigorous study addressing the privacy concerns of DCL. Fundamentally, we agree that it is necessary to discuss comprehensively the privacy preservation advantage of functional information transfer through knowledge distillation. This seems to be a very interesting topic, but it is beyond the scope of our work and left for a further study.
>
> In summary, our work shares the belief widely assumed in the previous DCL works (particularly in the context of FL), but we understand the reviewer’s concern about using terms like “privacy-preserving” without rigorous discussion. We will strive to soften these expressions in future versions to ensure that readers understand accurately.

---

> ### Comment · Reviewer_Dtec · 2024-08-11
> **Thank you for the responses**
>
> Thank you for the responses. I will keep my positive score.

---

### Author Rebuttal · Authors · 2024-08-05

We appreciate all reviewers for the detailed comments. In this general response, we provide additional explanations to help the reviewers better understand our work. In detail, we (1) briefly outline the overall flow of our manuscript, (2) provide additional explanations on the connection between DCL-KR and DCL-NN, and (3) provide additional explanations on Assumptions introduced in Section 3. If permitted, we will include them in the revised version.

**Overall Flow of Our Manuscript**

As explained in Section 1 of our paper, we first provide a theoretical foundation for a nonparametric version of FedMD [1] (called DCL-KR) and use this result to design its neural network version (called DCL-NN). In detail, we first obtain the nearly minimax optimality of DCL-KR through kernel regression theory. This implies that DCL-KR with sufficiently large public inputs has an almost same convergence rate as minimax optimal central training. (Section 3)

While this theoretical analysis represents significant improvements compared to previous results (Table 1, lines 179 – 209), it does not fully explain the performance guarantee of neural network-based DCL. So, we design a new DCL algorithm (DCL-NN) inspired by the theory of DCL-KR (lines 50-56).

[1] D. Li and J. Wang. Fedmd: Heterogeneous federated learning via model distillation. arXiv preprint arXiv:1910.03581, 2019.

**Additional Explanation on Connection between DCL-KR and DCL-NN**

Indeed, we explicitly state that the equality of local kernels contributes the successful analysis of DCL-KR (lines 215-218) and DCL-NN is a natural extension of DCL-KR in this view. Here, we provide a detailed explanation on this point.

For simplicity, consider the case of $E=1$ in DCL-KR where $E$ is the number of local iterations. After the consensus prediction $u$ is distributed to local parties, the server receives the updated local prediction on $Z$ : $(I-\frac{\eta}{n_i} K_{ZX_i}K_{X_i\tilde{Z}}K_{\tilde{Z}\tilde{Z}}^{-1})u + \frac{\eta}{n_i} K_{ZX_i}y_i$ from the $i$-th local party. (The notation is consistent with the main body and Appendix A.2.1) Suppose the data at the $i$-th local party and the $j$-th local party are exactly the same. If the same kernel is used in these two parties, the updated local predictions will be identical. However, if the kernels of the two parties are different, this will not be the case. For kernels like the Gaussian kernel, which has high correlation between close inputs, the updated local prediction will be strongly influenced more by data points close to each input. We can observe this fact from the above formula. On the other hand, for kernels like the linear kernel, which has high correlation between distant inputs, the updated local prediction on $Z$ will be influenced more by data points farther from each input. This observation implies that aggregating local learning information becomes very challenging when the kernels differ.

In summary, using the same kernel ensures that the shift mechanisms of predictions on $Z$ at edges are identical, which makes it possible for the aggregation through simple weighted averaging to work well. This is a key of the strong theoretical results of DCL-KR. Therefore, DCL-KR and DCL-NN are deeply connected.

**Additional Explanation on Assumptions in Section 3**

Note that, as mentioned in lines 160 – 162, Assumption 3.1, 3.2, and 3.3 are commonly used when deriving the upper rate of expected generalization error. In recent decades, these assumptions have been used as the standard setting in kernel regression analysis. Relevant literature [2, 3] often introduces these assumptions without detailed explanations.

Nonetheless, we acknowledge that these assumptions might seem unfriendly to those unfamiliar with this research context. Therefore, we provide an explanation of these assumptions here.

Basically, as noted in line 160-162, Assumption 3.1 is about the regularity of noise, Assumption 3.2 is about the regularity of the kernel, and Assumption 3.3 is about the regularity of the target function. These assumptions influence the minimax lower rate [4]. In detail,

* Assumption 3.1 implies that the noise is not excessively large. In fact, noise with Bernstein condition satisfies Assumption 3.1. For instance, Gaussian/sub-Gaussian noises and bounded noises satisfy Assumption 3.1. Therefore, this assumption is a very general noise condition that encompasses a wide range of cases.
* Assumption 3.2 is about the eigenvalue decay of the kernel. This is a crucial factor when studying the behavior and properties of kernel. For example, from this assumption, one can derive bounds on the effective dimension that is related to covering and entropy number conditions.
* Assumption 3.3 is related to the regularity of the target function, specifically how well the RKHS induced by the kernel represents the target function.

Under the above assumptions, the minimax lower rate is given by $O(n^{-r/(2r+s)})$. Thus, under these assumptions that precisely specify the minimax lower rate, our work analyzes whether DCL-KR has a similar upper rate. As a result, we demonstrate that DCL-KR is nearly minimax optimal and so we can conclude that DCL-KR has an almost same performance as the central kernel regresion model.

Many prior works also study the minimax optimality of other kernel regression-based algorithms under similar assumptions (Lines 92-117, Table 1, Lines 179 – 209 in our manuscript).

[2] Y. Li, H. Zhang, and Q. Lin. On the saturation effect of kernel ridge regression. In International Conference on Learning Representations, 2023.

[3] S. Park, K. Hong, and G. Hwang. Towards understanding ensemble distillation in federated learning. In International Conference on Machine Learning, pages 27132-27187. PMLR, 2023.

[4] A. Caponnetto and E. De Vito. Optimal rates for the regularized least-squares algorithms. Foundations of Computational Mathematics, 7:331-368, 2007.

---

### Decision · Program_Chairs · 2024-09-25

**Decision:**

Accept (poster)

**Comment:**

The paper studies distillation-based collaborative learning (DCL) in a massively distributed heterogeneous setting. The authors analyze DCL kernel regression and prove its near-minimax optimality in their theoretical results. The paper also presents supporting experiments to demonstrate the effectiveness of the proposed method.

There is a consensus among reviewers that this work is theoretically sound and provides complementary experimental results to back the theoretical analysis. There are several suggestions by the reviewers to improve the paper further, especially the ones regarding the paper organization and the addition of discussions on (i) potential shortcomings if the local data belongs to a private domain without appropriate public data, and (ii) comparison of the theoretical bound in this work to those in prior works.

Overall the reviews consistently agree that the paper makes a significant contribution over the prior work and provides a useful contribution to the domain. I would request the authors to consider the discussions summarized above and appropriately update the draft before the final submission.